# A Non-parametric Regression viewpoint : Generalization of Overparametrized Deep ReLU Network under Noisy Observations

**Namjoon Suh, Hyunouk Ko, Xiaoming Huo**
H.Milton Stewart School of Industrial and Systems Engineering
Georgia Institute of Technology
Atlanta, GA, USA
{namjsuh,hko39,huo}@gatech.edu

## Abstract

We study the generalization properties of the overparameterized deep neural network (DNN) with Rectified Linear Unit (ReLU) activations. Under the non-parametric regression framework, it is assumed that the ground-truth function is from a reproducing kernel Hilbert space (RKHS) induced by a neural tangent kernel (NTK) of ReLU DNN, and a dataset is given with the noises. Without a delicate adoption of early stopping, we prove that the overparametrized DNN trained by vanilla gradient descent does not recover the ground-truth function. It turns out that the estimated DNN's $L_2$ prediction error is bounded away from 0. As a complement of the above result, we show that the $\ell_2$-regularized gradient descent enables the overparametrized DNN to achieve the minimax optimal convergence rate of the $L_2$ prediction error, without early stopping. Notably, the rate we obtained is faster than $\mathcal{O}(n^{-1/2})$ known in the literature.

## 1 Introduction

Over the past few years, Neural Tangent Kernel (NTK) [Arora et al., 2019b; Jacot et al., 2018; Lee et al., 2018; Chizat & Bach, 2018] has been one of the most seminal discoveries in the theory of neural network. The underpinning idea of the NTK-type theory comes from the observation that in a wide-enough neural net, model parameters updated by gradient descent (GD) stay close to their initializations during the training, so that the dynamics of the networks can be approximated by the first-order Taylor expansion with respect to its parameters at initialization. The linearization of learning dynamics on neural networks has been helpful in showing the linear convergence of the training error on both overparametrized shallow [Li & Liang, 2018; Du et al., 2018] and deep neural networks [Allen-Zhu et al., 2018; Zou et al., 2018; 2020], as well as the characterizations of generalization error on both models [Arora et al., 2019a; Cao & Gu, 2019]. These findings clearly lead to the equivalence between learning dynamics of neural networks and the kernel methods in reproducing kernel Hilbert spaces (RKHS) associated with NTK. [1] Specifically, Arora et al. [2019a] provided the $\mathcal{O}(n^{-1/2})$ generalization bound of shallow neural network, where $n$ denotes the training sample size.

Recently, in the context of nonparametric regression, two papers, Nitanda & Suzuki [2020] and Hu et al. [2021], showed that neural network can obtain the convergence rate faster than $\mathcal{O}(n^{-1/2})$ by specifying the complexities of target function and hypothesis space. Specifically, Nitanda & Suzuki [2020] showed that the shallow neural network with smoothly approximated ReLU (swish, see Ramachandran et al. [2017]) activation trained via $\ell_2$-regularized averaged stochastic gradient descent (SGD) can recover the target function from RKHSs induced from NTK with swish activation. Similarly, Hu et al. [2021] showed that a shallow neural network with ReLU activation trained via $\ell_2$-regularized GD can generalize well, when the target function (i.e., $f_\rho^\star$) is from $\mathcal{H}_1^{\mathbf{NTK}}$.

---

[1]Henceforth, we denote $\mathcal{H}_1^{\mathbf{NTK}}$ and $\mathcal{H}_L^{\mathbf{NTK}}$ as RKHSs induced from NTK of shallow $L = 1$ and deep neural networks $L \geq 2$ with ReLU activations, respecitvely.

Notably, the rate that the papers Nitanda & Suzuki [2020] and Hu et al. [2021] obtained is minimax optimal, meaning that no estimators perform substantially better than the $\ell_2$-regularized GD or averaged SGD algorithms for recovering functions from respective function spaces. Nevertheless, these results are restricted to shallow neural networks, and cannot explain the generalization abilities of deep neural network (DNN). Similarly with Arora et al. [2019a], Cao & Gu [2019] obtained the $\mathcal{O}(n^{-1/2})$ generalization bound, showing that the SGD generalize well for $f_\rho^\star \in \mathcal{H}_L^{\mathbf{NTK}}$, when $f_\rho^\star$ has a bounded RKHS norm. However, the rate they obtained is slower than the minimax rate we can actually achieve. Furthermore, their results become vacuous under the presence of additive noises on the data set. Motivated from these observations, the fundamental question in this study is as follows:

*When the noisy dataset is generated from a function from $\mathcal{H}_L^{NTK}$, does the overparametrized DNN obtained via ($\ell_2$-regularized) GD provably generalize well the unseen data?*

We consider a neural network that has $L \geq 2$ hidden layers with width $m \gg n$. (i.e., over-parametrized deep neural network.) We focus on the least-squares loss and assume that the activation function is ReLU. A positivity assumption of NTK from ReLU DNN is imposed, meaning that $\lambda_\infty > 0$, where $\lambda_\infty$ denotes the minimum eigenvalue of the NTK. We give a more formal mathematical definition of ReLU DNN in the following Subsection 2.2. Under these settings, we provide an affirmative answer to the above question by investigating the behavior of $L_2$-prediction error of the obtained neural network with respect to GD iterations.

## 1.1 Contributions

Our derivations of algorithm-dependent prediction risk bound require the analysis on training dynamics of the estimated neural network through (regularized) GD algorithm. We include these results as the contributions of our paper, which can be of independent interests as well.

- In an unregulaized case, under the assumption $\lambda_\infty > 0$, we show that the training loss converges to $0$ at a linear rate. As will be detailed in subsection 3.3, this is the different result from the seminal work of Allen-Zhu et al. [2018], where they also prove a linear convergence of training loss of ReLU DNN, but under different data distribution assumption.
- We show that the DNN updated via vanilla GD does not recover the ground truth function $f_\rho^\star \in \mathcal{H}_L^{\mathbf{NTK}}$ under noisy observations, if the DNN is trained for either too short or too long: that is, the prediction error is bounded away from $0$ by some constant as $n$ goes to infinity.
- In regularized case, we prove the mean-squared error (MSE) of DNN is upper bounded by some positive constant. Additionally, we proved the dynamics of the estimated neural network get close to the solution of kernel ridge regression associated with NTK from ReLU DNN.
- We show that the $\ell_2$-regularization can be helpful in achieving the minimax optimal rate of the prediction risk for recovering $f_\rho^\star \in \mathcal{H}_L^{\mathbf{NTK}}$ under the noisy data. Specifically, it is shown that after some iterations of $\ell_2$-regularized GD, the minimax optimal rate (which is $\mathcal{O}\left(n^{-\frac{d}{2d-1}}\right)$, where $d$ is a feature dimension.) can be achieved.

Note that our paper is an extension of Hu et al. [2021] to DNN model, showing that the $\ell_2$-regularized DNN can achieve a minimax optimal rate of prediction error for recovering $f_\rho^\star \in \mathcal{H}_L^{\mathbf{NTK}}$. However, we would like to emphasize that our work is not a trivial application of their work from at least two technical aspects. These aspects are more detailed in the following subsection.

## 1.2 Technical Comparisons with Hu et al. [2021]

Firstly, in the analysis of training loss of regularized shallow neural-net, Hu et al. [2021] begin the proof by decomposing the difference between two individual predictions into two terms: one that is related with the gram matrix evaluated at each iteration of the algorithm and the perturbation term. Henceforth, we name this decompostion as "Gram+Pert" decomposition. This decomposition can be checked with the equality (E.2) in the supplementary PDF of Hu et al. [2021]. The key ingredients for the decomposition are (i) the simple gradient structure of the shallow neural net, and (ii) the partitioning of the nodes in the hidden-layer into two sets: a set of nodes whose activation

patterns change from their initializations during training, and the complement of the set. This construction of the sets peels off the ReLU activation in the difference so that the GD algorithm can be involved in the analysis. However, because of the compositional structure of the network, the same nodes partitioning technique cannot be applied for obtaining the decomposition in the DNN setting with ReLU activation. To avoid this difficulty, we employ a specially designed diagonal matrix $\widetilde{\mathbf{\Sigma}}$ and this matrix can peel off the ReLU function for each layer of the network. (See the definition of $\widetilde{\mathbf{\Sigma}}$ in the proof of Theorem 3.5 in the Appendix.) Recursive applications of this diagonal matrix across the entire hidden layers enable the Gram+Pert decomposition in our setting. It should be noted that the diagnoal matrix $\widetilde{\mathbf{\Sigma}}$ had been employed in Zou et al. [2020], which analyzed the behavior of training loss of classification problem via ReLU DNN under logistic loss. However, since their result is dependent on different data distribution assumption under the different loss function from ours, they didn't employ the Gram+Pert decomposition. Thus their technical approaches are different from ours.

Secondly, Hu et al. [2021] directly penalized the weight parameter $\mathbf{W}$ by adding $\|\mathbf{W}\|_F^2$ to the objective function. The $\ell_2$-regularization solely on the $\mathbf{W}$ has an effect of pushing the weight towards the origin. This makes $\|\mathbf{W}^{(k)} - \mathbf{W}^{(0)}\|_2 \leq \mathcal{O}(1)^2$, allowing most activation patterns of the nodes in the hidden layer can change during the training, even in overparametrized setting. Here, $\mathbf{W}^{(k)}$ denotes the updated weight parameter at $k$th itertaion of algorithm, and $\| \cdot \|_2$ denotes the spectral norm of the matrix. Nonetheless, this doesn't affect the analysis on obtaining the upper-bound of MSE in shallow neural net, since the network has only a single hidden layer. In contrast, in the DNN setting, we allow the non-convex interactions of parameters across the hidden layers. To the best of our knowledge, a technique for controlling the size of $\ell_2$-norm of network gradient has not been developed under this setting, yet. We circumvent this difficulty by regularizing the distance between the updated and the initialized parameter, instead by directly regularizing the updated parameter. This ensures that the updated parameter by $\ell_2$-regularized GD stays in a close neighborhood to its initialization, so that with heavy over-parametrization, the dynamics of network becomes linearized in parameter and we can ignore the non-convex interactions of parameters across the hidden layers. Specifically, under suitable model parameter setting, we prove that $\|\mathbf{W}_\ell^{(k)} - \mathbf{W}_\ell^{(0)}\|_2 \leq \widetilde{\mathcal{O}}_{\mathbb{P}}\left(\frac{1}{\sqrt{m}}\right)^3$ over all $\ell \in \{1, \dots, L\}$. Here, $\widetilde{\mathcal{O}}_{\mathbb{P}}(\cdot)$ hides the dependencies on the model parameters; $L$, $\omega$, and $n$. This result allows us to adopt the so-called "Forward Stability" argument developed by Allen-Zhu et al. [2018], and eventually leads to the control of network gradient under $\ell_2$ sense.

### 1.3 ADDITIONAL RELATED WORKS

There has been another line of work trying to characterize the generalizabilities of DNN under noisy observation settings. Specifically, it has been shown that the neural network model can achieve minimax style optimal convergence rates of $L_2$-prediction risk both in regression [Bauer & Kohler, 2019; Liu et al., 2019; Schmidt-Hieber, 2020] and classification [Kim et al., 2021] problems. Nonetheless, a limitation of the aforementioned papers is that they assume an adequate minimizer of the empirical risk can be obtained. In other words, the mathematical proofs of their theorems do not correspond to implementable algortihms.

Recently, several papers, which study the generalization properties of neural network with algorithmic guarantees, appear online. Specifically, Kohler & Krzyzak [2019] showed that the data interpolants obtained through DNN by vanilla GD is inconsistent. This result is consistent with our result, but they consider the overparametrized DNN that is a linear combination of $\Omega(n^{10d^2})$ smaller neural network, and the activation function they consider is sigmoid function, which is smooth and differentiable. Along this line of research, Kuzborskij & Szepesvári [2021] (regression) and Ji et al. [2021] (classification) showed that when training overparametrized shallow neural network, early stopping of vanilla GD enables us to obtain consistent estimators.

---

[2]This was empirically shown to be true in paper Wei et al. [2019]. See Figure 3 in their paper. We provide a brief mathematical explanation on why this result is hard to be shown in Appendix C.

[3]Readers can find the proof of this result in Appendix G.

**Notation.** We use the following notation for asymptotics: For sufficiently large $n$, we write $f(n) = \mathcal{O}(g(n))$, if there exists a constant $K > 0$ such that $f(n) \leq Kg(n)$, and $f(n) = \Omega(g(n))$ if $f(n) \geq K'g(n)$ for some constant $K' > 0$. The notation $f(n) = \Theta(g(n))$ means that $f(n) = \mathcal{O}(g(n))$ and $f(n) = \Omega(g(n))$. Let $\langle A, B \rangle_{\text{Tr}} := \text{Tr}(A^\top B)$ for the two matrices $A, B \in \mathbb{R}^{d_1 \times d_2}$. We adopt the shorthand notation denoting $[n] := \{1, 2, \dots, n\}$ for $n \in \mathbb{N}$.

## 2 PROBLEM FORMULATION

### 2.1 NON-PARAMETRIC REGRESSION

Let $\mathcal{X} \subset \mathbb{R}^d$ and $\mathcal{Y} \subset \mathbb{R}$ be the measureable feature space and output space. We denote $\rho$ as a joint probability measure on the product space $\mathcal{X} \times \mathcal{Y}$, and let $\rho_\mathcal{X}$ be the marginal distribution of the feature space $\mathcal{X}$. We assume that the noisy data-set $\mathcal{D} := \{(\mathbf{x}_i, \mathbf{y}_i)\}_{i=1}^n$ are generated from the non-parametric regression model $\mathbf{y}_i = f_\rho^\star(\mathbf{x}_i) + \varepsilon_i$, where $\varepsilon_i \overset{\text{i.i.d.}}{\sim} \mathcal{N}(0, 1^2)$ for $i = 1, \dots, n$. Let $\widehat{f}_{W(k)}(\cdot)$ be the value of neural network evaluated with the parameters $\mathbf{W}$ at the $k$-th iterations of GD update rule. At $k = 0$, we randomly initialize the weight parameters in the model following He initialization [He et al., 2015] with a slight modification. Then, the $L_2$ prediction risk is defined as the difference between two expected risks (i.e., excess risk) $\mathcal{R}(\widehat{f}_{W(k)}) := \mathbb{E}_{\rho \sim (\mathbf{x}, \mathbf{y})}\big[\big(\mathbf{y} - \widehat{f}_{W(k)}(\mathbf{x})\big)^2\big]$ and $\mathcal{R}(f_\rho^\star) := \mathbb{E}_{\rho \sim (\mathbf{x}, \mathbf{y})}[(\mathbf{y} - f_\rho^\star(\mathbf{x}))^2]$, where $f_\rho^\star(\mathbf{x}) := \mathbb{E}[\mathbf{y}|\mathbf{x}]$. Then, we can easily show the prediction risk has a following form:

$$\mathcal{R}(\widehat{f}_k, f_\rho^\star) := \mathcal{R}(\widehat{f}_{W(k)}) - \mathcal{R}(f^\star) = \mathbb{E}_{\rho_\mathbf{x}, \varepsilon}\left[\big(\widehat{f}_{W(k)}(\mathbf{x}) - f_\rho^\star(\mathbf{x})\big)^2\right]. \tag{1}$$

Note that the expectation is taken over the marginal probability measure of feature space, $\rho_\mathbf{x}$, and the noise of the data, $\varepsilon$. However, the (1) is still a random quantity due to the randomness of the initialized parameters $\big(\mathbf{W}_\ell^{(0)}\big)_{\ell=1,\dots,L}$.

### 2.2 DEEP NEURAL NETWORK WITH RELU ACTIVATION

Following the setting introduced in Allen-Zhu et al. [2018], we consider a fully-connected deep neural networks with $L$ hidden layers and $m$ network width. For $L \geq 2$, the output of the network $f_\mathbf{W}(\cdot) \in \mathbb{R}$ with input data $\mathbf{x} \in \mathcal{X}$ can be formally written as follows:

$$f_\mathbf{W}(\mathbf{x}) = \sqrt{m} \cdot \mathbf{v}^\mathsf{T} \sigma\big(\mathbf{W}_L \sigma\big(\mathbf{W}_{L-1} \cdots \sigma\big(\mathbf{W}_1 \mathbf{x}\big) \cdots\big)\big), \tag{2}$$

where $\mathcal{S}^{d-1}$ is a unit sphere in $d$-dimensional euclidean space, $\sigma(\cdot)$ is an entry-wise activation function, $\mathbf{W}_1 \in \mathbb{R}^{m \times d}$, $\mathbf{W}_2, \dots, \mathbf{W}_L \in \mathbb{R}^{m \times m}$ denote the weight matrices for hidden layers and $\mathbf{v} \in \mathbb{R}^{m \times 1}$ denote the weight vector for the output layer. Following the existing literature, we will consider ReLU activation function $\sigma(x) = \max(x, 0)$, which is the most commonly used activation function by practitioners.

***Random Initialization.*** Each entries of weight matrices in hidden layers are assumed to be generated from $\big(\mathbf{W}_{i,j}\big)_{\ell=1,\dots,L} \sim \mathcal{N}(0, \frac{2}{m})$, and entries of the output layer are drawn from $\mathbf{v}_j \sim \mathcal{N}(0, \frac{\omega}{m})$. This initialization scheme helps the forward propagation neither explode nor vanish at the initialization, seeing Allen-Zhu et al. [2018]; Zou et al. [2018; 2020]. Note that we initialize the parameters in the last layer with variance $\frac{\omega}{m}$, where $\omega \leq 1$ is a model parameter to be chosen later for technical convenience.

***Unregularized GD update rule.*** We solve a following $\ell_2$-loss function with the given dataset $\mathcal{D}$:

$$\mathcal{L}_\mathbf{S}(\mathbf{W}) = \frac{1}{2} \sum_{i=1}^n \big(y_i - f_\mathbf{W}(\mathbf{x}_i)\big)^2. \tag{3}$$

Let $\mathbf{W}_1^{(0)}, \dots, \mathbf{W}_L^{(0)}$ be the initialized weight matrices introduced above, and we consider a following gradient descent update rule:

$$\mathbf{W}_\ell^{(k)} = \mathbf{W}_\ell^{(k-1)} - \eta \nabla_{\mathbf{w}_\ell}\big(\mathcal{L}_\mathcal{S}(\mathbf{W}_\ell^{(k-1)})\big), \quad \ell \in [L], \quad k \geq 1, \tag{4}$$

where $\nabla_{\mathbf{W}_\ell}\big(\mathcal{L}_\mathcal{S}(\cdot)\big)$ is a partial gradient of the loss function $\mathcal{L}_\mathcal{S}(\cdot)$ with respect to the $\ell$-th layer parameters $\mathbf{W}_\ell$, and $\eta > 0$ is the learning rate of the gradient descent.

$\ell_2$-**regularized GD update rule.**   The estimator is obtained by minimizing a $\ell_2$-regularized function;

$$\mathbf{\Phi}_D(\mathbf{W}) := \mathcal{L}_\mathbf{S}\big(\mathbf{W}_D\big) + \frac{\mu}{2}\sum_{\ell=1}^{L}\left\|\mathbf{W}_{D,\ell} - \mathbf{W}_{D,\ell}^{(0)}\right\|_F^2. \tag{5}$$

Naturally, we update the model parameters $\big\{\mathbf{W}_{D,\ell}\big\}_{\ell=1,\dots,L}$ via modified GD update rule:

$$\mathbf{W}_{D,\ell}^{(k)} = \big(1 - \eta_2\mu\big)\mathbf{W}_{D,\ell}^{(k-1)} - \eta_1\nabla_{\mathbf{W}_\ell}\big[\mathcal{L}_\mathbf{S}\big(\mathbf{W}_D^{(k-1)}\big)\big] + \eta_2\mu\mathbf{W}_{D,\ell}^{(0)}, \quad \forall\ell \in [L], \quad \forall k \geq 1. \tag{6}$$

The notations $\eta_1$, $\eta_2$ are step sizes, and $\mu > 0$ is a tuning parameter on regularization. We adopt the different step sizes for the partial gradient and regularized term for the theoretical conveniences. Furthermore, we add the additional subscript $D$ to the update rule (6) to denote the variables are under the regularized GD update rule. Recall that the $\mathbf{W}_{D,\ell}^{(0)}$ are initialized parameters same with the unregularized case. For simplicity, we fix the output layer, and train $L$ hidden layers for both unregularized and regularized cases.

## 3   MAIN THEORY

First, we describe the neural tangent kernel (NTK) matrix of (2), which is first proposed by Jacot et al. [2018] and further studied by Arora et al. [2019b]; Du et al. [2019]; Lee et al. [2018]; Yang [2019]. NTK matrix of DNN is a $L$-times recursively defined $n \times n$ kernel matrix, whose entries are the infinite-width limit of the gram matrix. Let $\nabla_{\mathbf{W}_\ell}\big[f_{\mathbf{W}(0)}(\cdot)\big]$ be the gradient of the ReLU DNN (2) with respect to the weight matrix in the $\ell$th hidden layer at random initialization. Note that when $\ell = 1$, $\nabla_{\mathbf{W}_\ell}\big[f_{\mathbf{W}(0)}(\cdot)\big] \in \mathbb{R}^{m \times d}$ and when $\ell \in \{2,\dots,L\}$, $\nabla_{\mathbf{W}_\ell}\big[f_{\mathbf{W}(0)}(\cdot)\big] \in \mathbb{R}^{m \times m}$. Then, as $m \to \infty$,

$$\mathbf{H}(0) := \left(\frac{1}{m}\sum_{\ell=1}^{L}\big\langle\nabla_{\mathbf{W}_\ell}\big[f_{\mathbf{W}(0)}(\mathbf{x}_i)\big], \nabla_{\mathbf{W}_\ell}\big[f_{\mathbf{W}(0)}(\mathbf{x}_j)\big]\big\rangle_{\mathrm{Tr}}\right)_{n \times n} \to \mathbf{H}_L^\infty, \tag{7}$$

where $\mathbf{H}_L^\infty := \big\{\mathbf{Ker}(\mathbf{x}_i, \mathbf{x}_j)\big\}_{i,j=1}^{n}$. Here, $\mathbf{Ker}(\cdot,\cdot)$ denotes a NTK function of (2) to be defined as follows:

**Definition 3.1.** *(NTK function of (2)).* For any $\mathbf{x}, \mathbf{x}' \in \mathcal{X}$ and $\ell \in [L]$, define

$$\Phi^{(0)}(\mathbf{x}, \mathbf{x}') = \langle\mathbf{x}, \mathbf{x}'\rangle,$$

$$\Theta^{(\ell)}(\mathbf{x}, \mathbf{x}') = \begin{pmatrix} \Phi^{(\ell-1)}(\mathbf{x}, \mathbf{x}) & \Phi^{(\ell-1)}(\mathbf{x}, \mathbf{x}') \\ \Phi^{(\ell-1)}(\mathbf{x}', \mathbf{x}) & \Phi^{(\ell-1)}(\mathbf{x}', \mathbf{x}') \end{pmatrix} \in \mathbb{R}^{2 \times 2},$$

$$\Phi^{(\ell)}(\mathbf{x}, \mathbf{x}') = 2 \cdot \mathop{\mathbb{E}}_{(u,v)\sim\mathcal{N}(0,\Theta^{(\ell)})}\big[\sigma(u)\cdot\sigma(v)\big], \quad and$$

$$\dot{\Phi}^{(\ell)}(\mathbf{x}, \mathbf{x}') = 2 \cdot \mathop{\mathbb{E}}_{(u,v)\sim\mathcal{N}(0,\Theta^{(\ell)})}\big[\dot{\sigma}(u)\cdot\dot{\sigma}(v)\big],$$

*where $\dot{\sigma}(u) = \mathbb{1}\big(u \geq 0\big)$. Then, we can derive the final expression of NTK function of (2) as follows:*

$$\mathbf{Ker}(\mathbf{x}, \mathbf{x}') = \frac{\omega}{2}\cdot\sum_{\ell=1}^{L}\left(\Phi^{(\ell-1)}(\mathbf{x}, \mathbf{x}')\cdot\prod_{\ell'=\ell}^{L}\dot{\Phi}^{(\ell')}(\mathbf{x}, \mathbf{x}')\right). \tag{8}$$

The expression in (8) is adapted from Cao & Gu [2019]. As remarked in Cao & Gu [2019], a coefficient 2 in $\Phi^{(\ell)}$ and $\dot{\Phi}^{(\ell)}$ remove the exponential dependence on the network depth $L$ in the NTK function. However, when compared with the NTK formula in Cao & Gu [2019], (8) is different from two aspects: (i) An additional factor $\omega$ in (8)) comes from the difference in initialization settings of the output layer, in which Cao & Gu [2019] considers $v_j \sim \mathcal{N}(0, \frac{1}{m})$, whereas we consider $v_j \sim \mathcal{N}(0, \frac{\omega}{m})$. (ii) $\Phi^{(L)}$ is not added in the final expression of (8)), whereas it is added in the

definition provided in Cao & Gu [2019]. This is because we only train the $L$ hidden layers but fix the output layer, while Cao & Gu [2019] train the entire layers of the network including the output layer.

As already been pointed by several papers, Cho & Saul [2009] and Jacot et al. [2018], it can be proved that the NTK function (8) is a positive semi-definite kernel function. Furthermore, Cho & Saul [2009] prove that the expectations in $\Phi$ and $\dot{\Phi}$ have closed form solutions, when the covariance matrices have the form $\left( \begin{smallmatrix} 1 & t \\ t & 1 \end{smallmatrix} \right)$ with $|t| \leq 1$:

$$
\underset{(u,v) \sim \mathcal{N}(0,\Theta^{(\ell)})}{\mathbb{E}} \big[ \sigma(u) \cdot \sigma(v) \big] = \frac{1}{2\pi} \left( t \cdot (\pi - \arccos(t)) + \sqrt{1 - t^2} \right),
$$
$$
\underset{(u,v) \sim \mathcal{N}(0,\Theta^{(\ell)})}{\mathbb{E}} \big[ \dot{\sigma}(u) \cdot \dot{\sigma}(v) \big] = \frac{1}{2\pi} \big( \pi - \arccos(t) \big).
\tag{9}
$$

Clearly, (8) is symmetric and continuous on the product space $\mathcal{X} \times \mathcal{X}$, from which it can be implied that $\mathbf{Ker}(\cdot, \cdot)$ is a Mercer kernel inducing an unique RKHS. Following Ghorbani et al. [2020], we define the RKHS induced by (8) as:

**Definition 3.2.** *(NTK induced RKHS). For some integer $p \in \mathbb{N}$, set of points $\{\tilde{\mathbf{x}}_j\}_{j=1}^p \subset \mathcal{X}$, and weight vector $\alpha := \{\alpha_1, \ldots, \alpha_p\} \in \mathbb{R}^p$, define a complete vector space of functions, $f : \mathcal{X} \to \mathbb{R}$,*

$$
\mathcal{H}_L^{\textit{NTK}} := cl\bigg( \bigg\{ f(\cdot) = \sum_{j=1}^p \alpha_j \mathbf{Ker}(\cdot, \tilde{\mathbf{x}}_j) \bigg\} \bigg),
\tag{10}
$$

*where $cl(\cdot)$ denotes closure.*

In the remaining of our work, we assume the regression function $f_\rho^\star(\mathbf{x}) := \mathbb{E}[\mathbf{y}|\mathbf{x}]$ belongs to $\mathcal{H}_L^{\mathbf{NTK}}$.

### 3.1 ASSUMPTIONS.

In this subsection, we state the assumptions imposed on the data distribution with some remarks.

**(A1)** $\rho_{\mathcal{X}}$ is an uniform distribution on $\mathcal{S}^{d-1} := \{\mathbf{x} \in \mathbb{R}^d \mid \|\mathbf{x}\|_2 = 1\}$, and noisy observations are assumed to be bounded. (i.e., $\rho_{\mathbf{x}} \sim \mathbf{Unif}(\mathcal{S}^{d-1})$, $\mathbf{y}_i = \mathcal{O}(1), \forall i \in [n]$.)

**(A2)** Draw $n$ i.i.d. samples $\{\mathbf{x}_i, f_\rho^\star(\mathbf{x}_i)\}_{i=1}^n$ from the joint measure $\rho$. Then, with probability at least $1 - \delta$, we have $\lambda_{\min}(\mathbf{H}_L^\infty) = \lambda_\infty > 0$.

**Remark 3.3.**

- When the feature space is restricted on the unit sphere, the NTK function in (8) becomes rotationally invariant zonal kernel. This setting allows to adopt the results of spectral decay of (8) in the basis of spherical harmonic polynomials for measuring the complexity of hypothesis space, $\mathcal{H}_L^{\mathbf{NTK}}$. See the subsection 3.2 and references therein.

- Assumption (A2) is commonly employed in NTK related literature for proving global convergence of training error and generalization error of both deep and shallow neural network, Du et al. [2018; 2019]; Arora et al. [2019a]. Note that the (A2) holds as long as no two $\mathbf{x}_i$ and $\mathbf{x}_j$ are parallel to each other, which is true for most of the real-world distributions. See the proof of this claim in Du et al. [2019].

### 3.2 MINIMAX RATE FOR RECOVERING $f_\rho^\star \in \mathcal{H}_L^{\mathbf{NTK}}$

The obtainable minimax rate of $L_2$-prediction error is directly related with the complexity of function space of interest. In our setting, the complexity of RKHS $\mathcal{H}_L^{\mathbf{NTK}}$ can be characterized by the eigen-decay rate of the NTK function. Since $\mathbf{Ker}(\mathbf{x}, \mathbf{x}')$ is defined on the sphere, the decomposition can be given in the basis of spherical harmonics as follows:

$$
\mathbf{Ker}(\mathbf{x}, \mathbf{x}') = \sum_{k=0}^\infty \mu_k \sum_{j=1}^{N(d,k)} Y_{k,j}(\mathbf{x}) Y_{k,j}(\mathbf{x}'),
$$

where $Y_{k,j}, j = 1, \ldots, N(d,k)$ are spherical harmonic polynomials of degree $k$ and $\{\mu_k\}_{k=0}^{\infty}$ are non-negative eigenvalues. Recently, several researchers, both empirically [Basri et al., 2020] and theoretically [Chen & Xu, 2020; Geifman et al., 2020; Bietti & Bach, 2021], showed that, for large enough harmonic function frequency $k$, the decay rate of the eigenvalues $\mu_k$ is in the order of $\Theta\left(k^{-d}\right)$ [4]. Given this result and the fact $N(d,k) = \frac{2k+d-3}{k}\binom{k+d-3}{d-2}$ grows as $k^{d-2}$ for large $k$, it can be easily shown $\lambda_j = \Theta\left(j^{-\frac{d}{d-1}}\right)$, when $\mathbf{Ker}(\mathbf{x},\mathbf{x}') = \sum_{j=1}^{\infty} \lambda_j \phi_j(\mathbf{x})\phi_j(\mathbf{x}')$, for eigen-values $\lambda_1 \geq \lambda_2 \geq \cdots \geq 0$ and orthonormal basis $\{\phi_j\}_{j=1}^{\infty}$. Furthermore, it is a well known fact that if the eigenvalues decay at the rate $\lambda_j = \Theta(j^{-2\nu})$, then the corresponding minimax rate for estimating function in RKHS is $\mathcal{O}\left(n^{-\frac{2\nu}{2\nu+1}}\right)$, [Raskutti et al., 2014; Yuan & Zhou, 2016; Hu et al., 2021]. By setting $2\nu = \frac{d}{d-1}$, we can see the minimax rate for recovering $f_\rho^\star \in \mathcal{H}_L^{\mathbf{NTK}}$ is $\mathcal{O}\left(n^{-\frac{d}{2d-1}}\right)$.

**Remark 3.4.** *We defer all the technical proofs of the Theorems in subsections $3.3$ and $3.4$ in the Appendix for conciseness of the paper. We also provide numerical experiments which can corroborate our theoretical findings in the Appendix A.*

## 3.3 ANALYSIS OF UNREGULARIZED DNN

In this subsection, we provide the results on the training loss of DNN estimator obtained via minimizing unregularized $\ell_2$-loss (3) and on the corresponding estimator's $L_2$-prediction risk $\mathcal{R}\left(\widehat{f}_k, f_\rho^\star\right)$.

**Theorem 3.5.** *(Optimization) For some $\delta \in [0,1]$, set the width of the network as $\frac{m}{\log^3(m)} \geq \Omega\left(\frac{\omega^7 n^8 L^{18}}{\lambda_\infty^8 \delta^2}\right)$, and set the step-size of gradient descent as $\eta = \mathcal{O}\left(\frac{\lambda_\infty}{n^2 L^2 m}\right)$. Then, with probability at least $1 - \delta$ over the randomness of initialized parameters $\mathcal{W}^{(0)} := \left\{\mathbf{W}_\ell^{(0)}\right\}_{\ell=1}^{L+1}$ with $\mathbf{W}_{L+1}^{(0)} = \mathbf{v}$, we have for $k = 0, 1, 2, \ldots$,*

$$\mathcal{L}_\mathbf{S}\left(\mathcal{W}^{(k)}\right) \leq \left(1 - \frac{\eta m \lambda_\infty}{2}\right)^k \mathcal{L}_\mathbf{S}\left(\mathcal{W}^{(0)}\right). \tag{11}$$

*In other words, the training loss drops to $0$ at a linear rate.*

We acknowledge a series of past works Allen-Zhu et al. [2018]; Du et al. [2019] have similar spirits with those in Theorem 3.5. However, it is worth noting that their results are not applicable in our problem settings and data assumptions. Specifically, the result of Du et al. [2019] is based on the smooth and differentiable activation function, whereas the Theorem 3.5 is about the training error of ReLU activation function, which is not differentiable at 0. Furthermore, the result of Allen-Zhu et al. [2018] relies on $\phi$-separateness assumption stating that the every pair of feature vectors $\left\{\mathbf{x}_i, \mathbf{x}_j\right\}_{i \neq j}^n$ is apart from each other by some constant $\phi > 0$ in a Euclidean norm. In our work, the positivity assumption on the minimum eigenvalue of the NTK is imposed (i.e., $\lambda_\infty > 0$).

**Remark 3.6.** *Reducing the order of network width is definitely another line of interesting research direction. We are aware of some works in literature, but we chose not to adopt the techniques since this can make the analysis overly complicated. To the best of our knowledge, the paper that most neatly summarizes this line of literature is Zou & Gu [2019]. See the table in page $3$ in their paper. The order of width they obtained is $\Omega\left(\frac{n^8 L^{12}}{\phi^8}\right)$, where they impose $\phi$-separateness assumption.*

**Remark 3.7.** *There has been an attempt to make a connection between the positivity and $\phi$-separateness assumptions. Recently, Zou & Gu [2019] proved the relation $\lambda_\infty = \Omega\left(\phi n^{-2}\right)$ [5] in a shallow-neural net setting. See Proposition $3.6$. of their work. However, it is still an open question on whether this relation holds in DNN setting as well. The results in Theorem 3.5 suggest a positive conjecture on this question. Indeed, plugging the relation $\lambda_\infty = \Omega\left(\phi n^{-2}\right)$ in (11) and in the $\eta = \mathcal{O}\left(\frac{\lambda_\infty}{n^2 L^2 m}\right)$ yield the discount factor $\left(1 - \Omega\left(\frac{\eta m \phi}{n^2}\right)\right)^k$ and step-size $\eta = \mathcal{O}\left(\frac{\phi}{n^4 L^2 m}\right)$, which*

---

[4] In shallow neural network with ReLU activation without bias terms, it is shown that $\mu_k$ satisfy $\mu_0, \mu_1 > 0$, $\mu_k = 0$ if $k = 2j + 1$ with $j \geq 1$, and otherwise $\mu_k = \Theta\left(k^{-d}\right)$. See Bietti & Mairal [2019]. However, in ReLU DNN, it is shown that these parity constraints can be removed even without bias terms and $\mu_k$ achieves $\Theta\left(k^{-d}\right)$ decay rate for large enough $k$. Readers can refer Bietti & Bach [2021] for this result.

[5] We conjecture that this is not the tightest lower bound on $\lambda_\infty$. Recently, Bartlett et al. [2021] proves that $\lambda_\infty \gtrsim d/n$ in shallow neural net setting. See Lemma 5.3 in their paper.

*are exactly the same orders as presented in Allen-Zhu et al. [2018]. See Theorem 1 of their ArXiv version paper for the clear comparison. We leave the proof of this conjecture as a future work.*

**Theorem 3.8.** *(**Generalization**) Let $f_\rho^\star \in \mathcal{H}_L^{NTK}$. Fix a failure probability $\delta \in [0, 1]$. Set the width of the network as $\frac{m}{\log^3(m)} \geq \Omega\left(\frac{\omega^7 n^8 L^{18}}{\lambda_\infty^8 \delta^2}\right)$, the step-size of gradient descent as $\eta = \mathcal{O}\left(\frac{\lambda_\infty}{n^2 L^2 m}\right)$, and the variance parameter $\omega \leq \mathcal{O}\left(\left(\frac{\lambda_\infty \delta}{n}\right)^{2/3}\right)$. Then, if the GD iteration $k \geq \Omega\left(\frac{\log(n)}{\eta m \lambda_\infty}\right)$ or $k \leq \mathcal{O}\left(\frac{1}{\eta m \omega L}\right)$, with probability at least $1 - \delta$ over the randomness of initialized parameters $\mathcal{W}^{(0)}$, we have*

$$\mathcal{R}\left(\widehat{f}_k, f_\rho^\star\right) = \Omega(1).$$

This theorem states that if the network is trained for too long or too short, the $L_2$-prediction error of $\widehat{f}_{\mathbf{W}^{(k)}}$ is bounded away from 0 by some constant factor. Specifically, the former scenario indicates that the overfitting can be harmful for recovering $f_\rho^\star \in \mathcal{H}_L^{\mathbf{NTK}}$ given the noisy observations.

**Remark 3.9.** *Readers should note that the Theorem 3.8 does not consider if the GD algorithm can achieve low prediction risk $\mathcal{R}\left(\widehat{f}_k, f_\rho^\star\right)$ over the range of iterations $(\eta m \omega L)^{-1} \lesssim k \lesssim (\eta m \lambda_\infty)^{-1} \log(n)$. In the numerical experiment to be followed in Appendix A, we observe that for some algorithm iterations $k^*$, the risk indeed decreases to the same minimum as low as the $\ell_2$-regularized algorithm can achieve, and increases again. This observation implies that the unregularized algorithm can achieve the minimax rate of prediction risk. However, analytically deriving a data-dependent stopping time $k^*$ in our scenario requires further studies, since we need a sharp characterization of eigen-distribution of NTK matrix of ReLU DNN, denoted as $\mathbf{H}_L^\infty$ in this paper. Readers can refer the Theorem 4.2. of Hu et al. [2021] in shallow-neural network and equation (6) in Raskutti et al. [2014] in kernel regression context on how to compute $k^\star$ with the given eigenvalues of the associated kernel matrices.*

## 3.4 ANALYSIS OF $\ell_2$-REGULARIZED DNN

In this subsection, we study the training dynamics of $\ell_2$-regularized DNN and the effects of the regularization for obtaining the minimax optimal convergence rate of $L_2$-prediction risk. In the results to be followed, we set the orders of model parameters $\mu, \eta_1, \eta_2$ in (6), and a variance parameter of output layer, $\omega$ as follows:

$$\mu = \Theta\left(n^{\frac{d-1}{2d-1}}\right), \quad \eta_1 = \Theta\left(\frac{1}{m} n^{-\frac{3d-2}{2d-1}}\right), \quad \eta_2 = \Theta\left(\frac{1}{L} n^{-\frac{3d-2}{2d-1}}\right), \quad \omega = \mathcal{O}\left(\frac{1}{L^{3/2}} n^{-\frac{5d-2}{2d-1}}\right). \tag{12}$$

**Theorem 3.10.** *(**Optimization**) Suppose we minimize $\ell_2$-regularized objective function (5) via modified GD (6). Set the network width $\frac{m}{\log^3(m)} \geq \Omega\left(\frac{L^{20} n^{24}}{\delta^2}\right)$ and model parameters as in (12). Then, with probability at least $1 - \delta$, the mean-squared error follows*

$$\mathcal{L}_{\mathbf{S}}\left(\mathcal{W}_D^{(k)}\right)/n \leq \left(1 - \eta_2 \mu L\right)^k \cdot \mathcal{L}_{\mathbf{S}}\left(\mathcal{W}_D^{(0)}\right)/n + \mathcal{O}_{\mathbb{P}}(1), \tag{13}$$

*for $k \geq 0$. Additionally, after $k \geq \Omega\left((\eta_2 \mu L)^{-1} \log(n^{3/2})\right)$ iterations of (6), for some constant $C > 0$, we have*

$$\left\| \mathbf{u}_D(k) - \mathbf{H}_L^\infty \left(C\mu \cdot \mathcal{I} + \mathbf{H}_L^\infty\right)^{-1} \mathbf{y} \right\|_2 \leq \mathcal{O}_{\mathbb{P}}\left(\frac{1}{n}\right), \tag{14}$$

*where we denote $\mathbf{u}_D(k) := [\widehat{f}_{\mathbf{W}_D^{(k)}}(\mathbf{x}_1), \ldots, \widehat{f}_{\mathbf{W}_D^{(k)}}(\mathbf{x}_n)]^\top$.*

Several comments are in sequel. Theorem 3.10 is, to our knowledge, the first result that rigorously shows the training dynamics of $\ell_2$-regularized ReLU DNN in overparametrized setting. Observe that the first term on the right-hand side of the inequality (13) converges linearly to 0, and the second term is some positive constant that is bounded away from 0. This implies that the MSE of regularized DNN is upper-bounded by some positive constant. Note that we only provide the upper

bound, but the results of our numerical experiments indicate that the MSE is lower-bounded by $\mathcal{O}_{\mathbb{P}}(1)$ as well. We leave the proof of this conjecture for the future work.

The inequality (14) states that the trained dynamics of the regularized neural network can approximate the optimal solution (denoted as $g_\mu^\star$) of the following kernel ridge regression problem:

$$\min_{f \in \mathcal{H}^{\mathbf{NTK}}} \left\{ \frac{1}{2} \sum_{i=1}^n \left( y_i - f(\mathbf{x}_i) \right)^2 + \frac{C\mu}{2} \|f\|_{\mathcal{H}_L^{\mathbf{NTK}}}^2 \right\}, \tag{15}$$

where $\| \cdot \|_{\mathcal{H}_L^{\mathbf{NTK}}}$ denotes a NTK-induced RKHS norm. Note that the optimization problem in (15) is not normalized by sample size $n$. The inequality (14) states that after approximately $(\eta_2 \mu L)^{-1}$ iterations of (6), the error rate becomes $\mathcal{O}_{\mathbb{P}}\left(\frac{1}{n}\right)$. The approximation error is computed at the training data points under $\ell_2$ norm. This should be compared with the Theorem 5.1 of Hu et al. [2021], where they showed that the similar approximation holds "within" a certain range of algorithm in shallow neural network setting. In contrast, we show that the approximation holds "after" $k \geq \Omega\left((\eta_2 \mu L)^{-1} \log(n^{3/2})\right)$ in deep neural network. It should be noted that the difference of results comes from the regularization scheme, where we penalize the $\sum_{\ell=1}^L \|\mathbf{W}_\ell - \mathbf{W}_\ell^{(0)}\|_F^2$, whereas Hu et al. [2021] regularized the term $\|\mathbf{W}_1\|_F^2$.

As another important comparison, Hu et al. [2019] showed the equivalence of a solution of kernel ridge regression associated with NTK and the first order Taylor expansion of the regularized neural network dynamics; note, however, that the $\mathbf{u}_D(k)$ in (14) is a full neural network dynamics. Let $\mathcal{R}(\widehat{f}_{\mathbf{W}_D^{(k)}}, f_\rho^\star)$ be the $L_2$-prediction risk of the regularized estimator $\widehat{f}_{\mathbf{W}_D^{(k)}}$ via modified GD (6). Next theorem states the result of generalization ability of $\widehat{f}_{\mathbf{W}_D^{(k)}}$.

**Theorem 3.11.** *(Generalization) Let $f_\rho^\star \in \mathcal{H}_L^{NTK}$. Suppose the network width $\frac{m}{\log^3(m)} \geq \Omega\left(\frac{L^{20} n^{24}}{\delta^2}\right)$ and model parameters are set as suggested in (12). Then, with probability tending to 1, we have*

$$\mathcal{R}\left(\widehat{f}_{\mathbf{W}_D^{(k)}}, f_\rho^\star\right) = \mathcal{O}_{\mathbb{P}}\left(n^{-\frac{d}{2d-1}}\right).$$

The resulting convergence rate is $\mathcal{O}\left(n^{-\frac{d}{2d-1}}\right)$ with respect to the training sample size $n$. Note that the rate is always faster than $\mathcal{O}\left(n^{-1/2}\right)$ and turns out to be the minimax optimal [Caponnetto & De Vito, 2007; Blanchard & Mücke, 2018] for recovering $f_\rho^\star \in \mathcal{H}_L^{\mathbf{NTK}}$ in the following sense:

$$\lim_{r \to 0} \liminf_{n \to \infty} \inf_{\widehat{f}} \sup_\rho \mathbb{P}\left[\mathcal{R}\left(\widehat{f}, f_\rho^\star\right) > r n^{-\frac{d}{2d-1}}\right] = 1, \tag{16}$$

where $\rho$ is a data distribution class satisfying the Assumptions (A1), (A2) and $f_\rho^\star \in \mathcal{H}_L^{\mathbf{NTK}}$, and infimum is taken over all estimators $\mathcal{D} \to \widehat{f}$. It is worth noting that the minimax rate in (16) is same with the minimax rate for recovering $f_\rho^\star \in \mathcal{H}_1^{\mathbf{NTK}}$. (i.e., Hu et al. [2021]) This result can be derived from the recent discovery of the equivalence between two function spaces , $\mathcal{H}_1^{\mathbf{NTK}} = \mathcal{H}_L^{\mathbf{NTK}}$. See Geifman et al. [2020] and Chen & Xu [2020].

**Remark 3.12.** *A particular choice of $\mu = \Theta\left(n^{\frac{d-1}{2d-1}}\right)$ in (12) is for obtaining an optimal minimax rate for prediction error in Theorem 3.11. Specifically, the order of $\mu$ determines the $L_2$ distance between the $f_\rho^\star$ and the kernel regressor $g_\mu^\star$. That is, $\|f_\rho^\star - g_\mu^\star\|_2^2 = \mathcal{O}_{\mathbb{P}}(\frac{\mu}{n})$. With the result $\mathcal{H}_1^{NTK} = \mathcal{H}_L^{NTK}$, the same proof of Lemma D.2. in Hu et al. [2021] can be applied for proving this result.*

## 4 CONCLUSION

We analyze the convergence rate of $L_2$-prediction error of both the unregularized and the regularized gradient descent for overparameterized DNN with ReLU activation for a regression problem. Under a positivity assumption of NTK, we show that without the adoption of early stopping, the $L_2$-prediction error of the estimated DNN via vanilla GD is bounded away from 0 (Theorem 3.5), whereas the prediction error of the DNN via $\ell_2$-regularized GD achieves the optimal minimax rate (Theorem 3.11). The minimax rate $\mathcal{O}\left(n^{-\frac{d}{2d-1}}\right)$ is faster than the $\mathcal{O}(n^{-1/2})$ by specifying the complexities of target function and hypothesis space.

ACKNOWLEDGMENTS

This project is partially supported by the Transdisciplinary Research Institute for Advancing Data Science (TRIAD), http://triad.gatech.edu, which is a part of the TRIPODS program at NSF and locates at Georgia Tech, enabled by the NSF grant CCF-1740776. Authors are also partially supported by NSF grant DMS-2015363.

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

## A  Numerical illustrations

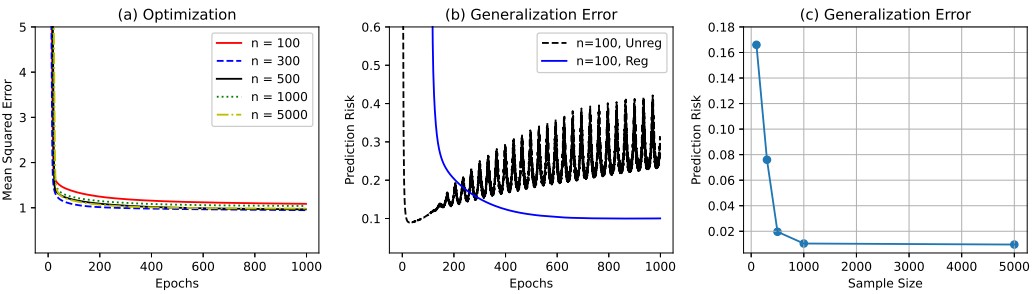

Figure 1: Results on synthetic data.

In this section, we use synthetic data to corroborate our theoretical findings. We use the He initialization [He et al., 2015] and employ ($\ell_2$-regularized) GD as introduced in subsection 2.2. For the experiments, we run 1000 epochs of GD and use a fixed step size, setting $\eta_1 = \eta_2 = 0.001$. We uniformly generate $n$ feature data $\mathbf{x_i}^{\text{train}}$ from $\mathcal{S}^{d-1}$ with $d = 2$ and generate $\mathbf{y}_i$ from $f_\rho^\star(\mathbf{x}_i^{\text{train}})$ with $\varepsilon_i \sim \mathcal{N}(0, 1)$. To create a function $f_\rho^\star \in \mathcal{H}_L^{\mathbf{NTK}}$, we use the definition in (10) with $\alpha \in \mathbf{Unif}(\mathcal{S}^{p-1})$ and with $p$ fixed points $\{\tilde{\mathbf{x}}_j\}_{j=1}^p \subset \mathbf{Unif}(\mathcal{S}^{d-1})$, where $p$ is simply set as 1. Note that $\mathbf{Ker}(\cdot, \cdot)$ in (10) can be calculated via the formulas (8) and (9) with specified network depth $L$. We consider a scenario where we have a network with depth $L = 8$ and width $m = 2000$. The variance parameter of the output layer ($\omega$) is set as 1 for unregularized and 0.001 for regularized cases.

In Fig 1.(a), we record the training errors of regularized networks over the GD epochs $k \leq 1000$, where we have $n \in \{100, 300, 500, 1000, 5000\}$ training samples. This aims to verify the inequality (13) that the MSE of regularized network is bounded away from 0 by some constant. In Fig 1.(b), the prediction risks of both unregularized and regularized networks are displayed. We approximate the risk with $\frac{1}{500} \sum_{j=1}^{500} \left(\widehat{f}_k(\mathbf{x}_j^{\text{test}}) - f_\rho^\star(\mathbf{x}_j^{\text{test}})\right)^2$ with a new test data set $\{\mathbf{x}_j^{\text{test}}, f_\rho^\star(\mathbf{x}_j^{\text{test}})\}_{j=1}^{500}$ over $k \leq 1000$ for both unregularized and regularized cases. In both cases, they reach the same minimal risks, but the risk of unregularized network increase after it hits the minimal point, whereas the risk of regularized network stays stable. Theorem 3.6 tells us that for the iteration less than the order $\mathcal{O}\left(\frac{1}{\eta m \omega L}\right)$, the prediction error is bounded away from 0. In the experiment for unregularized case, we set $\eta = 0.01$, $m = 2000$, $L = 8$, and $\omega = 1$. Plugging in these parameters in the bound says that the minimum can be achieved within a very few iterations. Note that the optimal risk is non-zero as long as we have finite sample sizes $n$, but converges to 0 at the rate $\mathcal{O}\left(n^{-\frac{d}{2d-1}}\right)$. In Fig 1.(c), we verify that the more training sample sizes we have, the closer the risks of the regularized networks get to 0. The risk is evaluated at the sample sizes $n = \{100, 300, 500, 1000, 5000\}$.

We have to acknowledge that there is a discrepancy between our experiment setting and theory. Specifically, due to the limited computing power, we could not run the experiment under the regime of width $\frac{m}{\log^3(m)} \geq \Omega\left(\frac{\omega^7 n^8 L^{18}}{\lambda_\infty^8 \delta^2}\right)$. But the prediction risk behaves similarly as expected by our theorems, which can be a partial evidence that the statement in theorems still holds in the narrower width of the network.

## B  Preliminary Notations

Before presenting the formal proofs of Lemmas and main results, we introduce several notations used frequently throughout the proofs. First, we denote $\mathbf{x}_{\ell,i}$ the output of the $\ell$th hidden layer with the input data $\mathbf{x}_i$ after applying entry-wise ReLU activation function.

$$\mathbf{x}_{\ell,i} = \sigma\big(\mathbf{W}_\ell \sigma\big(\mathbf{W}_{\ell-1} \cdots \sigma\big(\mathbf{W}_1 \mathbf{x}_i\big) \cdots\big)\big).$$

Denote $f_{\mathbf{W}(k)}(x)$ a value of neural network (2) evaluated at the collection of network parameters $\mathcal{W}^{(k)} := \big\{\mathbf{W}_\ell^{(k)}\big\}_{\ell=1,\ldots,L}$ and $\mathbf{W}_\ell^{(k)}$ denotes the $\ell$th hidden layer parameter updated by $k$th GD

iterations.

***Partial gradient of*** $f_{\mathbf{W}(k)}(x)$***.*** We employ the following matrix product notation which was used in several other papers [Zou et al., 2018; Cao & Gu, 2019]:

$$\prod_{r=\ell_1}^{\ell_2} A_r := \begin{cases} A_{\ell_2} A_{\ell_2-1} \cdots A_{\ell_1} & \text{if} \quad \ell_1 \leq \ell_2, \\ \mathcal{I} & \text{otherwise.} \end{cases} \tag{17}$$

Then, the partial gradient of $f_{\mathbf{W}(k)}(x)$ with respect to $\mathbf{W}_\ell^{(k)}$ for $1 \leq \ell \leq L$ has a following form: for $i \in \{1, \ldots, n\}$,

$$\nabla_{\mathbf{W}_\ell}\big[f_{\mathbf{W}(k)}(\mathbf{x}_i)\big] = \sqrt{m} \cdot \left[ \mathbf{x}_{\ell-1,i}^{(k)} \mathbf{v}^{\mathsf{T}} \left( \prod_{r=\ell+1}^{L} \boldsymbol{\Sigma}_{r,i}^{(k)} \mathbf{W}_r^{(k)} \right) \boldsymbol{\Sigma}_{\ell,i}^{(k)} \right]^{\top}, \qquad \ell \in [L],$$

where $\boldsymbol{\Sigma}_{\ell,i}^{(k)} := \mathrm{Diag}\big(\mathbb{1}(\langle \mathbf{w}_{\ell,1}^{(k)}, \mathbf{x}_{\ell-1,i}^{(k)} \rangle \geq 0), \ldots, \mathbb{1}(\langle \mathbf{w}_{\ell,m}^{(k)}, \mathbf{x}_{\ell-1,i}^{(k)} \rangle \geq 0)\big) \in \mathbb{R}^{m \times m}$ and $\mathbf{w}_{\ell,j}^{(k)}$ denotes $j$th column of the matrix $\mathbf{W}_\ell^{(k)}$.

***Gram matrix*** $\mathbf{H}(k)$***.*** Each entries of empirical gram matrix evaluated at the $k$th GD update are defined as follows:

$$\mathbf{H}_{i,j}(k) = \frac{1}{m} \sum_{\ell=1}^{L} \big\langle \nabla_{\mathbf{W}_\ell}\big[f_{\mathbf{W}(k)}(\mathbf{x}_i)\big], \nabla_{\mathbf{W}_\ell}\big[f_{\mathbf{W}(k)}(\mathbf{x}_j)\big] \big\rangle_{\mathrm{Tr}}.$$

Note that $\mathbf{H}(0) \to \mathbf{H}_L^\infty$ as $m \to \infty$ which is proved in Jacot et al. [2018]; Yang [2019]; Lee et al. [2018]; Arora et al. [2019b].

***Perturbation region of weight matrices.*** Consider a collection of weight matrices $\widetilde{\mathbf{W}} = \{\widetilde{\mathbf{W}}_\ell\}_{\ell=1,\ldots,L}$ such that

$$\widetilde{\mathbf{W}} \in \mathcal{B}(\mathcal{W}^{(0)}, \tau) := \left\{ \widetilde{\mathbf{W}}_\ell : \|\widetilde{\mathbf{W}}_\ell - \mathbf{W}_\ell^{(0)}\|_2 \leq \tau, \quad \forall \ell \in [L] \right\}. \tag{18}$$

For all $i \in \{1, \ldots, n\}$ and $\ell = 1, \ldots, L$, we denote $\mathbf{x}_{\ell,i}$ and $\widetilde{\mathbf{x}}_{\ell,i}$ as the outputs of the $\ell$-th layer of the neural network with weight matrices $\mathbf{W}^{(0)}$ and $\widetilde{\mathbf{W}}$, and $\Sigma_{\ell,i}$ and $\widetilde{\Sigma}_{\ell,i}$ are diagonal matrices with $(\Sigma_{\ell,i})_{jj} = \mathbb{1}(\langle \mathbf{w}_{\ell,j}^{(0)}, \mathbf{x}_{\ell-1,i} \rangle \geq 0)$ and $(\widetilde{\Sigma}_{\ell,i})_{jj} = \mathbb{1}(\langle \widetilde{\mathbf{w}}_{\ell,j}, \widetilde{\mathbf{x}}_{\ell-1,i} \rangle \geq 0)$, respectively.

## C  WHY IS IT HARD TO PROVE $\|\mathbf{W}_{D,\ell}^{(k)} - \mathbf{W}_{D,\ell}^{(0)}\|_2 \leq \mathcal{O}(1)$?

In this subsection, we provide a heuristic argument on why it is hard to prove $\|\mathbf{W}_{D,\ell}^{(k)} - \mathbf{W}_{D,\ell}^{(0)}\|_2 \leq \mathcal{O}(1)$, where $\mathbf{W}_{D,\ell}^{(k)}$ is the model parameter of $\ell$th layer in $k$th iteration of algorithm. Here, we regularize solely on the model parameter, instead on the relative to the initialization. In this case, we can write the update rule as follows :

$$\mathbf{W}_{D,\ell}^{(k)} = \big(1 - \eta_2 \mu\big) \mathbf{W}_{D,\ell}^{(k-1)} - \eta_1 \nabla_{\mathbf{W}_\ell}\big[\mathcal{L}_{\mathbf{S}}\big(\mathbf{W}_D^{(k-1)}\big)\big], \quad \forall 1 \leq \ell \leq L \quad \text{and} \quad \forall k \geq 1. \tag{19}$$

By recursively applying above equation (4.3), we can write $\mathbf{W}_{D,\ell}^{(k)}$ with respect to $\mathbf{W}_{D,\ell}^{(0)}$ as follows:

$$\mathbf{W}_{D,\ell}^{(k)} = (1 - \eta_2 \mu)^k \mathbf{W}_{D,\ell}^{(0)} - \eta_1 \sum_{\ell=0}^{k-1} (1 - \eta_2 \mu)^\ell \nabla_{\mathbf{W}_\ell}\big[\mathcal{L}_{\mathbf{S}}\big(\mathbf{W}_D^{(k-\ell-1)}\big)\big].$$

Then, we can control the bound as follows:

$$\|\mathbf{W}_{D,\ell}^{(k)} - \mathbf{W}_{D,\ell}^{(0)}\|_2 \leq \left(1 - \big(1 - \eta_2 \mu\big)^k\right) \left\|\mathbf{W}_{D,\ell}^{(0)}\right\|_2 + \frac{\eta_1}{\eta_2 \mu} \max_{\ell=0,\ldots,k-1} \left\|\nabla_{\mathbf{W}_\ell}\big[\mathcal{L}_{\mathbf{S}}\big(\mathbf{W}_D^{(k-\ell-1)}\big)\big]\right\|_2.$$

We know under the initialization setting in our paper, $\|\mathbf{W}_{D,\ell}^{(k)}\|_2 \leq \mathcal{O}(1)$ with high-probability (see Vershynin [2018]), and as long as we can prove the $\ell_2$-norm of gradient is bounded, then we can conclude $\|\mathbf{W}_{D,\ell}^{(k)} - \mathbf{W}_{D,\ell}^{(0)}\|_2 \leq \mathcal{O}(1)$. However, we are not aware of works in which they control the size of $\|\nabla_{\mathbf{W}_\ell}[\mathcal{L}_\mathbf{S}(\mathbf{W}_D^{(k-\ell-1)})]\|_2$ where the non-convex interactions between model parameters across the hidden layers are allowed. To the best of our knowledge, we know the work Allen-Zhu et al. [2019] deals with the three layer case under this setting. But we need further investigations on whether the techniques employed in their paper can be generalized to arbitrary $L$-hidden layer setting.

## D  USEFUL LEMMAS

***A simple fact.*** Suppose $\mathbf{v}_j \overset{\text{i.i.d}}{\sim} \mathcal{N}(0, \frac{\omega}{m})$ for $j \in [m]$. Then, with probability at least $1 - \exp[-\Omega(m)]$, $\|\mathbf{v}\|_2^2 \leq \mathcal{O}(\omega)$.

*Proof.* Since $\left\|\mathbf{v}_j^2\right\|_{\Psi_1} \leq \mathcal{O}\left(\frac{\omega}{m}\right)$ for $j \in [m]$, where $\|\cdot\|_{\Psi_1}$ denotes a sub-exponential norm, Bernstein's inequality for i.i.d. centered sub-exponential random variables can be employed : For any $t \geq 0$,

$$\mathbb{P}\left(\left|\sum_{j=1}^m \left(\mathbf{v}_j^2 - \frac{\omega}{m}\right)\right| \geq t\right) \leq 2\exp\left(-c\min\left(\frac{t^2}{\sum_{j=1}^m \left\|\mathbf{v}_j^2\right\|_{\Psi_1}^2}, \frac{t}{\max_j \left\|\mathbf{v}_j^2\right\|_{\Psi_1}}\right)\right), \quad (20)$$

where $c > 0$ is an absolute constant. Note that we used the fact centering does not hurt the sub-exponentiality of random variable. Choosing $t = \mathcal{O}(\omega)$ concludes the proof. $\qquad\square$

**Lemma 4.1 (Lemma 7.1. Allen-Zhu et al. [2018]).** *With probability at least* $1 - \mathcal{O}(nL) \cdot \exp[-\Omega(m/L)]$, $3/4 \leq \|\mathbf{x}_{\ell,i}^{(0)}\|_2 \leq 5/4$ *for all* $i \in \{1, \ldots, n\}$ *and* $\ell \in \{1, \ldots, L\}$.

**Lemma 4.2 (Lemma B.1. Cao & Gu [2019]).** *If* $\tau \leq \mathcal{O}(L^{-9/2}[\log(m)]^{-3})$, *then with probability at least* $1 - \mathcal{O}(nL) \cdot \exp[-\Omega(m\tau^{2/3}L)]$, $1/2 \leq \|\widetilde{\mathbf{x}}_{\ell,i}\|_2 \leq 3/2$ *for all* $\widetilde{\mathbf{W}} \in \mathcal{B}(\mathcal{W}^{(0)}, \tau)$, $i \in \{1, \ldots, n\}$ *and* $\ell \in \{1, \ldots, L\}$.

**Lemma 4.3 ( Allen-Zhu et al. [2018]).** *Uniformly over* $i \in \{1, \ldots, n\}$ *and* $1 \leq \ell_1 \leq \ell_2 \leq L$, *the following results hold:*

1. ***(Lemma.7.3,*** *Allen-Zhu et al. [2018]) Suppose* $m \geq \Omega(nL\log(nL))$, *then with probability at least,* $1 - \mathcal{O}(nL^2) \cdot \exp[-\Omega(m\tau^{2/3}L)]$,

$$\left\|\prod_{r=\ell_1}^{\ell_2} \mathbf{\Sigma}_{r,i}^{(0)}\mathbf{W}_r^{(0)}\right\|_2 \leq \mathcal{O}(\sqrt{L}).$$

2. ***(Lemma.7.4,*** *Allen-Zhu et al. [2018]) Suppose* $m \geq \Omega(nL\log(nL))$, *then with probability at least,* $1 - \mathcal{O}(nL) \cdot \exp[-\Omega(m/L)]$,

$$\left\|\mathbf{v}^\top\left(\prod_{r=\ell_1}^{L} \mathbf{\Sigma}_{r,i}^{(0)}\mathbf{W}_r^{(0)}\right)\right\|_2 \leq \mathcal{O}(\sqrt{w}).$$

3. ***(Lemma.8.2,*** *Allen-Zhu et al. [2018]) Suppose* $\tau \leq \mathcal{O}(L^{-9/2}[\log(m)]^{-3})$. *For all* $\widetilde{\mathbf{W}} \in \mathcal{B}(\mathcal{W}^{(0)}, \tau)$, *with probability at least,* $1 - \mathcal{O}(nL^2) \cdot \exp[-\Omega(m\tau^{2/3}L)]$,

$$\left\|\widetilde{\mathbf{x}}_{\ell_1,i} - \mathbf{x}_{\ell_1,i}^{(0)}\right\|_2 \leq \mathcal{O}(\tau L^{5/2}\sqrt{\log(m)}).$$

4. ***(Corollary.8.4,*** *Allen-Zhu et al. [2018]) Suppose* $\tau \leq \mathcal{O}(L^{-9/2}[\log(m)]^{-3})$, *then with probability at least,* $1 - \mathcal{O}(nL^2) \cdot \exp[-\Omega(m\tau^{2/3}L)]$,

$$\left\|\widetilde{\mathbf{\Sigma}}_{\ell_1,i} - \mathbf{\Sigma}_{\ell_1,i}^{(0)}\right\|_0 \leq \mathcal{O}(m\tau^{2/3}L).$$

5. (**Lemma.8.7,** Allen-Zhu et al. [2018]) *For all $\ell \in [L]$, let $\boldsymbol{\Sigma}''_{\ell,i} \in [-3,3]^{m \times m}$ be the diagonal matrices with at most $s = \mathcal{O}(m\tau^{2/3}L)$ non-zero entries. For all $\widetilde{\mathbf{W}} \in \mathcal{B}(\mathcal{W}^{(0)}, \tau)$, where $\tau = \mathcal{O}\big(\frac{1}{L^{1.5}}\big)$, with probability at least $1 - \mathcal{O}(nL) \cdot \exp[-\Omega(s\log(m))]$,*

$$\left\| \mathbf{v}^T \Big( \prod_{r=\ell_1+1}^{L} (\boldsymbol{\Sigma}''_{r,i} + \boldsymbol{\Sigma}^{(0)}_{r,i}) \widetilde{\mathbf{W}}_{r,i} \Big)(\boldsymbol{\Sigma}''_{\ell_1,i} + \boldsymbol{\Sigma}^{(0)}_{r,i}) - \mathbf{v}^T \Big( \prod_{r=\ell_1+1}^{L} \boldsymbol{\Sigma}^{(0)}_{r,i} \mathbf{W}^{(0)}_{r,i} \Big)\boldsymbol{\Sigma}^{(0)}_{\ell_1,i} \right\|_2$$
$$\leq \mathcal{O}\big(\tau^{1/3}L^2 \sqrt{\omega \log(m)}\big).$$

**Lemma 4.4** (**Lemma B.3. Cao & Gu [2019]**). *There exists an absolute constant $\kappa$ such that, with probability at least $1 - \mathcal{O}(nL^2) \cdot \exp[-\Omega(m\tau^{2/3}L)]$, $i \in 1, \ldots, n$ and $\ell \in 1, \ldots, L$ and for all $\widetilde{\mathbf{W}} \in \mathcal{B}(\mathcal{W}^{(0)}, \tau)$, with $\tau \leq \kappa L^{-6}[\log(m)]^{-3}$, it holds uniformly that*

$$\big\| \nabla_{\mathbf{W}_\ell}[f_{\widetilde{\mathbf{W}}}(\mathbf{x}_i)] \big\|_2 \leq \mathcal{O}\big(\sqrt{\omega m}\big).$$

**Lemma 4.5.** *Suppose $\widetilde{\mathbf{W}} \in \mathcal{B}(\mathcal{W}^{(0)}, \tau)$ and $\tau \leq \mathcal{O}\big(L^{-9/2}[\log(m)]^{-3}\big)$. For all $u \in \mathbb{R}^m$ with a cardinality $\|u\|_0 \leq s$, for any $1 \leq \ell \leq L$ and $i \in \{1, \ldots, n\}$, with probability at least $1 - \mathcal{O}(nL) \cdot \exp\big(-\Omega(s\log(m))\big) - \mathcal{O}(nL) \cdot \exp\big(-\Omega(m\tau^{2/3}L)\big)$,*

$$\left| \mathbf{v}^\top \Big( \prod_{r=\ell}^{L} \widetilde{\boldsymbol{\Sigma}}_{r,i} \widetilde{\mathbf{W}}_{r,i} \Big)u \right| \leq \sqrt{\frac{\omega s \log(m)}{m}} \cdot \mathcal{O}\big(\|u\|_2\big).$$

*Proof.* Recall Lemma 4.2. For any fixed vector $u \in \mathbb{R}^m$, with probability at least $1 - \mathcal{O}(nL) \cdot \exp[-\Omega(m\tau^{2/3}L)]$ for $\tau \leq \mathcal{O}\big(L^{-9/2}[\log(m)]^{-3}\big)$, for any $1 \leq \ell \leq L$ and $i \in \{1, \ldots, n\}$, we have the event $\mathcal{T}$,

$$\left\| \Big( \prod_{r=\ell}^{L} \widetilde{\boldsymbol{\Sigma}}_{r,i} \widetilde{\mathbf{W}}_{r,i} \Big)u \right\|_2 \leq 3 \|u\|_2. \tag{21}$$

Conditioned on this event happens, it is easy to see the random variable $\mathbf{v}^\top \big( \prod_{r=a}^{L} \widetilde{\boldsymbol{\Sigma}}_{r,i} \widetilde{\mathbf{W}}_{r,i} \big)u \sim SG\big(\frac{9\omega}{m}\|u\|_2^2\big)$. Based on this observation, we have the probability,

$$\mathbb{P}\left( \left| \mathbf{v}^\top \Big( \prod_{r=\ell}^{L} \widetilde{\boldsymbol{\Sigma}}_{r,i} \widetilde{\mathbf{W}}_{r,i} \Big)u \right| \geq \sqrt{\frac{\omega s \log(m)}{m}} \cdot \mathcal{O}\big(\|u\|_2\big) \right)$$
$$\leq \mathbb{P}\left( \left| \mathbf{v}^\top \Big( \prod_{r=\ell}^{L} \widetilde{\boldsymbol{\Sigma}}_{r,i} \widetilde{\mathbf{W}}_{r,i} \Big)u \right| \geq \sqrt{\frac{\omega s \log(m)}{m}} \cdot \mathcal{O}\big(\|u\|_2\big) \;\Big|\; \mathcal{T} \right) + \mathbb{P}\big(\mathcal{T}^c\big)$$
$$\leq \mathcal{O}(nL) \cdot \exp\big(-\Omega(s\log(m))\big) + \mathcal{O}(nL) \cdot \exp\big(-\Omega(m\tau^{2/3}L)\big),$$

where in the last inequality, union bounds over the indices $\ell$ and $i$, and over the vector $u \in \mathbb{R}^m$ with $\|u\|_0 \leq s$ are taken. $\qquad\square$

**Lemma 4.6.** *Suppose $\tau \leq \frac{1}{CL^{9/2}[\log(m)]^3}$ for some constant $C > 0$. Then, for all $i \in [n]$ and $\ell \in [L]$, with probability at least $1 - \mathcal{O}(nL) \cdot \exp[-\Omega(m\tau^{2/3}L)]$, we have*

$$\big\| \nabla_{\mathbf{W}_\ell}\big[f_{\mathbf{W}(k)}(\mathbf{x}_i)\big] - \nabla_{\mathbf{W}_\ell}\big[f_{\mathbf{W}(0)}(\mathbf{x}_i)\big] \big\|_2 \leq \mathcal{O}\left(\tau^{1/3}L^2 \sqrt{\omega m \log(m)}\right).$$

*Proof.* By using the results from Lemma 4.3, we can control the term :

$$\|\nabla_{\mathbf{W}_\ell}\big[f_{\mathbf{W}(k)}(\mathbf{x}_i)\big] - \nabla_{\mathbf{W}_\ell}\big[f_{\mathbf{W}(0)}(\mathbf{x}_i)\big]\|_2$$

$$= \sqrt{m} \cdot \left\| \mathbf{x}_{\ell-1}^{(k)}\mathbf{v}^{\mathrm{T}}\Big(\prod_{r=\ell+1}^{L}\mathbf{\Sigma}_r^{(k)}\mathbf{W}_r^{(k)}\Big)\mathbf{\Sigma}_\ell^{(k)} - \mathbf{x}_{\ell-1}^{(0)}\mathbf{v}^{\mathrm{T}}\Big(\prod_{r=\ell+1}^{L}\mathbf{\Sigma}_r^{(0)}\mathbf{W}_r^{(0)}\Big)\mathbf{\Sigma}_\ell^{(0)} \right\|_2$$

$$\leq \sqrt{m} \cdot \underbrace{\|\mathbf{x}_{\ell-1}^{(k)} - \mathbf{x}_{\ell-1}^{(0)}\|_2}_{\leq \mathcal{O}(\tau L^{5/2}\sqrt{\log(m)})} \cdot \underbrace{\left\|\mathbf{v}^{\mathrm{T}}\Big(\prod_{r=\ell+1}^{L}\mathbf{\Sigma}_r^{(k)}\mathbf{W}_r^{(k)}\Big)\mathbf{\Sigma}_\ell^{(k)}\right\|_2}_{\leq \mathcal{O}(\sqrt{\omega})}$$

$$+ \sqrt{m} \cdot \underbrace{\left\|\mathbf{x}_{\ell-1}^{(0)}\right\|_2}_{\leq \mathcal{O}(1)} \cdot \underbrace{\left\|\mathbf{v}^{\mathrm{T}}\Big(\prod_{r=\ell+1}^{L}\mathbf{\Sigma}_r^{(k)}\mathbf{W}_r^{(k)}\Big)\mathbf{\Sigma}_\ell^{(k)} - \mathbf{v}^{\mathrm{T}}\Big(\prod_{r=\ell+1}^{L}\mathbf{\Sigma}_r^{(0)}\mathbf{W}_r^{(0)}\Big)\mathbf{\Sigma}_\ell^{(0)}\right\|_2}_{\leq \mathcal{O}(\tau^{1/3}L^2\sqrt{\omega\log(m)})}$$

$$\leq \mathcal{O}\Big(\tau^{1/3}L^2\sqrt{\omega m \log(m)}\Big),$$

where, in the last inequality, we used the condition on $\tau \leq \frac{1}{CL^{9/2}[\log(m)]^3} < 1$. $\qquad\square$

**Remark 4.7.** *Note that the results in Lemmas* 6.3 *(second and fifth items),* 6.4, 6.5, 6.6 *are in the setting of* $v_j \sim \mathcal{N}(0, \frac{\omega}{m})$ *for* $j \in [m]$.

For the notational convenience, in following Lemmas we denote $f_{\mathbf{W}(k)}(\mathbf{x}_i)$ as $\mathbf{u}_i(k)$ and let $\mathbf{u}(k) := [\mathbf{u}_1(k), \ldots, \mathbf{u}_n(k)]^\top$ for $k \geq 0$.

**Lemma 4.8.** *For some* $\delta \in [0,1]$, *if* $m \geq \Omega\big(L\log(nL/\delta)\big)$, *then with probability at least* $1 - \delta$, $\|\mathbf{u}(k)\|_2 \leq \mathcal{O}\big(\frac{\sqrt{n\omega}}{\delta}\big)$ *for any* $k \geq 0$.

*Proof.* Recall the Lemma 4.2 stating that $\left\|\mathbf{x}_{L,i}^{(k)}\right\|_2 = \mathcal{O}(1)$ for any input data $\mathbf{x}_i$ for $i \in [n]$. Also recall that $\mathbf{v}_j \sim \mathcal{N}(0, \frac{\omega}{m})$ for $j \in [m]$, $\mathbf{x}_{L,i} \in \mathbb{R}^m$ and $\mathbf{u}_i(k) = \sqrt{m}\mathbf{v}^\top\mathbf{x}_{L,i} \sim \mathcal{N}\big(0, \mathcal{O}(\omega)\big)$. Then, we have a following via simple Markov inequality: for any $t \geq 0$,

$$\mathbb{P}\Big(\|\mathbf{u}(k)\|_2 \geq t\Big) \leq \frac{\mathbb{E}\big[\|\mathbf{u}(k)\|_2\big]}{t} \leq \frac{\sqrt{\mathbb{E}\big[\|\mathbf{u}(k)\|_2^2\big]}}{t} \leq \frac{\mathcal{O}(\sqrt{n\omega})}{t}.$$

$\qquad\square$

**Lemma 4.9.** *For some* $\delta \in [0,1]$, *if* $m \geq \Omega\big(L\log(nL/\delta)\big)$, *then with probability at least* $1 - \delta$, *we have*

$$\|\mathbf{u}(0) - \mathbf{y}\|_2 \leq \mathcal{O}\left(\sqrt{\frac{n}{\delta}}\right).$$

*Proof.* By Markov's inequality, for any $t \geq 0$,

$$\mathbb{P}\Big(\|\mathbf{u}(0) - \mathbf{y}\|_2 \geq t\Big) \leq \frac{\mathbb{E}_{\varepsilon,\mathbf{W}(0),\mathbf{v}}\Big[\|\mathbf{u}(0) - \mathbf{y}\|_2^2\Big]}{t^2}. \tag{22}$$

Note that the expectation in the nominator of (22) is taken over the random noise $\varepsilon$ and initialized parameter $\mathbf{W}(0), \mathbf{v}$. We can expand the nominator as follows:

$$\mathbb{E}_{\varepsilon,\mathbf{W}(0),\mathbf{v}}\Big[\|\mathbf{u}(0) - \mathbf{y}\|_2^2\Big] = \mathbb{E}_{\mathbf{W}(0),\mathbf{v}}\|\mathbf{u}(0)\|_2^2 + \mathbb{E}_\varepsilon\|\mathbf{y}\|_2^2 - 2\mathbb{E}_{\varepsilon,\mathbf{W}(0),\mathbf{v}}\Big[\mathbf{y}^\top\mathbf{u}(0)\Big]. \tag{23}$$

For the convenience of notation, let $\mathbf{y}^* := [f_\rho^\star(\mathbf{x}_1), \ldots, f_\rho^\star(\mathbf{x}_n)]^\top$ and $\varepsilon := [\varepsilon_1, \ldots, \varepsilon_n]^\top$. Recall that we have $\mathbf{y} = \mathbf{y}^* + \varepsilon$, and $\|\mathbf{y}^*\|_2^2 = \mathcal{O}(n)$. Also note that by Lemma 4.1, with probability at least $1 - \mathcal{O}(nL) \cdot \exp[-\Omega(m/L)]$, for any $i = 1, \ldots, n$, $\|\mathbf{x}_{L,i}^{(0)}\|_2^2 = \mathcal{O}(1)$. Then, we have a random

variable $\mathbf{u}_i(0) = \sqrt{m}\mathbf{v}^\top \mathbf{x}_{L,i} \sim \mathcal{N}(0, \mathcal{O}(\omega))$. Now, we are ready to derive the orders of three terms on the RHS of (23).

$$\mathbb{E}_{\mathbf{W}(0),\mathbf{v}}\|\mathbf{u}(0)\|_2^2 = \mathcal{O}(n),$$

$$\mathbb{E}_\varepsilon \|\mathbf{y}\|_2^2 = \mathbb{E}_\varepsilon\Big[\|\mathbf{y}^*\|_2^2 + \|\varepsilon\|_2^2 - 2\mathbf{y}^\top \varepsilon\Big] = \mathcal{O}(n),$$

$$\mathbb{E}_{\varepsilon,\mathbf{W}(0),\mathbf{v}}\Big[\mathbf{y}^\top \mathbf{u}(0)\Big] = \mathbb{E}_{\varepsilon,\mathbf{W}(0),\mathbf{v}}\Big[(\mathbf{y}^* + \varepsilon)^\top \mathbf{u}(0)\Big] = 0.$$

Combining the above three equalities, we conclude the proof. $\qquad\square$

**Lemma 4.10.** *Suppose* $\tau = \mathcal{O}\big(\frac{n\sqrt{\omega}}{\sqrt{m}\delta\lambda_\infty}\big)$. *For some* $\delta \in [0,1]$ *such that* $\delta \geq \mathcal{O}(nL) \cdot \exp[-\Omega(m\tau^{2/3}L)]$, *then with probability at least* $1 - \delta$, *we have*

$$\|\mathbf{H}(k) - \mathbf{H}(0)\|_2 \leq \mathcal{O}\bigg(\omega^{7/6}n^{4/3}L^3 \sqrt[6]{\frac{\log^3(m)}{m\delta\lambda_\infty^2}}\bigg).$$

*Proof.* By the definition of gram matrix $\mathbf{H}_{i,j}(k)$ for any $k \geq 0$, we have

$$|\mathbf{H}_{i,j}(k) - \mathbf{H}_{i,j}(0)|$$

$$= \bigg|\frac{1}{m}\sum_{\ell=1}^L \Big\langle \nabla_{\mathbf{W}_\ell}[f_{\mathbf{W}(k)}(\mathbf{x}_i)], \nabla_{\mathbf{W}_\ell}[f_{\mathbf{W}(k)}(\mathbf{x}_j)]\Big\rangle_{\mathrm{Tr}} - \Big\langle \nabla_{\mathbf{W}_\ell}[f_{\mathbf{W}(0)}(\mathbf{x}_i)], \nabla_{\mathbf{W}_\ell}[f_{\mathbf{W}(0)}(\mathbf{x}_j)]\Big\rangle_{\mathrm{Tr}}\bigg|$$

$$\leq \frac{1}{m}\sum_{\ell=1}^L \bigg\{\Big|\Big\langle \nabla_{\mathbf{W}_\ell}[f_{\mathbf{W}(k)}(\mathbf{x}_i)], \nabla_{\mathbf{W}_\ell}[f_{\mathbf{W}(k)}(\mathbf{x}_j)] - \nabla_{\mathbf{W}_\ell}[f_{\mathbf{W}(0)}(\mathbf{x}_j)]\Big\rangle_{\mathrm{Tr}}\Big|$$

$$+ \Big|\Big\langle \nabla_{\mathbf{W}_\ell}[f_{\mathbf{W}(0)}(\mathbf{x}_j)], \nabla_{\mathbf{W}_\ell}[f_{\mathbf{W}(k)}(\mathbf{x}_i)] - \nabla_{\mathbf{W}_\ell}[f_{\mathbf{W}(0)}(\mathbf{x}_i)]\Big\rangle_{\mathrm{Tr}}\Big|\bigg\}$$

$$\leq \frac{1}{m}\sum_{\ell=1}^L \bigg\{\underbrace{\big\|\nabla_{\mathbf{W}_\ell}[f_{\mathbf{W}(k)}(\mathbf{x}_i)]\big\|_2}_{\leq \mathcal{O}(\sqrt{\omega m})} \cdot \underbrace{\big\|\nabla_{\mathbf{W}_\ell}[f_{\mathbf{W}(k)}(\mathbf{x}_j)] - \nabla_{\mathbf{W}_\ell}[f_{\mathbf{W}(0)}(\mathbf{x}_j)]\big\|_2}_{\leq \mathcal{O}(\tau^{1/3}L^2\sqrt{\omega m \log(m)})}$$

$$+ \underbrace{\big\|\nabla_{\mathbf{W}_\ell}[f_{\mathbf{W}(0)}(\mathbf{x}_j)]\big\|_2}_{\leq \mathcal{O}(\sqrt{\omega m})} \cdot \underbrace{\big\|\nabla_{\mathbf{W}_\ell}[f_{\mathbf{W}(k)}(\mathbf{x}_i)] - \nabla_{\mathbf{W}_\ell}[f_{\mathbf{W}(0)}(\mathbf{x}_i)]\big\|_2}_{\leq \mathcal{O}(\tau^{1/3}L^2\sqrt{\omega m \log(m)})}\bigg\}$$

$$\leq \mathcal{O}\bigg(\omega^{7/6}n^{1/3}L^3 \sqrt[6]{\frac{\log^3(m)}{m\delta\lambda_\infty^2}}\bigg).$$

In the second inequality, Lemmas 4.4 and 4.6 are used, and in the last inequality, $\tau = \mathcal{O}\big(\frac{n\sqrt{\omega}}{\sqrt{m}\delta\lambda_\infty}\big)$ is plugged in. With this, using the fact that Frobenius norm of a matrix is bigger than the operator norm, we bound the term $\|\mathbf{H}(k) - \mathbf{H}(0)\|_2$ as follows:

$$\|\mathbf{H}(k) - \mathbf{H}(0)\|_2 \leq \|\mathbf{H}(k) - \mathbf{H}(0)\|_F \leq \mathcal{O}\bigg(\omega^{7/6}n^{4/3}L^3 \sqrt[6]{\frac{\log^3(m)}{m\delta\lambda_\infty^2}}\bigg).$$

$\qquad\square$

**Lemma 4.11.** *For some* $\delta \in [0,1]$, *with probability at least* $1 - \delta$,

$$\|\mathbf{H}_L^\infty - \mathbf{H}(0)\|_2 \leq \mathcal{O}\bigg(\omega n L^{5/2} \sqrt[4]{\frac{\log(nL/\delta)}{m}}\bigg)$$

*Proof.* For some $\delta' \in [0,1]$, set $\varepsilon = L^{3/2}\sqrt[4]{\frac{\log(L/\delta')}{m}}$ from Theorem 3.1. of Arora et al. [2019b]. For any fixed $i,j \in [n]$, we have

$$\mathbb{P}\bigg[\big|\mathbf{H}_{i,j}^\infty - \mathbf{H}_{i,j}(0)\big| \leq \mathcal{O}\bigg(\omega L^{5/2}\sqrt[4]{\frac{\log(L/\delta')}{m}}\bigg)\bigg] \geq 1 - \delta'.$$

After applying the union bound over all $i, j \in [n]$, setting $\delta = \frac{\delta'}{n^2}$, and using the fact that Frobenius norm of a matrix is bigger than the operator norm, we conclude the proof. $\qquad\square$

For two positive semi-definite matrices $\mathbf{A}$ and $\mathbf{B}$, if we write $\mathbf{A} \succeq \mathbf{B}$, then it means $\mathbf{A} - \mathbf{B}$ is positive semi-definite matrix. Similarly, if we write $\mathbf{A} \succ \mathbf{B}$, then it means $\mathbf{A} - \mathbf{B}$ is positive definite matrix. With these notations, we introduce a following Lemma.

**Lemma 4.12** (**Lemma D.6. Hu et al. [2021]**)**.** *For two positive semi-definite matrices $\mathbf{A}$ and $\mathbf{B}$,*

1. *Suppose $\mathbf{A}$ is non-singular, then $\mathbf{A} \succeq \mathbf{B} \iff \lambda_{max}(\mathbf{B}\mathbf{A}^{-1}) \leq 1$ and $\mathbf{A} \succ \mathbf{B} \iff \lambda_{max}(\mathbf{B}\mathbf{A}^{-1}) < 1$, where $\lambda_{max}(\cdot)$ denotes the maximum eigenvalue of the input matrix.*

2. *Suppose $\mathbf{A}$, $\mathbf{B}$ and $\mathbf{Q}$ are positive definite matrices, $\mathbf{A}$ and $\mathbf{B}$ are exchangeable, then $\mathbf{A} \succeq \mathbf{B} \implies \mathbf{A}\mathbf{Q}\mathbf{A} \succeq \mathbf{B}\mathbf{Q}\mathbf{B}$.*

## E   PROOF OF THEOREM 3.5

For the convenience of notation, denote $u_i(k) = f_{\mathbf{W}(k)}(\mathbf{x}_i)$ and let $\mathbf{u}(k) = \big[u_1(k), u_2(k), \ldots, u_n(k)\big]^\top$. In order to achieve linear convergence rate of the training error, $\|\mathbf{u}(k) - y\|_2^2$, we decompose the term as follows:

$$\|\mathbf{u}(k+1) - y\|_2^2 = \big\|\mathbf{u}(k) - y + \big(\mathbf{u}(k+1) - \mathbf{u}(k)\big)\big\|_2^2$$
$$= \|\mathbf{u}(k) - y\|_2^2 - 2\big(\mathbf{u}(k) - y\big)^\top \big(\mathbf{u}(k+1) - \mathbf{u}(k)\big) + \|\mathbf{u}(k+1) - \mathbf{u}(k)\|_2^2.$$

Equipped with this decomposition, the proof consists of the following steps:

1. Similarly with Du et al. [2019], a term $\big(\mathbf{u}(k+1) - \mathbf{u}(k)\big)$ is decomposed into two terms, where we denote them as $\mathbf{I}_1^{(k)}$ and $\mathbf{I}_2^{(k)}$, respectively. We note that the first term $\mathbf{I}_1^{(k)}$ is related with a gram matrix $\mathbf{H}(k)$ and a second term $\mathbf{I}_2^{(k)}$ can be controlled small enough in $\ell_2$ sense with proper choices of the step size and the width of network.

2. A term $\|\mathbf{u}(k+1) - \mathbf{u}(k)\|_2^2$ needs to be controlled small enough to ensure $2\big(\mathbf{u}(k) - y\big)^\top \big(\mathbf{u}(k+1) - \mathbf{u}(k)\big) > \|\mathbf{u}(k+1) - \mathbf{u}(k)\|_2^2$ so that the loss decreases.

3. It is shown that the distance between the gram matrix $\mathbf{H}(k)$ and the NTK matrix $\mathbf{H}_L^\infty$ is close enough in terms of operator norm.

4. Lastly, we inductively show that the weights generated from gradient descent stay within a perturbation region $\mathcal{B}\big(\mathcal{W}^{(0)}, \tau\big)$, irrespective with the number of iterations of algorithm,

We start the proof by analyzing the term $\mathbf{u}(k+1) - \mathbf{u}(k)$.

**_Step 1. Control on_** $\mathbf{u(k+1)} - \mathbf{u(k)}$**_._**   Recall $\big(\mathbf{\Sigma}_{\ell,i}^{(k)}\big)_{jj} = \mathbb{1}\big(\langle \mathbf{w}_{\ell,j}^{(k)}, \mathbf{x}_{\ell-1,i}^{(k)}\rangle \geq 0\big)$ and we introduce a diagonal matrix $\widetilde{\mathbf{\Sigma}}_{\ell,i}^{(k)}$, whose $j$th entry is defined as follows:

$$\big(\widetilde{\mathbf{\Sigma}}_{\ell,i}^{(k)}\big)_{jj} = \big(\mathbf{\Sigma}_{\ell,i}^{(k+1)} - \mathbf{\Sigma}_{\ell,i}^{(k)}\big)_{jj} \cdot \frac{\langle \mathbf{w}_{\ell,j}^{(k+1)}, \mathbf{x}_{\ell-1,i}^{(k+1)}\rangle}{\langle \mathbf{w}_{\ell,j}^{(k+1)}, \mathbf{x}_{\ell-1,i}^{(k+1)}\rangle - \langle \mathbf{w}_{\ell,j}^{(k)}, \mathbf{x}_{\ell-1,i}^{(k)}\rangle}.$$

With this notation, the difference $\mathbf{x}_{L,i}^{(k+1)} - \mathbf{x}_{L,i}^{(k)}$ can be rewritten via the recursive applications of $\widetilde{\mathbf{\Sigma}}_{\ell,i}^{(k)}$:

$$\mathbf{x}_{L,i}^{(k+1)} - \mathbf{x}_{L,i}^{(k)} = \big(\mathbf{\Sigma}_{L,i}^{(k)} + \widetilde{\mathbf{\Sigma}}_{L,i}^{(k)}\big)\big(\mathbf{W}_L^{(k+1)}\mathbf{x}_{L-1,i}^{(k+1)} - \mathbf{W}_L^{(k)}\mathbf{x}_{L-1,i}^{(k)}\big)$$
$$= \big(\mathbf{\Sigma}_{L,i}^{(k)} + \widetilde{\mathbf{\Sigma}}_{L,i}^{(k)}\big)\mathbf{W}_L^{(k+1)}\big(\mathbf{x}_{L-1,i}^{(k+1)} - \mathbf{x}_{L-1,i}^{(k)}\big) + \big(\mathbf{\Sigma}_{L,i}^{(k)} + \widetilde{\mathbf{\Sigma}}_{L,i}^{(k)}\big)\big(\mathbf{W}_L^{(k+1)} - \mathbf{W}_L^{(k)}\big)\mathbf{x}_{L-1,i}^{(k)}$$
$$= \sum_{\ell=1}^{L} \bigg( \prod_{r=\ell+1}^{L} \big(\mathbf{\Sigma}_{r,i}^{(k)} + \widetilde{\mathbf{\Sigma}}_{r,i}^{(k)}\big)\mathbf{W}_r^{(k+1)} \bigg)\big(\mathbf{\Sigma}_{\ell,i}^{(k)} + \widetilde{\mathbf{\Sigma}}_{\ell,i}^{(k)}\big)\big(\mathbf{W}_\ell^{(k+1)} - \mathbf{W}_\ell^{(k)}\big)\mathbf{x}_{\ell-1,i}^{(k)}$$

$$(24)$$

Then, we introduce following notations :

$$\mathbf{D}_{\ell,i}^{(k)} = \left( \prod_{r=\ell+1}^{L} \mathbf{\Sigma}_{r,i}^{(k)} \mathbf{W}_{r}^{(k)} \right) \mathbf{\Sigma}_{\ell,i}^{(k)}, \qquad \widetilde{\mathbf{D}}_{\ell,i}^{(k)} = \left( \prod_{r=\ell+1}^{L} (\mathbf{\Sigma}_{r,i}^{(k)} + \widetilde{\mathbf{\Sigma}}_{r,i}^{(k)}) \mathbf{W}_{r}^{(k+1)} \right) (\mathbf{\Sigma}_{\ell,i}^{(k)} + \widetilde{\mathbf{\Sigma}}_{\ell,i}^{(k)}).$$

Now, we can write $u_i(k+1) - u_i(k)$ by noting that $u_i(k) = \sqrt{m} \cdot \mathbf{v}^{\mathrm{T}} \mathbf{x}_{L,i}^{(k)}$:

$$\begin{aligned}
u_i(k+1) - u_i(k) &= \sqrt{m} \cdot \mathbf{v}^{\mathrm{T}} \big( \mathbf{x}_{L,i}^{(k+1)} - \mathbf{x}_{L,i}^{(k)} \big) \\
&= \sqrt{m} \cdot \mathbf{v}^{\mathrm{T}} \sum_{\ell=1}^{L} \widetilde{\mathbf{D}}_{\ell,i}^{(k)} \left( \mathbf{W}_{\ell}^{(k+1)} - \mathbf{W}_{\ell}^{(k)} \right) \mathbf{x}_{\ell-1,i}^{(k)} \\
&= \underbrace{-\eta \sqrt{m} \cdot \mathbf{v}^{\mathrm{T}} \sum_{\ell=1}^{L} \mathbf{D}_{\ell,i}^{(k)} \nabla_{\mathbf{W}_{\ell}} \big[ \mathcal{L}_{\mathbf{S}}(\mathbf{W}^{(k)}) \big] \mathbf{x}_{\ell-1,i}^{(k)}}_{\mathbf{I}_{1,i}^{(k)}}
\end{aligned}$$

$$\underbrace{-\eta \sqrt{m} \cdot \mathbf{v}^{\mathrm{T}} \sum_{\ell=1}^{L} \left( \widetilde{\mathbf{D}}_{\ell,i}^{(k)} - \mathbf{D}_{\ell,i}^{(k)} \right) \nabla_{\mathbf{W}_{\ell}} \big[ \mathcal{L}_{\mathbf{S}}(\mathbf{W}^{(k)}) \big] \mathbf{x}_{\ell-1,i}^{(k)}}_{\mathbf{I}_{2,i}^{(k)}}$$

(25)

Here, $\mathbf{I}_{1,i}^{(k)}$ can be rewritten as follows:

$$\begin{aligned}
\mathbf{I}_{1,i}^{(k)} &= -\eta \sqrt{m} \cdot \mathbf{v}^{\mathrm{T}} \sum_{\ell=1}^{L} \mathbf{D}_{\ell,i}^{(k)} \sum_{j=1}^{n} \big( u_j(k) - y_j \big) \nabla_{\mathbf{W}_{\ell}} \big[ f_{\mathbf{W}^{(k)}}(\mathbf{x}_j) \big] \mathbf{x}_{\ell-1,i}^{(k)} \\
&= -\eta \cdot \sum_{j=1}^{n} \big( u_j(k) - y_j \big) \cdot \left( \sqrt{m} \sum_{\ell=1}^{L} \mathbf{v}^{\mathrm{T}} \mathbf{D}_{\ell,i}^{(k)} \nabla_{\mathbf{W}_{\ell}} \big[ f_{\mathbf{W}^{(k)}}(\mathbf{x}_j) \big] \mathbf{x}_{\ell-1,i}^{(k)} \right) \\
&= -m\eta \cdot \sum_{j=1}^{n} \big( u_j(k) - y_j \big) \cdot \frac{1}{m} \sum_{\ell=1}^{L} \left\langle \nabla_{\mathbf{W}_{\ell}} \big[ f_{\mathbf{W}^{(k)}}(\mathbf{x}_i) \big], \nabla_{\mathbf{W}_{\ell}} \big[ f_{\mathbf{W}^{(k)}}(\mathbf{x}_j) \big] \right\rangle_{\mathrm{Tr}} \\
&= -m\eta \cdot \sum_{j=1}^{n} \big( u_j(k) - y_j \big) \cdot \mathbf{H}_{i,j}(k).
\end{aligned}$$

For $\mathbf{I}_{2,i}^{(k)}$, we need a more careful control. First, we pay our attention on bounding the term $\|\mathbf{v}^{\top}(\widetilde{\mathbf{D}}_{\ell,i}^{(k)} - \mathbf{D}_{\ell,i}^{(k)})\|_2$ as follows: By triangle inequality, we have

$$\left\| \mathbf{v}^{\top} \left( \widetilde{\mathbf{D}}_{\ell,i}^{(k)} - \mathbf{D}_{\ell,i}^{(k)} \right) \right\|_2 \leq \left\| \mathbf{v}^{\top} \left( \mathbf{D}_{\ell,i}^{(k)} - \mathbf{D}_{\ell,i}^{(0)} \right) \right\|_2 + \left\| \mathbf{v}^{\top} \left( \widetilde{\mathbf{D}}_{\ell,i}^{(k)} - \mathbf{D}_{\ell,i}^{(0)} \right) \right\|_2. \qquad (26)$$

We control the first term of the right-hand side (R.H.S) in (26). By the fourth item of the Lemma 4.3, we know $\|\mathbf{\Sigma}_{r,i}^{(k)} - \mathbf{\Sigma}_{r,i}^{(0)}\|_0 \leq \mathcal{O}(m\tau^{2/3}L)$ and $|(\mathbf{\Sigma}_{r,i}^{(k)} - \mathbf{\Sigma}_{r,i}^{(0)})_{j,j}| \leq 1$ for $j \in [m]$. Then, we can plug $\mathbf{\Sigma}_{r,i}'' = \mathbf{\Sigma}_{r,i}^{(k)} - \mathbf{\Sigma}_{r,i}^{(0)}$ in the inequality of the fifth item of Lemma 4.3. So, the first term of the R.H.S in (26) can be bounded by $\mathcal{O}(\tau^{1/3}L^2\sqrt{\omega \log(m)})$.

The second term of the R.H.S in (26) can be similarly controlled as the first term. Observe that $|(\mathbf{\Sigma}_{r,i}^{(k)} + \widetilde{\mathbf{\Sigma}}_{r,i}^{(k)})_{jj}| \leq 1$, then we have $|(\mathbf{\Sigma}_{r,i}^{(k)} + \widetilde{\mathbf{\Sigma}}_{r,i}^{(k)} - \mathbf{\Sigma}_{r,i}^{(0)})_{j,j}| \leq 2$ for all $j \in [m]$. Note that by the definition of $\widetilde{\mathbf{\Sigma}}_{r,i}^{(k)}$, we have $\|\widetilde{\mathbf{\Sigma}}_{r,i}^{(k)}\|_0 = \|\mathbf{\Sigma}_{r,i}^{(k+1)} - \mathbf{\Sigma}_{r,i}^{(k)}\|_0 \leq \|\mathbf{\Sigma}_{r,i}^{(k+1)} - \mathbf{\Sigma}_{r,i}^{(0)}\|_0 + \|\mathbf{\Sigma}_{r,i}^{(k)} - \mathbf{\Sigma}_{r,i}^{(0)}\|_0 \leq \mathcal{O}(m\tau^{2/3}L)$. Thus, by the triangle inequality, we have $\|\mathbf{\Sigma}_{r,i}^{(k)} + \widetilde{\mathbf{\Sigma}}_{r,i}^{(k)} - \mathbf{\Sigma}_{r,i}^{(0)}\|_0 \leq \mathcal{O}(m\tau^{2/3}L)$. These observations enable us to plug $\mathbf{\Sigma}_{r,i}'' = \mathbf{\Sigma}_{r,i}^{(k)} + \widetilde{\mathbf{\Sigma}}_{r,i}^{(k)} - \mathbf{\Sigma}_{r,i}^{(0)}$ in the inequality of the fifth item of Lemma 4.3, and give the bound on the second term as $\mathcal{O}(\tau^{1/3}L^2\sqrt{\omega \log(m)})$.

We have $\|\mathbf{v}^\top(\widetilde{\mathbf{D}}_{\ell,i}^{(k)} - \mathbf{D}_{\ell,i}^{(k)})\|_2 \leq \mathcal{O}\big(\tau^{1/3} L^2 \sqrt{\omega \log(m)}\big)$. Now, we control the $\ell_2$-norm of the $\mathbf{I}_2^{(k)}$ as follows:

$$
\begin{aligned}
\left\|\mathbf{I}_2^{(k)}\right\|_2 &\leq \sum_{i=1}^{n}\left|\mathbf{I}_{2,i}^{(k)}\right| \\
&\leq \eta\sqrt{m}\cdot\sum_{i=1}^{n}\left[\sum_{\ell=1}^{L}\underbrace{\left\|\mathbf{v}^\top\left(\widetilde{\mathbf{D}}_{\ell,i}^{(k)} - \mathbf{D}_{\ell,i}^{(k)}\right)\right\|_2}_{\leq\mathcal{O}(L^2\tau^{1/3}\sqrt{\omega\log(m)})}\cdot\left\|\nabla_{\mathbf{W}_\ell}\big[\mathcal{L}_{\mathbf{S}}\big(\mathbf{W}^{(k)}\big)\big]\right\|_2\cdot\underbrace{\left\|\mathbf{x}_{\ell-1,i}^{(k)}\right\|_2}_{\leq\mathcal{O}(1)\,:\,\textbf{Lemma 4.2}}\right] \\
&\leq \mathcal{O}\left(\eta n L^2\tau^{1/3}\sqrt{\omega m\log(m)}\right)\sum_{\ell=1}^{L}\left\|\nabla_{\mathbf{W}_\ell}\big[\mathcal{L}_{\mathbf{S}}\big(\mathbf{W}^{(k)}\big)\big]\right\|_2 \\
&\leq \mathcal{O}\left(\eta n L^{5/2}\tau^{1/3}\sqrt{\omega m\log(m)}\right)\sqrt{\sum_{\ell=1}^{L}\left\|\nabla_{\mathbf{W}_\ell}\big[\mathcal{L}_{\mathbf{S}}\big(\mathbf{W}^{(k)}\big)\big]\right\|_F^2} \\
&\leq \mathcal{O}\left(\eta n L^{5/2}\tau^{1/3}\sqrt{\omega m\log(m)}\right)\sqrt{\sum_{j=1}^{n}\big(u_j(k)-y_j\big)^2\sum_{\ell=1}^{L}\left\|\nabla_{\mathbf{W}_\ell}\big[f_{\mathbf{W}(k)}(\mathbf{x}_j)\big]\right\|_F^2} \\
&\leq \mathcal{O}\left(\eta n L^3\tau^{1/3}\omega m\sqrt{\log(m)}\right)\|\mathbf{u}(k)-y\|_2.
\end{aligned}
\tag{27}
$$

**_Step 2. Control on_** $\|\mathbf{u}(\mathbf{k}+\mathbf{1})-\mathbf{u}(\mathbf{k})\|_2^2$. Recall that by (25), $\mathbf{x}_{L,i}^{(k+1)}-\mathbf{x}_{L,i}^{(k)}$ can be written as follows:

$$
\begin{aligned}
\mathbf{x}_{L,i}^{(k+1)}-\mathbf{x}_{L,i}^{(k)} &= \sum_{\ell=1}^{L}\widetilde{\mathbf{D}}_{\ell,i}^{(k)}\left(\mathbf{W}_\ell^{(k+1)}-\mathbf{W}_\ell^{(k)}\right)\mathbf{x}_{\ell-1,i}^{(k)} \\
&= -\eta\cdot\sum_{\ell=1}^{L}\widetilde{\mathbf{D}}_{\ell,i}^{(k)}\nabla_{\mathbf{W}_\ell}\big[\mathcal{L}_{\mathbf{S}}\big(\mathbf{W}^{(k)}\big)\big]\mathbf{x}_{\ell-1,i}^{(k)}.
\end{aligned}
$$

It is worth noting that,

$$
\begin{aligned}
\left\|\nabla_{\mathbf{W}_\ell}\big[\mathcal{L}_{\mathbf{S}}\big(\mathbf{W}^{(k)}\big)\big]\right\|_2^2 &= \left\|\sum_{j=1}^{n}\big(u_j(k)-y_j\big)\nabla_{\mathbf{W}_\ell}\big[f_{\mathbf{W}(k)}(\mathbf{x}_j)\big]\right\|_2^2 \\
&\leq \sum_{j=1}^{n}\big(u_j(k)-y_j\big)^2\sum_{j=1}^{n}\left\|\nabla_{\mathbf{W}_\ell}\big[f_{\mathbf{W}(k)}(\mathbf{x}_j)\big]\right\|_2^2 \\
&\leq \mathcal{O}(nm\omega)\|\mathbf{u}(k)-y\|_2^2.
\end{aligned}
\tag{28}
$$

Also, observe that $|\big(\boldsymbol{\Sigma}_{r,i}^{(k)} + \widetilde{\boldsymbol{\Sigma}}_{r,i}^{(k)}\big)_{jj}| \leq 1$ for all $j \in [m]$, so by Lemma A.3 of Zou et al. [2020], we know $\|\widetilde{\mathbf{D}}_{\ell,i}^{(k)}\|_2 \leq \mathcal{O}\big(\sqrt{L}\big)$. Combining all the facts, we can conclude:

$$
\begin{aligned}
\|\mathbf{u}(k+1)-\mathbf{u}(k)\|_2^2 &= m\cdot\sum_{i=1}^{n}\left(\mathbf{v}^{\mathrm{T}}\mathbf{x}_{L,i}^{(k+1)}-\mathbf{v}^{\mathrm{T}}\mathbf{x}_{L,i}^{(k)}\right)^2 \\
&\leq m\cdot\|\mathbf{v}\|_2^2\sum_{i=1}^{n}\left\|\mathbf{x}_{L,i}^{(k+1)}-\mathbf{x}_{L,i}^{(k)}\right\|_2^2 \\
&\leq \eta^2 m\cdot\|\mathbf{v}\|_2^2\sum_{i=1}^{n}\left[\sum_{\ell=1}^{L}\left\|\widetilde{\mathbf{D}}_{\ell,i}^{(k)}\right\|_2^2\cdot\left\|\nabla_{\mathbf{W}_\ell}\big[\mathcal{L}_{\mathbf{S}}\big(\mathbf{W}^{(k)}\big)\big]\right\|_2^2\cdot\left\|\mathbf{x}_{\ell-1,i}^{(k)}\right\|_2^2\right] \\
&\leq \mathcal{O}\big(\eta^2 n^2 L^2 m^2\omega^2\big)\|\mathbf{u}(k)-y\|_2^2 \\
&\leq \mathcal{O}\big(\eta^2 n^2 L^2 m^2\big)\|\mathbf{u}(k)-y\|_2^2,
\end{aligned}
\tag{29}
$$

where in the third inequality, we additionally used the fact $\|\mathbf{v}\|_2^2 = \mathcal{O}(\omega)$ with probability at least $1 - \exp(-\Omega(m))$, and the inequality (28). In the last inequality, we used the assumption $\omega \leq 1$.

**Step 3.** $\lambda_{min}(H(k)) \geq \frac{\lambda_\infty}{2}$ **with sufficiently large** $m$. Denote $\rho(A)$ as a sprectral radius of a matrix $A$. Then, we have

$$
\begin{aligned}
\|\mathbf{H}(k) - \mathbf{H}_L^\infty\|_2 &\geq \rho(\mathbf{H}(k) - \mathbf{H}_L^\infty) \\
&\geq -\lambda_{\min}(\mathbf{H}(k) - \mathbf{H}_L^\infty) \\
&\geq \lambda_{\min}(\mathbf{H}_L^\infty) - \lambda_{\min}(\mathbf{H}(k)) \\
&\geq \lambda_\infty - \lambda_{\min}(\mathbf{H}(k)),
\end{aligned}
\tag{30}
$$

where, in the second inequality, we used a triangle inequality, $\lambda_{\min}(\mathbf{H}(k) - \mathbf{H}_L^\infty) + \lambda_{\min}(\mathbf{H}_L^\infty) \leq \lambda_{\min}(\mathbf{H}(k))$. By Lemmas 4.10 and 4.11, setting $m \geq \Omega\left(\omega^7 n^8 L^{18} \frac{\log^3(m)}{\lambda_\infty^8 \delta}\right)$ and $\tilde{\mathcal{O}}\left(\frac{\lambda_\infty^{4/3} \delta^{1/3}}{n^{4/3} L^4}\right) \leq \omega \leq 1$, we have

$$
\begin{aligned}
\|\mathbf{H}(k) - \mathbf{H}_L^\infty\|_2 &\leq \|\mathbf{H}(k) - \mathbf{H}(0)\|_2 + \|\mathbf{H}(0) - \mathbf{H}_L^\infty\|_2 \\
&\leq \mathcal{O}\left(\omega^{7/6} n^{4/3} L^3 \sqrt[6]{\frac{\log^3(m)}{m\delta\lambda_\infty^2}}\right) + \mathcal{O}\left(\omega n^2 L^{5/2} \sqrt[4]{\frac{\log(nL/\delta)}{m}}\right) \\
&\leq \mathcal{O}\left(\omega^{7/6} n^{4/3} L^3 \sqrt[6]{\frac{\log^3(m)}{m\delta\lambda_\infty^2}}\right) \\
&\leq \frac{\lambda_\infty}{2}.
\end{aligned}
\tag{31}
$$

Thus, combining (30) and (31) yields that $\lambda_{\min}(H(k)) \geq \frac{\lambda_\infty}{2}$.

**Step 4. Concluding the proof.** Recall that $\mathbf{I}_1^{(k)} = -m\eta \cdot \mathbf{H}(k)(\mathbf{u}(k) - y)$. Then observe that

$$
\begin{aligned}
(\mathbf{u}(k) - y)^\top \mathbf{I}_1^{(k)} &= -\eta m \cdot (\mathbf{u}(k) - y)^\top \mathbf{H}(k)(\mathbf{u}(k) - y) \\
&\leq -\eta m \cdot \lambda_{\min}(\mathbf{H}(k)) \|\mathbf{u}(k) - y\|_2^2 \\
&\leq -\eta m \cdot \frac{\lambda_\infty}{2} \|\mathbf{u}(k) - y\|_2^2.
\end{aligned}
\tag{32}
$$

We set the step size $\eta$, radius of perturbation region $\tau$ and network width $m$ as follows,

$$
\begin{aligned}
\eta &= \Omega\left(\frac{\lambda_\infty}{n^2 L^2 m}\right), \\
\tau &= \mathcal{O}\left(\frac{n\sqrt{\omega}}{\sqrt{m}\delta\lambda_\infty}\right), \\
m &\geq \Omega\left(\omega^7 n^8 L^{18} \frac{\log^3(m)}{\lambda_\infty^8 \delta}\right).
\end{aligned}
$$

With the above settings, we can control the $\|\mathbf{u}(k+1) - y\|_2^2$ by combining (27), (29) and (32) as follows,

$$
\begin{aligned}
\|\mathbf{u}(k+1) - y\|_2^2 &= \left\|\mathbf{u}(k) - y + (\mathbf{u}(k+1) - \mathbf{u}(k))\right\|_2^2 \\
&= \|\mathbf{u}(k) - y\|_2^2 - 2\eta m \cdot (\mathbf{u}(k) - y)^\top \mathbf{H}(k)(\mathbf{u}(k) - y) \\
&\quad - (\mathbf{u}(k) - y)^\top \mathbf{I}_2^{(k)} + \|\mathbf{u}(k+1) - \mathbf{u}(k)\|_2^2 \\
&\leq \left(1 - \eta m\lambda_\infty + \mathcal{O}(\eta n L^3 \tau^{1/3} m\omega\sqrt{\log(m)}) + \mathcal{O}(\eta^2 n^2 L^2 m^2)\right) \|\mathbf{u}(k) - y\|_2^2 \\
&\leq \left(1 - \frac{\eta m\lambda_\infty}{2}\right) \|\mathbf{u}(k) - y\|_2^2.
\end{aligned}
$$

So far, we have shown from Step 1 to Step 4 that given the radius of perturbation region $\tau$ has the order $\mathcal{O}\left(\frac{n\sqrt{\omega}}{\sqrt{m}\delta\lambda_\infty}\right)$, then we can show the training error drops linearly to 0 with the discount factor $\left(1 - \frac{\eta m\lambda_\infty}{2}\right)$ along with the proper choices of $\eta$ and $m$. It remains us to prove the iterates $\mathbf{W}_\ell^{(k)}$ for all $\ell \in [L]$ generated by GD algorithm indeed stay in the perturbation region $\mathcal{B}(\mathcal{W}^{(0)}, \tau)$ over $k \geq 0$ with $\tau = \mathcal{O}\left(\frac{n\sqrt{\omega}}{\sqrt{m}\delta\lambda_\infty}\right)$.

***Step 5. The order of the radius of perturbation region.*** We employ the induction process for the proof. The induction hypothesis is : $\forall s \in [k+1]$,

$$\left\|\mathbf{W}_\ell^{(s)} - \mathbf{W}_\ell^{(0)}\right\|_2 \leq \eta \cdot \mathcal{O}\left(\sqrt{nm\omega}\right) \sum_{t=0}^{s-1}\left(1 - \frac{\eta m\lambda_\infty}{2}\right)^{\frac{t}{2}} \mathcal{O}\left(\sqrt{\frac{n}{\delta}}\right) \leq \mathcal{O}\left(\frac{n\sqrt{\omega}}{\sqrt{m}\delta\lambda_\infty}\right). \quad (33)$$

First, it is easy to see it holds for $s = 0$. Now, suppose it holds for $s = 0, \ldots, k$, we consider $s = k+1$.

$$\left\|\mathbf{W}_\ell^{(k+1)} - \mathbf{W}_\ell^{(k)}\right\|_2 = \left\|\nabla_{\mathbf{W}_\ell}[\mathcal{L}_{\mathbf{S}}(\mathbf{W}^{(k)})]\right\|_2$$

$$= \eta \cdot \left\|\sum_{j=1}^n \left(u_j(k) - y_j\right)\nabla_{\mathbf{W}_\ell}[f_{\mathbf{W}(k)}(\mathbf{x}_j)]\right\|_2$$

$$\leq \eta \cdot \sqrt{\sum_{j=1}^n \left\|\nabla_{\mathbf{W}_\ell}[f_{\mathbf{W}(k)}(\mathbf{x}_j)]\right\|_2^2}\sqrt{\sum_{j=1}^n \left(u_j(k) - y_j\right)^2}$$

$$\leq \eta \cdot \mathcal{O}\left(\sqrt{nm\omega}\right)\sqrt{2\mathcal{L}_{\mathbf{S}}(\mathbf{W}^{(k)})}$$

$$\leq \eta \cdot \mathcal{O}\left(\sqrt{nm\omega}\right)\left(1 - \frac{\eta m\lambda_\infty}{2}\right)^{\frac{k}{2}} \mathcal{O}\left(\sqrt{\frac{n}{\delta}}\right), \quad (34)$$

where in the second inequality, we used Lemmas 4.4. Note that since it is assumed that $\mathbf{W}_\ell^{(k)} \in \mathcal{B}(\mathcal{W}^{(0)}, \tau)$, the Lemma is applicable with $m \geq \Omega\left(\omega^7 n^8 L^{18}\frac{\log^3(m)}{\lambda_\infty^8 \delta}\right)$. Similarly, since it is assumed that the induction hypothesis holds for $s = 0, \ldots, k$, we can see $\|\mathbf{u}(k) - y\|_2^2 \leq \left(1 - \frac{\eta m\lambda_\infty}{2}\right)^k \|\mathbf{u}(0) - y\|_2^2$. This inequality is plugged in the last inequality with Lemma 4.9.

By combining the inequalities (33) for $s \in [k]$ and (34), and triangle inequality, we conclude the proof:

$$\left\|\mathbf{W}_\ell^{(k+1)} - \mathbf{W}_\ell^{(0)}\right\|_2 \leq \eta \cdot \mathcal{O}\left(\sqrt{nm\omega}\right)\sum_{t=0}^k\left(1 - \frac{\eta m\lambda_\infty}{2}\right)^{\frac{t}{2}}\mathcal{O}\left(\sqrt{\frac{n}{\delta}}\right) \leq \mathcal{O}\left(\frac{n\sqrt{\omega}}{\sqrt{m}\delta\lambda_\infty}\right).$$

**Proposition 5.1.** *For some $\delta \in [0,1]$, set the width of the network as $m \geq \Omega\left(\omega^7 n^8 L^{18}\frac{\log^3(m)}{\lambda_\infty^8 \delta^2}\right)$, and the step-size of gradient descent as $\eta = \mathcal{O}\left(\frac{\lambda_\infty}{n^2 L^2 m}\right)$. Then, with probability at least $1 - \delta$ over the randomness of initialized parameters $\mathcal{W}^{(0)}$, we have for $k = 0, 1, 2, \ldots$,*

$$\mathbf{u}(k) - y = \left(\mathbf{I} - \eta m\mathbf{H}_L^\infty\right)^k\left(\mathbf{u}(0) - y\right) + \xi(k),$$

*where*

$$\|\xi(k)\|_2 = k\left(1 - \frac{\eta m\lambda_\infty}{2}\right)^{k-1}\mathcal{O}\left(\eta m \cdot \omega^{7/6}n^{4/3}L^3\sqrt[6]{\frac{\log^3(m)}{m\lambda_\infty^2\delta}}\right)\|\mathbf{y} - \mathbf{u}(0)\|_2.$$

*Proof.* Define $u_i(k) := f_{\mathbf{W}(k)}(\mathbf{x}_i)$, then we have

$$\mathbf{u}(k+1) - \mathbf{u}(k) = -\eta m \cdot \mathbf{H}(k)\left(\mathbf{u}(k) - y\right) + \mathbf{I}_1^{(k)}$$

$$= -\eta m \cdot \mathbf{H}_L^\infty\left(\mathbf{u}(k) - y\right) - \eta m \cdot \left(\mathbf{H}(k) - \mathbf{H}_L^\infty\right)\left(\mathbf{u}(k) - y\right) + \mathbf{I}_1^{(k)}$$

$$= -\eta m \cdot \mathbf{H}_L^\infty\left(\mathbf{u}(k) - y\right) + \mathbf{e}(k).$$

By recursively applying the above equality, we can easily derive a following for any $k \geq 0$,

$$\mathbf{u}(k) - y = \left(\mathbf{I} - \eta m \mathbf{H}_L^\infty\right)^k \left(\mathbf{u}(0) - y\right) + \underbrace{\sum_{t=0}^{k-1} \left(\mathbf{I} - \eta m \mathbf{H}_L^\infty\right)^t \mathbf{e}(k-1-t)}_{=\xi(k)}. \tag{35}$$

Now, we want to show $\xi(k)$ can be controlled in arbitrarily small number. First, $e(k)$ needs to be bounded in an $\ell_2$ norm:

$$\|e(k)\|_2 \leq \eta m \left\|\mathbf{H}_L^\infty - \mathbf{H}(k)\right\|_2 \|\mathbf{u}(k) - y\|_2 + \left\|\mathbf{I}_2^{(k)}\right\|_2$$

$$\leq \eta m \cdot \mathcal{O}\left(\omega^{7/6} n^{4/3} L^3 \sqrt[6]{\frac{\log^3(m)}{m\lambda_\infty^2 \delta}}\right) \|\mathbf{u}(k) - y\|_2,$$

where, in the second inequality, $\tau = \mathcal{O}\left(\frac{n\sqrt{\omega}}{\sqrt{m\delta}\lambda_\infty}\right)$ is plugged in (27). Equipped with the bound on $\|e(k)\|_2$, we can easily bound the $\|\xi(k)\|_2$ as follows:

$$\left\|\sum_{t=0}^{k-1} \left(\mathbf{I} - \eta m \mathbf{H}_L^\infty\right)^t \mathbf{e}(k-1-t)\right\|_2$$

$$\leq \sum_{t=0}^{k-1} \left\|\mathbf{I} - \eta m \mathbf{H}_L^\infty\right\|_2^t \|\mathbf{e}(k-1-t)\|_2$$

$$\leq \sum_{t=0}^{k-1} \left(1 - \eta m \lambda_\infty\right)^t \mathcal{O}\left(\eta m \cdot \omega^{7/6} n^{4/3} L^3 \sqrt[6]{\frac{\log^3(m)}{m\lambda_\infty^2 \delta}}\right) \|\mathbf{u}(k-1-t) - y\|_2$$

$$\leq \sum_{t=0}^{k-1} \left(1 - \eta m \lambda_\infty\right)^t \mathcal{O}\left(\eta m \cdot \omega^{7/6} n^{4/3} L^3 \sqrt[6]{\frac{\log^3(m)}{m\lambda_\infty^2 \delta}}\right) \left(1 - \frac{\eta m \lambda_\infty}{2}\right)^{k-1-t} \|\mathbf{u}(0) - y\|_2$$

$$= k\left(1 - \frac{\eta m \lambda_\infty}{2}\right)^{k-1} \mathcal{O}\left(\eta m \cdot \omega^{7/6} n^{4/3} L^3 \sqrt[6]{\frac{\log^3(m)}{m\lambda_\infty^2 \delta^2}}\right) \|\mathbf{u}(0) - y\|_2. \tag{36}$$

Note that in the third inequality, we used the result from Theorem 1. $\qquad\square$

## F  PROOF OF THEOREM 3.8

We begin the proof by decomposing the error $\widehat{f}_{\mathbf{W}^{(k)}}(x) - f^*(x)$ for any fixed $x \in \mathbf{Unif}(\mathcal{S}^{d-1})$ into two terms as follows:

$$\widehat{f}_{\mathbf{W}^{(k)}}(x) - f^*(x) = \underbrace{\left(\widehat{f}_{\mathbf{W}^{(k)}}(x) - g^*(x)\right)}_{\Delta_1} + \underbrace{\left(g^*(x) - f^*(x)\right)}_{\Delta_2}. \tag{37}$$

Here, we denote the solution of kernel regression with kernel $\mathbf{H}_L^\infty$ as $g^*(x)$, which is a minimum RKHS norm interpolant of the noise-free data set $\{\mathbf{x}_i, f_\rho^\star(\mathbf{x}_i)\}_{i=1}^n$. To avoid the confusion of the notation, we write $\mathbf{Ker}(x, \mathbf{X}) = \left(\mathbf{H}_L^\infty(x, \mathbf{x}_1), \ldots, \mathbf{H}_L^\infty(x, \mathbf{x}_n)\right)_{i=1}^n \in \mathbb{R}^n$ and let $\mathbf{y}^* = [f_\rho^\star(\mathbf{x}_1), \ldots, f_\rho^\star(\mathbf{x}_n)]^\top$. Then, we have a following closed form solution $g^*(x)$ as,

$$g^*(x) := \mathbf{Ker}(x, \mathbf{X})\left(\mathbf{H}_L^\infty\right)^{-1} \mathbf{y}^*.$$

With the decomposition (37), the proof sketch of Theorem 3.8 is as follows.

1. Note that for any $\ell \in [L]$, we have $\widehat{f}_{\mathbf{W}(k)}(x) = \langle \mathbf{vec}(\nabla_{\mathbf{W}_\ell}[f_{\mathbf{W}(k)}(x)]), \mathbf{vec}(\mathbf{W}_\ell^{(k)}) \rangle$. We can write the term $\mathbf{vec}(\mathbf{W}_\ell^{(k)})$ with respect to $\mathbf{vec}(\mathbf{W}_\ell^{(0)})$, $\mathbf{H}_L^\infty$ and the residual term via recursive applications of GD update rule and the result from proposition 2.1. Readers can refer (38). Using the equality (38), we can further decompose $\Delta_1$ into three terms. That is, $\Delta_1 = \Delta_{11} + \Delta_{12} + \Delta_{13}$. Then, using the boundedness of $\ell_2$-norm of network gradient and the fact that the size of $\|\xi(k)\|_2$ can be controlled with wide enough network, we can control the size of $\|\Delta_{12}\|_2$ and $\|\Delta_{13}\|_2$ arbitarily small.

2. In the term $\Delta_2$, the $g^\star$ is an interpolant based on noiseless data. For large enough data points, $g^\star$ converges fastly to $f^\star$ at the rate $\mathcal{O}_\mathbb{P}(\frac{1}{\sqrt{n}})$.

3. Lastly, the $\Delta_{11}$ is the only term that is involved with random error $\varepsilon$, and we show that $\|\Delta_{11}\|_2$ is bounded away from 0 for small and large GD iteration index $k$.

***Step 1. Control on $\Delta_1$.*** For $n$ data points $(\mathbf{x}_1, \ldots, \mathbf{x}_n)$ and for the $k^{\text{th}}$ updated parameter $\mathbf{W}(k)$, denote:

$$\nabla_{\mathbf{W}_\ell}[f_{\mathbf{W}(k)}(\mathbf{X})] = \left[\mathbf{vec}\left(\nabla_{\mathbf{W}_\ell}[f_{\mathbf{W}(k)}(\mathbf{x}_1)]\right), \cdots, \mathbf{vec}\left(\nabla_{\mathbf{W}_\ell}[f_{\mathbf{W}(k)}(\mathbf{x}_n)]\right)\right].$$

Note that when $\ell = 1$, $\nabla_{\mathbf{W}_\ell}[f_{\mathbf{W}(k)}(\mathbf{X})] \in \mathbb{R}^{md \times n}$ and when $\ell = 2, \ldots, L$, $\nabla_{\mathbf{W}_\ell}[f_{\mathbf{W}(k)}(\mathbf{X})] \in \mathbb{R}^{m^2 \times n}$. With this notation, we can rewrite the Gradient Descent update rule as

$$\mathbf{vec}(\mathbf{W}_\ell^{(k+1)}) = \mathbf{vec}(\mathbf{W}_\ell^{(k)}) - \eta \nabla_{\mathbf{W}_\ell}[f_{\mathbf{W}(k)}(\mathbf{X})](\mathbf{u}(k) - \mathbf{y}), \quad k \geq 0.$$

Applying Proposition 3.8, we can get :

$$\mathbf{vec}(\mathbf{W}_\ell^{(k)}) - \mathbf{vec}(\mathbf{W}_\ell^{(0)})$$

$$= \sum_{j=0}^{k-1}\left(\mathbf{vec}(\mathbf{W}_\ell^{(j+1)}) - \mathbf{vec}(\mathbf{W}_\ell^{(j)})\right)$$

$$= -\eta \cdot \sum_{j=0}^{k-1} \nabla_{\mathbf{W}_\ell}[f_{\mathbf{W}(j)}(\mathbf{X})](\mathbf{u}(j) - \mathbf{y})$$

$$= \eta \cdot \sum_{j=0}^{k-1} \nabla_{\mathbf{W}_\ell}[f_{\mathbf{W}(j)}(\mathbf{X})](\mathbf{I} - \eta m \mathbf{H}_L^\infty)^j (\mathbf{y} - \mathbf{u}(0)) - \eta \cdot \sum_{j=0}^{k-1} \nabla_{\mathbf{W}_\ell}[f_{\mathbf{W}(k)}(\mathbf{X})]\xi(j)$$

$$= \eta \cdot \sum_{j=0}^{k-1} \nabla_{\mathbf{W}_\ell}[f_{\mathbf{W}(0)}(\mathbf{X})](\mathbf{I} - \eta m \mathbf{H}_L^\infty)^j (y - \mathbf{u}(0)) - \eta \cdot \sum_{j=0}^{k-1} \nabla_{\mathbf{W}_\ell}[f_{\mathbf{W}(k)}(\mathbf{X})]\xi(j)$$

$$\qquad + \eta \cdot \sum_{j=0}^{k-1}\left(\left[\nabla_{\mathbf{W}_\ell}[f_{\mathbf{W}(j)}(\mathbf{X})] - \nabla_{\mathbf{W}_\ell}[f_{\mathbf{W}(0)}(\mathbf{X})]\right](\mathbf{I} - \eta m \mathbf{H}_L^\infty)^j (y - \mathbf{u}(0))\right)$$

$$= \eta \cdot \sum_{j=0}^{k-1} \nabla_{\mathbf{W}_\ell}[f_{\mathbf{W}(0)}(\mathbf{X})](\mathbf{I} - \eta m \mathbf{H}_L^\infty)^j (y - \mathbf{u}(0)) + \xi'(k). \tag{38}$$

First, we control $\ell_2$-norm of the first term of $\xi'(k)$ as follows: Note that $\|\nabla_{\mathbf{W}_\ell}[f_{\mathbf{W}(j)}(\mathbf{X})]\|_F \leq \mathcal{O}(\sqrt{nm\omega})$ by Lemma 4.4 for $0 \leq j \leq k-1$. Then, we have

$$\left\|\eta \cdot \sum_{j=0}^{k-1} \nabla_{\mathbf{W}_\ell}[f_{\mathbf{W}(j)}(\mathbf{X})]\xi(j)\right\|_2$$

$$\leq \sum_{j=0}^{k-1} \mathcal{O}(\eta\sqrt{nm\omega}) \mathcal{O}\left(j\left(1 - \frac{\eta m \lambda_\infty}{2}\right)^{j-1}\right) \mathcal{O}\left(\eta m \cdot \omega^{7/6} n^{4/3} L^3 \sqrt[6]{\frac{\log^3(m)}{m\lambda_\infty^2 \delta}}\right) \|\mathbf{y} - \mathbf{u}(0)\|_2$$

$$\leq \mathcal{O}\left(\frac{n^{11/6} L^3 \omega^{5/3}}{m^{2/3} \lambda_\infty^{7/3} \delta^{1/6}} \sqrt{\log(m)}\right) \|\mathbf{y} - \mathbf{u}(0)\|_2. \tag{39}$$

In the second inequality, $\sum_{j=1}^{\infty} j\left(1 - \frac{\eta m \lambda_{\infty}}{2}\right)^j = \mathcal{O}\left(\frac{1}{\eta^2 m^2 \lambda_{\infty}^2}\right)$ is used. Then, we control $\ell_2$-norm of the second term of $\xi'(k)$ as follows:

$$
\left\| \eta \cdot \sum_{j=0}^{k-1} \left[ \nabla_{\mathbf{W}_\ell}\left[f_{\mathbf{W}(j)}(\mathbf{X})\right] - \nabla_{\mathbf{W}_\ell}\left[f_{\mathbf{W}(0)}(\mathbf{X})\right] \right] \left(\mathbf{I} - \eta m \mathbf{H}_L^{\infty}\right)^j \left(y - \mathbf{u}(0)\right) \right\|_2
$$
$$
\leq \sum_{j=0}^{k-1} \eta \left\| \mathbf{I} - \eta m \mathbf{H}_L^{\infty} \right\|_2^j \left\| \mathbf{y} - \mathbf{u}(0) \right\|_2 \sqrt{ \sum_{i=1}^{n} \left\| \nabla_{\mathbf{W}_\ell}\left[f_{\mathbf{W}(j)}(\mathbf{x}_i)\right] - \nabla_{\mathbf{W}_\ell}\left[f_{\mathbf{W}(0)}(\mathbf{x}_i)\right] \right\|_2^2 }
$$
$$
\leq \sum_{j=0}^{k-1} \eta \left(1 - \eta m \lambda_{\infty}\right)^j \mathcal{O}\left( \frac{n^{1/3} m^{1/3} L^2 \omega^{2/3}}{\lambda_{\infty}^{1/3} \delta^{1/6}} \sqrt{\log(m)} \right) \mathcal{O}(\sqrt{n}) \left\| \mathbf{y} - \mathbf{u}(0) \right\|_2
$$
$$
\leq \mathcal{O}\left( \frac{n^{5/6} L^2 \omega^{2/3}}{m^{2/3} \lambda_{\infty}^{4/3} \delta^{1/6}} \sqrt{\log(m)} \right) \left\| \mathbf{y} - \mathbf{u}(0) \right\|_2, \tag{40}
$$

where in the second inequality, we used Lemmas 4.6 with $\tau = \mathcal{O}\left(\frac{n\sqrt{\omega}}{\sqrt{m}\delta\lambda_{\infty}}\right)$.

Now, we are ready to control $\Delta_1$ term. By using the equality (38), we can decompose the term $\Delta_1$ as follows: Let us denote $G_k = \sum_{j=0}^{k-1} \eta m \left(\mathbf{I} - \eta m \mathbf{H}_L^{\infty}\right)^j$. Note that for any $\ell \in [L]$, $\widehat{f}_{\mathbf{W}^{(k)}}(x) = \langle \mathbf{vec}\left(\nabla_{\mathbf{W}_\ell}\left[f_{\mathbf{W}(k)}(x)\right]\right), \mathbf{vec}\left(\mathbf{W}_\ell^{(k)}\right) \rangle$ and recall that $\mathbf{y} = \mathbf{y}^* + \varepsilon$. Then, for any fixed $\ell' \in [L]$, we have:

$$
\Delta_1 = \left[ \left\langle \mathbf{vec}\left(\nabla_{\mathbf{W}_{\ell'}}\left[f_{\mathbf{W}(k)}(x)\right]\right), \mathbf{vec}\left(\mathbf{W}_{\ell'}^{(k)}\right) \right\rangle - \mathbf{Ker}(x,\mathbf{X})\left(\mathbf{H}_L^{\infty}\right)^{-1}\mathbf{y}^* \right]
$$
$$
\qquad + \mathbf{Ker}(x,\mathbf{X})G_k\mathbf{y} - \mathbf{Ker}(x,\mathbf{X})G_k\mathbf{y}
$$
$$
= \underbrace{\left[ \mathbf{Ker}(x,\mathbf{X})\left[G_k - \left(\mathbf{H}_L^{\infty}\right)^{-1}\right]\mathbf{y}^* + \mathbf{Ker}(x,\mathbf{X})G_k\varepsilon \right]}_{=\Delta_{11}}
$$
$$
\qquad + \underbrace{\left[ \frac{1}{m}\sum_{\ell=1}^{L} \mathbf{vec}\left(\nabla_{\mathbf{W}_\ell}\left[f_{\mathbf{W}(k)}(x)\right]\right)^\top \nabla_{\mathbf{W}_\ell}\left[f_{\mathbf{W}(0)}(\mathbf{X})\right] - \mathbf{Ker}(x,\mathbf{X}) \right] G_k\mathbf{y}}_{}
$$
$$
\qquad \underbrace{- \frac{1}{m}\sum_{\ell:\ell\neq\ell'} \mathbf{vec}\left(\nabla_{\mathbf{W}_\ell}\left[f_{\mathbf{W}(k)}(x)\right]\right)^\top \nabla_{\mathbf{W}_\ell}\left[f_{\mathbf{W}(0)}(\mathbf{X})\right] G_k\mathbf{y}}_{=\Delta_{12}}
$$
$$
\qquad + \underbrace{\left[ \left\langle \mathbf{vec}\left(\nabla_{\mathbf{W}_{\ell'}}\left[f_{\mathbf{W}(k)}(x)\right]\right), \mathbf{vec}\left(\mathbf{W}_{\ell'}^{(0)}\right) \right\rangle + \mathbf{vec}\left(\nabla_{\mathbf{W}_{\ell'}}\left[f_{\mathbf{W}(k)}(x)\right]\right)^\top \xi'(k) \right.}_{}
$$
$$
\qquad \underbrace{\left. - \frac{1}{m}\mathbf{vec}\left(\nabla_{\mathbf{W}_{\ell'}}\left[f_{\mathbf{W}(k)}(x)\right]\right)^\top \nabla_{\mathbf{W}_{\ell'}}\left[f_{\mathbf{W}(0)}(\mathbf{X})\right] G_k\mathbf{u}(0) \right]}_{=\Delta_{13}} \tag{41}
$$

Our goal in this step is to control $\|\Delta_{12}\|_2$ and $\|\Delta_{13}\|_2$. Then, in the third step, we will show $\|\Delta_{11}\|_2$ is the term, which governs the behavior of the prediction risk with respect to algorithm iteration $k$.

First, we bound the $\ell_2$ norm of the first term in $\Delta_{12}$ as:

$$\left\| \left[ \frac{1}{m} \sum_{\ell=1}^{L} \mathbf{vec}\big(\nabla_{\mathbf{W}_\ell}\big[f_{\mathbf{W}(k)}(x)\big]\big)^\top \nabla_{\mathbf{W}_\ell}\big[f_{\mathbf{W}(0)}(\mathbf{X})\big] - \mathbf{Ker}(x,\mathbf{X}) \right] G_k \mathbf{y} \right\|_2$$

$$\leq \frac{1}{mL} \sum_{\ell=1}^{L} \underbrace{\left\| \mathbf{vec}\big(\nabla_{\mathbf{W}_\ell}\big[f_{\mathbf{W}(k)}(x)\big]\big) - \mathbf{vec}\big(\nabla_{\mathbf{W}_\ell}\big[f_{\mathbf{W}(0)}(x)\big]\big) \right\|_2}_{\leq \mathcal{O}\big(\tau^{1/3}L^2\sqrt{\omega m \log(m)}\big)\,:\,\textbf{Lemma 4.6}} \underbrace{\left\| \nabla_{\mathbf{W}_\ell}\big[f_{\mathbf{W}(0)}(\mathbf{X})\big] \right\|_F}_{\leq \mathcal{O}\big(\sqrt{\omega n m}\big)\,:\,\textbf{Lemma 4.4}} \|G_k\mathbf{y}\|_2$$

$$+ \frac{1}{L} \sqrt{ \sum_{i=1}^{n} \left( \frac{1}{m} \sum_{\ell=1}^{L} \big\langle \nabla_{\mathbf{W}_\ell}\big[f_{\mathbf{W}(0)}(x)\big], \nabla_{\mathbf{W}_\ell}\big[f_{\mathbf{W}(0)}(\mathbf{x}_i)\big] \big\rangle_{\mathrm{Tr}} - \mathbf{Ker}(x,\mathbf{x}_i) \right)^2 } \|G_k\mathbf{y}\|_2$$

$$\leq \left\{ \mathcal{O}\left( \frac{n^{5/6}L^2\omega^{7/6}}{m^{1/6}\delta^{1/6}\lambda_\infty^{1/3}} \sqrt{\log(m)} \right) + \mathcal{O}\left( \omega n^{1/2} L^{3/2} \sqrt[4]{\frac{\log(nL/\delta)}{m}} \right) \right\} \|G_k\|_2 \|\mathbf{y}\|_2$$

$$\leq \mathcal{O}\left( \frac{n^{5/6}L^2\omega^{7/6}}{m^{1/6}\delta^{1/6}\lambda_\infty^{4/3}} \sqrt{\log(m)} \cdot \|\mathbf{y}\|_2 \right) + \mathcal{O}\left( \frac{\omega n^{1/2} L^{3/2}}{\lambda_\infty} \sqrt[4]{\frac{\log(nL/\delta)}{m}} \cdot \|\mathbf{y}\|_2 \right), \qquad (42)$$

where, in the second inequality, we plugged $\tau = \mathcal{O}\big(\frac{n\sqrt{\omega}}{\sqrt{m}\delta\lambda_\infty}\big)$ in the result of Lemma 4.6 and used Lemma 4.11. In the last inequality, we used $\|G_k\|_2 \leq \mathcal{O}\big(\frac{1}{\lambda_\infty}\big)$. Similarly, we can control the $\ell_2$ norm of the second term in $\Delta_{12}$ as follows:

$$\left\| \frac{1}{m} \sum_{\ell:\ell \neq \ell'} \mathbf{vec}\big(\nabla_{\mathbf{W}_\ell}\big[f_{\mathbf{W}(k)}(x)\big]\big)^\top \nabla_{\mathbf{W}_\ell}\big[f_{\mathbf{W}(0)}(\mathbf{X})\big] G_k \mathbf{y} \right\|_2$$

$$\leq \frac{1}{m} \sum_{\ell:\ell \neq \ell'} \underbrace{\big\| \mathbf{vec}\big(\nabla_{\mathbf{W}_\ell}\big[f_{\mathbf{W}(k)}(x)\big]\big) \big\|_2}_{\leq \mathcal{O}\big(\sqrt{\omega m}\big)} \cdot \underbrace{\big\| \nabla_{\mathbf{W}_\ell}\big[f_{\mathbf{W}(0)}(\mathbf{X})\big] \big\|_F}_{\leq \mathcal{O}\big(\sqrt{\omega m n}\big)} \cdot \underbrace{\|G_k\|_2}_{\leq \mathcal{O}\big(\frac{1}{\lambda_\infty}\big)} \|\mathbf{y}\|_2$$

$$\leq \mathcal{O}\left( \frac{\omega L\sqrt{n}}{\lambda_\infty} \right) \cdot \|\mathbf{y}\|_2. \qquad (43)$$

We turn our attention to controlling $\|\Delta_{13}\|_2$. The first term in $\Delta_{13}$; Recall that $\big\| \mathbf{vec}\big(\nabla_{\mathbf{W}_{\ell'}}\big[f_{\mathbf{W}(k)}(x)\big]\big) \big\|_2 \leq \mathcal{O}(\sqrt{m\omega})$ by Lemma 4.4. Then, the random variable $\mathbf{vec}\big(\nabla_{\mathbf{W}_\ell}\big[f_{\mathbf{W}(k)}(x)\big]\big)^\top \mathbf{vec}\big(\mathbf{W}_\ell^{(0)}\big)$ is simply a $\mathcal{N}\big(0, \mathcal{O}(\omega)\big)$ for $1 \leq \ell \leq L$. A straightforward application of Chernoff bound for normal random variable and taking union bound over the layer $1 \leq \ell \leq L$ yield that: with probability at least $1 - \delta$,

$$\left| \mathbf{vec}\big(\nabla_{\mathbf{W}_{\ell'}}\big[f_{\mathbf{W}(k)}(x)\big]\big)^\top \mathbf{vec}\big(\mathbf{W}_{\ell'}^{(0)}\big) \right| \leq \mathcal{O}\left( \sqrt{\omega \log\left(\frac{L}{\delta}\right)} \right). \qquad (44)$$

The $\ell_2$ norm of the third term in $\Delta_{13}$ can be bounded as follows:

$$\left\| \frac{1}{m} \mathbf{vec}\big(\nabla_{\mathbf{W}_{\ell'}}\big[f_{\mathbf{W}(k)}(x)\big]\big)^\top \nabla_{\mathbf{W}_{\ell'}}\big[f_{\mathbf{W}(0)}(\mathbf{X})\big] G_k \mathbf{u}(0) \right\|_2$$

$$\leq \frac{1}{m} \underbrace{\big\| \mathbf{vec}\big(\nabla_{\mathbf{W}_{\ell'}}\big[f_{\mathbf{W}(k)}(x)\big]\big) \big\|_2}_{\leq \mathcal{O}\big(\sqrt{m\omega}\big)} \underbrace{\big\| \nabla_{\mathbf{W}_{\ell'}}\big[f_{\mathbf{W}(0)}(\mathbf{X})\big] \big\|_F}_{\leq \mathcal{O}\big(\sqrt{\omega m n}\big)} \underbrace{\|G_k \mathbf{u}(0)\|_2}_{\leq \mathcal{O}\big(\frac{\sqrt{n\omega}}{\lambda_\infty\delta}\big)} \leq \mathcal{O}\left( \frac{n\omega^{3/2}}{\lambda_\infty\delta} \right). \qquad (45)$$

In the last inequality, we used the Lemma 4.8 and $\|G_k\|_2 \leq \mathcal{O}\left(\frac{1}{\lambda_\infty}\right)$. By combining (39), (40), (44), (45) with $\left\|\nabla_{\mathbf{W}_{\ell'}}[f_{\mathbf{W}(0)}(x)]\right\|_F \leq \mathcal{O}(\sqrt{m\omega})$, we have a following :

$$
\begin{aligned}
\|\Delta_{13}\|_2 &\leq \left\|\left\langle \mathbf{vec}(\nabla_{\mathbf{W}_{\ell'}}[f_{\mathbf{W}(k)}(x)]), \mathbf{vec}(\mathbf{W}_{\ell'}^{(0)})\right\rangle\right\|_2 + \left\|(\mathbf{vec}(\nabla_{\mathbf{W}_{\ell'}}[f_{\mathbf{W}(0)}(x)]))^\top \xi'(k)\right\|_2 \\
&\quad + \left\|\frac{1}{m}(\mathbf{vec}(\nabla_{\mathbf{W}_{\ell'}}[f_{\mathbf{W}(0)}(x)])^\top \nabla_{\mathbf{W}_{\ell'}}[f_{\mathbf{W}(0)}(\mathbf{X})]G_k\mathbf{u}(0)\right\|_2 \\
&\leq \mathcal{O}\left(\sqrt{\omega\log\left(\frac{L}{\delta}\right)}\right) + \mathcal{O}\left(\frac{n^{11/6}L^3\omega^{13/6}\|\mathbf{y}-\mathbf{u}(0)\|_2}{m^{1/6}\lambda_\infty^{4/3}\delta^{1/6}}\sqrt{\log(m)}\right) \\
&\quad + \mathcal{O}\left(\frac{n^{5/6}L^2\omega^{7/6}\|\mathbf{y}-\mathbf{u}(0)\|_2}{m^{1/6}\lambda_\infty^{7/3}\delta^{1/6}}\sqrt{\log(m)}\right) + \mathcal{O}\left(\frac{n\omega^{3/2}}{\lambda_\infty\delta}\right) \\
&= \mathcal{O}\left(\frac{n^{11/6}L^3\omega^{13/6}\|\mathbf{y}-\mathbf{u}(0)\|_2}{m^{1/6}\lambda_\infty^{4/3}\delta^{1/6}}\sqrt{\log(m)}\right) + \mathcal{O}\left(\frac{n^{5/6}L^2\omega^{7/6}\|\mathbf{y}-\mathbf{u}(0)\|_2}{m^{1/6}\lambda_\infty^{7/3}\delta^{1/6}}\sqrt{\log(m)}\right) \\
&\quad + \mathcal{O}\left(\frac{n\omega^{3/2}}{\lambda_\infty\delta}\right).
\end{aligned}
\tag{46}
$$

***Step 2. Control on $\Delta_2$.*** First, note that there is a recent finding that the reproducing kernel Hilbert spaces of NTKs with any number of layers (i.e., $L \geq 1$) have the same set of functions, if kernels are defined on $\mathcal{S}^{d-1}$. See Chen & Xu [2020]. Along with this result, we can apply the proof used in Lemma.D.2. in Hu et al. [2021] for proving a following :

$$
\|\Delta_2\|_2 = \mathcal{O}_{\mathbb{P}}\left(\frac{1}{\sqrt{n}}\right).
\tag{47}
$$

***Step 3. The behavior of $L_2$ risk is characterized by the term $\Delta_{11}$.*** Recall the decompositions (37) and (41), then we have:

$$
\widehat{f}_{\mathbf{W}^{(k)}}(x) - f^*(x) = \Delta_{11} + (\Delta_{12} + \Delta_{13} + \Delta_2) := \Delta_{11} + \Theta.
\tag{48}
$$

Our goal in this step is mainly two-folded: (i) Control $\mathbb{E}_\varepsilon\|\Theta\|_2^2$ arbitrarily small with proper choices of step-size of GD $\eta$ and width of the network $m$. (ii) Show that how $\mathbb{E}_\varepsilon\|\Delta_{11}\|_2^2$ affect the behavior of prediction risk over the GD iterations $k$. First, note that we have

$$
\mathbb{E}_\varepsilon\|\mathbf{y}\|_2^2 = \mathbb{E}_\varepsilon\|\mathbf{y}^* + \varepsilon\|_2^2 \leq 2(\mathbf{y}^*)^\top\mathbf{y}^* + 2\mathbb{E}_\varepsilon\|\varepsilon\|_2^2 = \mathcal{O}(n).
\tag{49}
$$

Second, recall Lemma 4.9 and note that over the random initialization, with probability at least $1-\delta$,

$$
\mathbb{E}_\varepsilon\|\mathbf{y} - \mathbf{u}(0)\|_2^2 \leq \mathcal{O}\left(\frac{n}{\delta}\right).
\tag{50}
$$

Now, by combining the bounds (42), (46) and (47), we have

$$
\begin{aligned}
\mathbb{E}_\varepsilon\|\Theta\|_2^2 &\leq 3\mathbb{E}_\varepsilon\left(\|\Delta_{12}\|_2^2 + \|\Delta_{13}\|_2^2 + \|\Delta_2\|_2^2\right) \\
&\leq \mathbb{E}_\varepsilon\left[\mathcal{O}\left(\frac{n^{5/3}L^4\omega^{7/3}}{m^{1/3}\lambda_\infty^{8/3}\delta^{1/3}}\log(m)\cdot\|\mathbf{y}\|_2^2\right) + \mathcal{O}\left(\frac{\omega^2nL^3}{\lambda_\infty^2}\sqrt{\frac{\log(nL/\delta)}{m}}\cdot\|\mathbf{y}\|_2^2\right)\right. \\
&\quad\left. + \mathcal{O}\left(\frac{n^{11/3}L^6\omega^{13/3}\|\mathbf{y}-\mathbf{u}(0)\|_2^2}{m^{1/3}\lambda_\infty^{8/3}\delta^{1/3}}\log(m)\right) + \mathcal{O}\left(\frac{n^{5/3}L^4\omega^{7/3}\|\mathbf{y}-\mathbf{u}(0)\|_2^2}{m^{1/3}\lambda_\infty^{14/3}\delta^{1/3}}\log(m)\right)\right] \\
&\quad + \mathcal{O}\left(\frac{n^2\omega^3}{\lambda_\infty^2\delta^2}\right) + \mathcal{O}\left(\frac{1}{n}\right) \\
&\leq \mathcal{O}\left(\frac{\omega^2n^2L^3}{\lambda_\infty^2}\sqrt{\frac{\log(nL/\delta)}{m}}\right) + \mathcal{O}\left(\frac{n^{14/3}L^6\omega^{13/3}}{m^{1/3}\lambda_\infty^{8/3}\delta^{4/3}}\log(m)\right) + \mathcal{O}\left(\frac{n^{8/3}L^4\omega^{7/3}}{m^{1/3}\lambda_\infty^{14/3}\delta^{4/3}}\log(m)\right) \\
&\quad + \mathcal{O}\left(\frac{n^2\omega^3}{\lambda_\infty^2\delta^2}\right) + \mathcal{O}\left(\frac{1}{n}\right),
\end{aligned}
\tag{51}
$$

where in the third inequality, we used (49) and (50).

### *Case 1.* **When $k$ is large, the $L_2$ risk is bounded away from zero by some constant.**

Now we control $\mathbb{E}_\varepsilon \|\Delta_{11}\|_2^2$. Recall the definitions $\|f\|_2^2 := \int_{\mathbf{x} \in \mathcal{S}^{d-1}} |f(\mathbf{x})|^2 d\mathbf{x}$ and $G_k = \sum_{j=0}^{k-1} \eta m (\mathbf{I} - \eta m \mathbf{H}_L^\infty)^j$. Let us denote $\mathbf{S} = \mathbf{y}^* \mathbf{y}^{*\top}$. Then, we have

$$
\mathbb{E}_\varepsilon \|\Delta_{11}\|_2^2 = \int_{x \in \mathcal{S}^{d-1}} \mathbf{Ker}(x, \mathbf{X}) \left[ \left( G_k - \left( \mathbf{H}_L^\infty \right)^{-1} \right) \mathbf{y}^* \mathbf{y}^{*\top} \left( G_k - \left( \mathbf{H}_L^\infty \right)^{-1} \right) + G_k^2 \right] \mathbf{Ker}(\mathbf{X}, x) dx
$$

$$
= \int_{x \in \mathcal{S}^{d-1}} \mathbf{Ker}(x, \mathbf{X}) \left( \mathbf{H}_L^\infty \right)^{-1} M_k \left( \mathbf{H}_L^\infty \right)^{-1} \mathbf{Ker}(\mathbf{X}, x) dx
$$

where

$$
M_k = \left( \mathbf{I} - \eta m \mathbf{H}_L^\infty \right)^k \mathbf{S} \left( \mathbf{I} - \eta m \mathbf{H}_L^\infty \right)^k + \left( \mathbf{I} - \left( \mathbf{I} - \eta m \mathbf{H}_L^\infty \right)^k \right)^2
$$

$$
= \left[ \left( \mathbf{I} - \eta m \mathbf{H}_L^\infty \right)^k - \left( \mathbf{S} + \mathcal{I} \right)^{-1} \right] \left( \mathbf{S} + \mathcal{I} \right) \left[ \left( \mathbf{I} - \eta m \mathbf{H}_L^\infty \right)^k - \left( \mathbf{S} + \mathcal{I} \right)^{-1} \right] + \mathcal{I} - \left( \mathbf{S} + \mathcal{I} \right)^{-1}.
$$

For the algorithm iterations $k \geq \left( \frac{\log(n)}{\eta m \lambda_\infty} \right) C_0$ with some constant $C_0 > 1$, we have

$$
\left( \mathcal{I} - \eta m \mathbf{H}_L^\infty \right)^k \preceq \left( 1 - \eta m \lambda_\infty \right)^k \cdot \mathcal{I} \preceq \exp(-\eta m \lambda_\infty k) \cdot \mathcal{I} \preceq \exp(-C_0 \log(n)) = \frac{1}{n^{C_0}} \cdot \mathcal{I}.
$$

Since $1 + \|\mathbf{y}\|_2^2 \leq C_1 n$ for some constant $C_1$, we have

$$
\lambda_{\max} \left( \frac{1}{n^{C_0}} \cdot \left( \mathbf{S} + \mathcal{I} \right) \right) = \frac{1 + \|\mathbf{y}\|_2^2}{n^{C_0}} \leq \frac{C_1}{n^{C_0 - 1}} < 1. \tag{52}
$$

Using the first item of Lemma (4.12) with the inequality (52), we have

$$
\left( \mathcal{I} - \eta m \mathbf{H}_L^\infty \right)^k \preceq \frac{1}{n^{C_0}} \cdot \mathcal{I} \prec \left( \mathbf{S} + \mathcal{I} \right)^{-1}. \tag{53}
$$

The above inequality (53) lead to a following result :

$$
\left( \mathbf{S} + \mathcal{I} \right)^{-1} - \left( \mathcal{I} - \eta m \mathbf{H}_L^\infty \right)^k \succeq \left( \mathbf{S} + \mathcal{I} \right)^{-1} - \frac{1}{n^{C_0}} \cdot \mathcal{I}. \tag{54}
$$

It is obvious that both $\left( \mathbf{S} + \mathcal{I} \right)^{-1} - \left( \mathcal{I} - \eta m \mathbf{H}_L^\infty \right)^k$ and $\left( \mathbf{S} + \mathcal{I} \right)^{-1} - \frac{1}{n^{C_0}} \cdot \mathcal{I}$ are positive definite matrices due to (54), and it is also easy to see that they are exchangeable. By using the second item of Lemma (4.12), we have

$$
M_k = \left[ \left( \mathbf{I} - \eta m \mathbf{H}_L^\infty \right)^k - \left( \mathbf{S} + \mathcal{I} \right)^{-1} \right] \left( \mathbf{S} + \mathcal{I} \right) \left[ \left( \mathbf{I} - \eta m \mathbf{H}_L^\infty \right)^k - \left( \mathbf{S} + \mathcal{I} \right)^{-1} \right] + \mathcal{I} - \left( \mathbf{S} + \mathcal{I} \right)^{-1}
$$

$$
\succeq \left[ \left( \mathbf{S} + \mathcal{I} \right)^{-1} - \frac{1}{n^{C_0}} \cdot \mathcal{I} \right] \left( \mathbf{S} + \mathcal{I} \right) \left[ \left( \mathbf{S} + \mathcal{I} \right)^{-1} - \frac{1}{n^{C_0}} \cdot \mathcal{I} \right] + \mathcal{I} - \left( \mathbf{S} + \mathcal{I} \right)^{-1}
$$

$$
= \frac{1}{n^{2C_0}} \mathbf{S} + \left( 1 - \frac{1}{n^{C_0}} \right)^2 \cdot \mathcal{I}.
$$

Then, we have

$$
\mathbb{E}_\varepsilon \|\Delta_{11}\|_2^2 \succeq \frac{1}{n^{2C_0}} \mathcal{A} + \left( 1 - \frac{1}{n^{C_0}} \right)^2 \mathcal{B} \succeq c_0 \mathcal{B},
$$

where $c_0 \in (0, 1)$ is a constant and

$$
\mathcal{A} = \int_{x \in \mathcal{S}^{d-1}} \left[ \mathbf{Ker}(x, \mathbf{X}) \left( \mathbf{H}_L^\infty \right)^{-1} \mathbf{y}^* \right]^2 dx, \quad \text{and} \quad \mathcal{B} = \int_{x \in \mathcal{S}^{d-1}} \left[ \mathbf{Ker}(x, \mathbf{X}) \left( \mathbf{H}_L^\infty \right)^{-1} \right]^2 dx. \tag{55}
$$

By triangle inequality with the decomposition (48) and the bound on $\mathbb{E}_\varepsilon \|\Theta\|_2^2$ in (51), we have:

$$
\mathbb{E}_\varepsilon \left\| \widehat{f}_{\mathbf{W}^{(k)}} - f^* \right\|_2^2 = \mathbb{E}_\varepsilon \|\Delta_{11} + \Theta\|_2^2
$$

$$
\geq \frac{1}{2} \mathbb{E}_\varepsilon \|\Delta_{11}\|_2 - \mathbb{E}_\varepsilon \|\Theta\|_2^2
$$

$$
\geq \frac{c_0}{2} \mathcal{B} - \mathcal{O} \left( \frac{1}{n} \right) - \mathcal{O} \left( \frac{n^2 \omega^3}{\lambda_\infty^2 \delta^2} \right) - \tilde{\mathcal{O}} \left( \frac{1}{m^{1/3}} \mathrm{poly} \left( \omega, n, L, \frac{1}{\lambda_\infty}, \frac{1}{\delta} \right) \right). \tag{56}
$$

For the third term in (56), we can choose $\omega \leq C_2 \left( \frac{\lambda_\infty \delta}{n} \right)^{2/3}$ for some constant $C_2 > 0$ such that the term can be bounded by $\frac{c_0}{8} \left\| \mathbf{Ker}(\cdot, \mathbf{X})(\mathbf{H}_L^\infty)^{-1} \right\|_2^2$. Similarly, the width $m$ can be chosen large enough such that the fourth term in (56) is upper-bounded by $\frac{c_0}{8} \| \mathbf{Ker}(\cdot, \mathbf{X})(\mathbf{H}_L^\infty)^{-1} \|_2^2$. Using the above choices of $k$, $\omega$, and $m$, we can further bound (56):

$$\mathbb{E}_\varepsilon \left\| f_{\mathbf{W}(k)} - f^* \right\|_2^2 \geq \frac{c_0}{4} \left\| \mathbf{Ker}(\cdot, \mathbf{X})(\mathbf{H}_L^\infty)^{-1} \right\|_2^2 - \mathcal{O}\left( \frac{1}{n} \right). \tag{57}$$

Note that $\mathbb{E}_\varepsilon \| \widehat{f}_\infty - g^* \|_2^2 = \| \mathbf{Ker}(\cdot, \mathbf{X})(\mathbf{H}_L^\infty)^{-1} \|_2^2$ where $g^* := 0$ and $\widehat{f}_\infty$ denotes the noise interpolator. Then, by Theorem 4.2. of Hu et al. [2021], we know that $\mathbb{E}_\varepsilon \| \widehat{f}_\infty - g^* \|_2^2 \geq c_1$ for some constant $c_1 > 0$. Then, we can take $n$ large enough such that the term $\mathcal{O}\left( \frac{1}{n} \right)$ is upper-bounded by $\frac{c_0 c_1}{8}$, and finish the proof.

### *Case 2. When $k$ is small, the $L_2$ risk is bounded away from zero by some constant.*

Recall the definition of $\Delta_{11}$ in the decomposition (41),

$$\begin{aligned} \Delta_{11} &:= \mathbf{Ker}(x, \mathbf{X}) G_k \left[ \mathbf{y}^* + \varepsilon \right] - \mathbf{Ker}(x, \mathbf{X}) \mathbf{H}_L^\infty \mathbf{y}^* \\ &:= \Delta_{11}^* - \mathbf{Ker}(x, \mathbf{X}) \mathbf{H}_L^\infty \mathbf{y}^*. \end{aligned} \tag{58}$$

We denote the eigen-decomposition of the matrix $\mathbf{H}_L^\infty := \sum_{i=1}^n \lambda_i \mathbf{v_i v_i}^\top$, then we can easily see a following:

$$G_k = \eta m \sum_{j=0}^{k-1} \left( \sum_{i=1}^n (1 - \eta m \lambda_i)^j \mathbf{v_i v_i}^\top \right) \preceq \eta m \sum_{j=0}^{k-1} \sum_{i=1}^n \mathbf{v_i v_i}^\top \preceq \eta m k \cdot \mathcal{I}.$$

By using the above inequality, we have

$$\begin{aligned} \mathbb{E}_\varepsilon \left\| \Delta_{11}^* \right\|_2^2 &= \int_{x \in \mathcal{S}^{d-1}} \mathbf{Ker}(x, \mathbf{X}) G_k (\mathbf{S} + \mathcal{I}) G_k \mathbf{Ker}(\mathbf{X}, x) dx \\ &\leq \eta^2 m^2 k^2 \left( \int_{x \in \mathcal{S}^{d-1}} \left[ \mathbf{Ker}(x, \mathbf{X}) \mathbf{y}^* \right]^2 dx + \| \mathbf{Ker}(\cdot, \mathbf{X}) \|_2^2 \right) = \mathcal{O}\left( \eta^2 m^2 k^2 \omega^2 n^2 L^2 \right). \end{aligned}$$

Recall the decompositions (37) and (41), then we have:

$$\begin{aligned} \mathbb{E}_\varepsilon \left\| \widehat{f}_{\mathbf{W}^{(k)}} - f^* \right\|_2^2 &= \mathbb{E}_\varepsilon \left\| \Delta_{11}^* + \Theta - \mathbf{Ker}(\cdot, \mathbf{X}) \mathbf{H}_L^\infty \mathbf{y}^* \right\|_2^2 \\ &\geq \frac{1}{2} \| \mathbf{Ker}(\cdot, \mathbf{X}) \mathbf{H}_L^\infty \mathbf{y}^* \|_2^2 - \mathbb{E}_\varepsilon \left\| \Delta_{11}^* + \Theta \right\|_2^2 \\ &\geq \frac{1}{2} \| \mathbf{Ker}(\cdot, \mathbf{X}) \mathbf{H}_L^\infty \mathbf{y}^* \|_2^2 - 2\mathbb{E}_\varepsilon \left\| \Delta_{11}^* \right\|_2^2 - 2\mathbb{E}_\varepsilon \left\| \Theta \right\|_2^2 \\ &\geq \frac{1}{2} \| \mathbf{Ker}(\cdot, \mathbf{X}) \mathbf{H}_L^\infty \mathbf{y}^* \|_2^2 - \mathcal{O}\left( \eta^2 m^2 k^2 \omega^2 n^2 L^2 \right) - \mathcal{O}\left( \frac{1}{n} \right) \\ &\quad - \mathcal{O}\left( \frac{n^2 \omega^3}{\lambda_\infty^2 \delta^2} \right) - \tilde{\mathcal{O}}\left( \frac{1}{m^{1/3}} \text{poly}\left( \omega, n, L, \frac{1}{\lambda_\infty}, \frac{1}{\delta} \right) \right). \end{aligned} \tag{59}$$

For some constant $C_1' > 0$, let $k \leq C_1' \left( \frac{1}{\eta m n \omega L} \right)$ such that the second term in the bound (59) can be bounded by $\frac{1}{8} \| \mathbf{Ker}(\cdot, \mathbf{X})(\mathbf{H}_L^\infty)^{-1} \mathbf{y}^* \|_2^2$. For the fourth term in (59), we can choose $\omega \leq C_2' \left( \frac{\lambda_\infty \delta}{n} \right)^{2/3}$ for some constant $C_2' > 0$ such that the term can be bounded by $\frac{1}{8} \left\| \mathbf{Ker}(\cdot, \mathbf{X})(\mathbf{H}_L^\infty)^{-1} \mathbf{y}^* \right\|_2^2$. Similarly, the width $m$ can be chosen large enough such that the fifth term in (59) is upper-bounded by $\frac{1}{8} \| \mathbf{Ker}(\cdot, \mathbf{X})(\mathbf{H}_L^\infty)^{-1} \mathbf{y}^* \|_2^2$. Using the above choices of $k$, $\omega$, and $m$, we can further bound (59):

$$\mathbb{E}_\varepsilon \left\| f_{\mathbf{W}(k)} - f^* \right\|_2^2 \geq \frac{1}{4} \left\| \mathbf{Ker}(\cdot, \mathbf{X})(\mathbf{H}_L^\infty)^{-1} \mathbf{y}^* \right\|_2^2 - \mathcal{O}\left( \frac{1}{n} \right) \geq C_3' \| f_\rho^\star \|_2^2 - \mathcal{O}\left( \frac{1}{n} \right). \tag{60}$$

In the second inequality, we used (47) with triangle inequality. In the third inequality, we can take $n$ large enough such that the term $\mathcal{O}\left( \frac{1}{n} \right)$ is upper-bounded by $\frac{C_3'}{2} \left\| f_\rho^\star \right\|_2^2$. Lastly, by using the assumption that $f_\rho^\star$ is a square-integrable function, we finish the proof.

## G    PROOF OF THEOREM 3.10-TRAINING ERROR

For the convenience of notation, we denote $u_{D,i}(k) = f_{\mathbf{W}_D^{(k)}}(\mathbf{x}_i)$ and let $\mathbf{u}_D(k) = \left[u_{1,D}(k), \ldots, u_{n,D}(k)\right]^\top$. In order to analyze the training error of $\ell_2$-regularized estimator, $\|\mathbf{u}_D(k) - y\|_2^2$, we decompose the term as follows:

$$\|\mathbf{u}_D(k+1) - y\|_2^2 = \|\mathbf{u}_D(k+1) - (1 - \eta_2\mu L)\mathbf{u}_D(k)\|_2^2 + \|(1 - \eta_2\mu L)\mathbf{u}_D(k) - y\|_2^2$$
$$- 2\big(y - (1 - \eta_2\mu L)\mathbf{u}_D(k)\big)^\top\big(\mathbf{u}_D(k+1) - (1 - \eta_2\mu L)\mathbf{u}_D(k)\big) \tag{61}$$

Equipped with this decomposition, the proof consists of the following steps:

1. We decompose the decayed prediction difference $\mathbf{u}_D(k+1) - (1 - \eta_2\mu L)\mathbf{u}_D(k)$ into two terms. We note that the first term is related with a gram matrix $\mathbf{H}_D(k)$ and denote a second term as $\mathbf{I}_D^{(k)}$.

2. The term $\mathbf{I}_D^{(k)}$ can be further decomposed into three terms, where we denote them as $\mathbf{I}_{2,D}^{(k)}$, $\mathbf{I}_{3,D}^{(k)}$ and $\mathbf{I}_{5,D}^{(k)}$. The crux for controlling the $\ell_2$-norm of the above three terms is to utilize the results from the Appendix $A.4$. The applications of Lemmas in the Appendix $A.4$ is possible, since we can inductively guarantee that $\|W_{D,\ell}^{(k)} - W_{D,\ell}^{(0)}\|_2$ is sufficiently small enough for large enough $m$.

3. Given the decomposition (61), we further decompose it into four terms as follows:

$$(61) = \underbrace{\|(1 - \eta_2\mu L)\mathbf{u}_D(k) - y\|_2^2}_{:=\mathbf{T}_1} + \underbrace{\|\mathbf{u}_D(k+1) - (1 - \eta_2\mu L)\mathbf{u}_D(k)\|_2^2}_{:=\mathbf{T}_2}$$
$$+ \underbrace{2m\eta_1\big(y - (1 - \eta_2\mu L)\mathbf{u}_D(k)\big)^\top\mathbf{H}_D(k)\big(\mathbf{u}_D(k) - y\big)}_{:=\mathbf{T}_3}$$
$$\underbrace{-2\big(y - (1 - \eta_2\mu L)\mathbf{u}_D(k)\big)^\top\mathbf{I}_D^{(k)}}_{:=\mathbf{T}_4}. \tag{62}$$

   In this step, we obtain the upper-bound of $\|\mathbf{T}_i\|_2$ for $i = 1, 2, 3, 4$ obtained in Step 4.

4. We combine the upper-bounds of $\|\mathbf{T}_i\|_2$ for $i = 1, 2, 3, 4$ in step 3 and obtain the bound on $\|\mathbf{u}_D(k+1) - y\|_2^2$ with respect to $\|\mathbf{u}_D(k) - y\|_2^2$ and $\|y\|_2$.

5. Lastly, we inductively show that the weights generated from regularized gradient descent stay within a perturbation region $\mathcal{B}(\mathcal{W}^{(0)}, \tau)$, irrespective with the number of iterations of algorithm.

We start the proof by analyzing the term $\mathbf{u}(k+1) - (1 - \eta_2\mu L)\mathbf{u}(k)$.

**_Step 1. Dynamics of_** $\mathbf{u_D}(\mathbf{k}+1) - (1 - \eta_2\mu L)\cdot\mathbf{u_D}(\mathbf{k})$. Recall $\big(\mathbf{\Sigma}_{D,\ell,i}^{(k)}\big)_{jj} = \mathbb{1}\big(\langle\mathbf{w}_{D,\ell,j}^{(k)}, \mathbf{x}_{D,\ell-1,i}^{(k)}\rangle \geq 0\big)$ and we introduce a diagonal matrix $\widetilde{\mathbf{\Sigma}}_{D,\ell,i}^{(k)}$, whose $j$th entry is defined as follows:

$$\big(\widetilde{\mathbf{\Sigma}}_{D,\ell,i}^{(k)}\big)_{jj} = \big(\mathbf{\Sigma}_{D,\ell,i}^{(k+1)} - \mathbf{\Sigma}_{D,\ell,i}^{(k)}\big)_{jj} \cdot \frac{\langle\mathbf{w}_{D,\ell,j}^{(k+1)}, \mathbf{x}_{D,\ell-1,i}^{(k+1)}\rangle}{\langle\mathbf{w}_{D,\ell,j}^{(k+1)}, \mathbf{x}_{D,\ell-1,i}^{(k+1)}\rangle - \langle\mathbf{w}_{D,\ell,j}^{(k)}, \mathbf{x}_{D,\ell-1,i}^{(k)}\rangle}.$$

With this notation, the difference $\mathbf{x}_{D,L,i}^{(k+1)} - \mathbf{x}_{D,L,i}^{(k)}$ can be rewritten via the recursive applications of $\widetilde{\mathbf{\Sigma}}_{D,\ell,i}^{(k)}$: Then, we introduce following notations :

$$\mathbf{D}_{D,\ell,i}^{(k)} = \bigg(\prod_{r=\ell+1}^{L}\mathbf{\Sigma}_{D,r,i}^{(k)}\mathbf{W}_{D,r}^{(k)}\bigg)\mathbf{\Sigma}_{D,\ell,i}^{(k)}, \qquad \widetilde{\mathbf{D}}_{D,\ell,i}^{(k)} = \bigg(\prod_{r=\ell+1}^{L}\big(\mathbf{\Sigma}_{D,r,i}^{(k)} + \widetilde{\mathbf{\Sigma}}_{D,r,i}^{(k)}\big)\mathbf{W}_{D,r}^{(k+1)}\bigg)\big(\mathbf{\Sigma}_{D,\ell,i}^{(k)} + \widetilde{\mathbf{\Sigma}}_{D,\ell,i}^{(k)}\big).$$

Now, we can write $u_{D,i}(k+1) - u_{D,i}(k)$ by noting that $u_{D,i}(k) = \sqrt{m} \cdot \mathbf{v}^{\mathrm{T}} \mathbf{x}_{D,L,i}^{(k)}$:

$$
\begin{aligned}
u_{D,i}&(k+1) - u_{D,i}(k) \\
&= \sqrt{m} \cdot \mathbf{v}^{\mathrm{T}} \big( \mathbf{x}_{D,L,i}^{(k+1)} - \mathbf{x}_{D,L,i}^{(k)} \big) \\
&= \sqrt{m} \cdot \mathbf{v}^{\mathrm{T}} \sum_{\ell=1}^{L} \widetilde{\mathbf{D}}_{D,\ell,i}^{(k)} \Big( \mathbf{W}_{D,\ell}^{(k+1)} - \mathbf{W}_{D,\ell}^{(k)} \Big) \mathbf{x}_{D,\ell-1,i}^{(k)} \\
&= \sqrt{m} \cdot \mathbf{v}^{\mathrm{T}} \sum_{\ell=1}^{L} \widetilde{\mathbf{D}}_{D,\ell,i}^{(k)} \Big( -\eta_1 \nabla_{\mathbf{W}_\ell} \big[ \mathcal{L}_{\mathbf{S}}(\mathbf{W}_D^{(k)}) \big] - \eta_2 \mu \mathbf{W}_{D,\ell}^{(k)} + \eta_2 \mu \mathbf{W}_{D,\ell}^{(0)} \Big) \mathbf{x}_{D,\ell-1,i}^{(k)} \\
&= \underbrace{-\eta_1 \sqrt{m} \cdot \mathbf{v}^{\mathrm{T}} \sum_{\ell=1}^{L} \mathbf{D}_{D,\ell,i}^{(k)} \nabla_{\mathbf{W}_\ell} \big[ \mathcal{L}_{\mathbf{S}}(\mathbf{W}_D^{(k)}) \big] \mathbf{x}_{D,\ell-1,i}^{(k)}}_{\mathbf{I}_{1,D,i}^{(k)}} \\
&\quad \underbrace{-\eta_1 \sqrt{m} \cdot \mathbf{v}^{\mathrm{T}} \sum_{\ell=1}^{L} \Big( \widetilde{\mathbf{D}}_{D,\ell,i}^{(k)} - \mathbf{D}_{D,\ell,i}^{(k)} \Big) \nabla_{\mathbf{W}_\ell} \big[ \mathcal{L}_{\mathbf{S}}(\mathbf{W}_D^{(k)}) \big] \mathbf{x}_{D,\ell-1,i}^{(k)}}_{\mathbf{I}_{2,D,i}^{(k)}} \\
&\quad \underbrace{-\eta_2 \mu \sqrt{m} \cdot \mathbf{v}^{\mathrm{T}} \sum_{\ell=1}^{L} \Big( \widetilde{\mathbf{D}}_{D,\ell,i}^{(k)} - \mathbf{D}_{D,\ell,i}^{(k)} \Big) \Big( \mathbf{W}_{D,\ell}^{(k)} - \mathbf{W}_{D,\ell}^{(0)} \Big) \mathbf{x}_{D,\ell-1,i}^{(k)}}_{\mathbf{I}_{3,D,i}^{(k)}} \\
&\quad \underbrace{-\eta_2 \mu \sqrt{m} \cdot \mathbf{v}^{\mathrm{T}} \sum_{\ell=1}^{L} \mathbf{D}_{D,\ell,i}^{(k)} \mathbf{W}_{D,\ell}^{(k)} \mathbf{x}_{D,\ell-1,i}^{(k)}}_{\mathbf{I}_{4,D,i}^{(k)}} \\
&\quad \underbrace{+\eta_2 \mu \sqrt{m} \cdot \mathbf{v}^{\mathrm{T}} \sum_{\ell=1}^{L} \mathbf{D}_{D,\ell,i}^{(k)} \mathbf{W}_{D,\ell}^{(0)} \mathbf{x}_{D,\ell-1,i}^{(k)}}_{\mathbf{I}_{5,D,i}^{(k)}} \quad (63)
\end{aligned}
$$

where in the second equality, we used the recursive relation (24), and in the third equality, modified GD update rule (6) is applied.

Furthermore, $\mathbf{I}_{1,D,i}^{(k)}$ can be rewritten as follows:

$$
\begin{aligned}
\mathbf{I}_{1,D,i}^{(k)} &= -\eta_1 \sqrt{m} \cdot \mathbf{v}^{\mathrm{T}} \sum_{\ell=1}^{L} \mathbf{D}_{D,\ell,i}^{(k)} \sum_{j=1}^{n} \big( u_{D,j}(k) - y_j \big) \nabla_{\mathbf{W}_\ell} \big[ f_{\mathbf{W}_D^{(k)}}(\mathbf{x}_j) \big] \mathbf{x}_{D,\ell-1,i}^{(k)} \\
&= -\eta_1 \cdot \sum_{j=1}^{n} \big( u_{D,j}(k) - y_j \big) \cdot \Big( \sqrt{m} \sum_{\ell=1}^{L} \mathbf{v}^{\mathrm{T}} \mathbf{D}_{D,\ell,i}^{(k)} \nabla_{\mathbf{W}_\ell} \big[ f_{\mathbf{W}_D^{(k)}}(\mathbf{x}_j) \big] \mathbf{x}_{D,\ell-1,i}^{(k)} \Big) \\
&= -m\eta_1 \cdot \sum_{j=1}^{n} \big( u_{D,j}(k) - y_j \big) \cdot \frac{1}{m} \sum_{\ell=1}^{L} \Big\langle \nabla_{\mathbf{W}_\ell} \big[ f_{\mathbf{W}_D^{(k)}}(\mathbf{x}_i) \big], \nabla_{\mathbf{W}_\ell} \big[ f_{\mathbf{W}_D^{(k)}}(\mathbf{x}_j) \big] \Big\rangle_{\mathrm{Tr}} \\
&= -m\eta_1 \cdot \sum_{j=1}^{n} \big( u_{D,j}(k) - y_j \big) \cdot \mathbf{H}_{D,i,j}(k). \quad (64)
\end{aligned}
$$

With $\mathbf{I}_{4,i}^{(k)} = (-\eta_2 \mu L) \cdot u_{D,i}(k)$ and (64), we can rewrite (63) as follows:

$$
u_{D,i}(k+1) - (1 - \eta_2 \mu L) u_{D,i}(k) = -m\eta_1 \cdot \sum_{j=1}^{n} \big( u_{D,j}(k) - y_j \big) \cdot \mathbf{H}_{D,i,j}(k) + \mathbf{I}_{2,D,i}^{(k)} + \mathbf{I}_{3,D,i}^{(k)} + \mathbf{I}_{5,D,i}^{(k)}.
$$

$$(65)$$

**_Step 2. Control of the size $\left\|\mathbf{I}_D^{(k)}\right\|_2$._**

Let $\mathbf{I}_D^{(k)} = [\mathbf{I}_{2,D,1}^{(k)} + \mathbf{I}_{3,D,1}^{(k)} + \mathbf{I}_{5,D,1}^{(k)}, \ldots, \mathbf{I}_{2,D,n}^{(k)} + \mathbf{I}_{3,D,n}^{(k)} + \mathbf{I}_{5,D,1}^{(k)}]^\top$. Now, we control the bound on the $\left\|\mathbf{I}_D^{(k)}\right\|_2^2$. Recall that in Eq. (27), we have

$$\left\|\mathbf{I}_{2,D}^{(k)}\right\|_2 \leq \mathcal{O}\left(\eta_1 n L^3 \tau^{1/3} \omega m \sqrt{\log(m)}\right) \|\mathbf{u}_D(k) - y\|_2. \tag{66}$$

Similarly, $\left\|\mathbf{I}_{3,D}^{(k)}\right\|_2$ can be bounded:

$$\left\|\mathbf{I}_{3,D}^{(k)}\right\|_2 \leq \sum_{i=1}^n \left|\mathbf{I}_{3,D,i}^{(k)}\right| \leq \eta_2 \mu \sqrt{m} \cdot \sum_{i=1}^n \left[\sum_{\ell=1}^L \underbrace{\left\|\mathbf{v}^\top\left(\widetilde{\mathbf{D}}_{D,\ell,i}^{(k)} - \mathbf{D}_{D,\ell,i}^{(k)}\right)\right\|_2}_{\leq \mathcal{O}\left(L^2 \tau^{1/3}\sqrt{\omega \log(m)}\right)} \cdot \underbrace{\left\|\mathbf{W}_{D,\ell}^{(k)} - \mathbf{W}_{D,\ell}^{(0)}\right\|_2}_{\leq \tau} \cdot \underbrace{\left\|\mathbf{x}_{D,\ell-1,i}^{(k)}\right\|_2}_{\leq \mathcal{O}(1)}\right]$$

$$\leq \mathcal{O}\left(\eta_2 \mu n L^3 \tau^{4/3} \sqrt{\omega m \log(m)}\right). \tag{67}$$

Lastly $\left\|\mathbf{I}_{5,D}^{(k)}\right\|_2$ can be bounded:

$$\left\|\mathbf{I}_{5,D}^{(k)}\right\|_2 \leq \sum_{i=1}^n \left|\mathbf{I}_{5,D,i}^{(k)}\right|$$

$$\leq \sum_{i=1}^n \left|\eta_2 \mu \sqrt{m} \cdot \mathbf{v}^\mathrm{T} \sum_{\ell=1}^L \mathbf{D}_{D,\ell,i}^{(k)} \mathbf{W}_{D,\ell}^{(k)} \mathbf{x}_{D,\ell-1,i}^{(k)}\right| + \sum_{i=1}^n \left|\eta_2 \mu \sqrt{m} \cdot \mathbf{v}^\mathrm{T} \sum_{\ell=1}^L \mathbf{D}_{D,\ell,i}^{(k)}\left(\mathbf{W}_{D,\ell}^{(k)} - \mathbf{W}_{D,\ell}^{(0)}\right)\mathbf{x}_{D,\ell-1,i}^{(k)}\right|$$

$$\leq \eta_2 \mu L \cdot \sum_{i=1}^n |\mathbf{u}_{i,D}(k)| + \eta_2 \mu \sqrt{m} \cdot \sum_{i=1}^n \left[\sum_{\ell=1}^L \underbrace{\|\mathbf{v}\|_2}_{\leq \mathcal{O}(\sqrt{\omega})} \cdot \underbrace{\left\|\mathbf{D}_{D,\ell,i}^{(k)}\right\|_2}_{\leq \mathcal{O}(\sqrt{L})} \cdot \underbrace{\left\|\mathbf{W}_{D,\ell}^{(k)} - \mathbf{W}_{D,\ell}^{(0)}\right\|_2}_{\leq \tau} \cdot \underbrace{\left\|\mathbf{x}_{D,\ell-1,i}^{(k)}\right\|_2}_{\leq \mathcal{O}(1)}\right]$$

$$\leq \mathcal{O}\left(\eta_2 \mu n L \sqrt{\omega \log(L/\delta)}\right) + \mathcal{O}\left(\eta_2 \mu n L^{3/2} \tau \sqrt{m\omega}\right), \tag{68}$$

where in the last inequality, we employed the same logic used in (44) with the Lemma 4.2 to obtain the upper-bound on the $|\mathbf{u}_{i,D}(k)|$. We set the orders of the parameters $\mu$, $\eta_1$, $\eta_2$, $\tau$, and $\omega$ as follows:

$$\mu = \Theta\left(n^{\frac{d-1}{2d-1}}\right), \quad \eta_1 = \Theta\left(\frac{1}{m}n^{-\frac{3d-2}{2d-1}}\right), \quad \eta_2 = \Theta\left(\frac{1}{L}n^{-\frac{3d-2}{2d-1}}\right),$$

$$\tau = \mathcal{O}\left(\frac{L\sqrt{\omega}}{\sqrt{m}\delta}n^{\frac{d}{2d-1}}\right), \quad \omega = \mathcal{O}\left(\frac{1}{L^{3/2}}n^{-\frac{5d-2}{2d-1}}\right). \tag{69}$$

Plugging the choices of parameters (69) with sufficiently large $m$ in (66), (67) and (68) yields

$$\left\|\mathbf{I}_D^{(k)}\right\|_2 \leq \mathcal{O}\left(L^{37/12} n^{-\frac{9d-8}{12d-6}} \frac{\sqrt{\log(m)}}{m^{1/6}\delta^{1/3}}\right) \cdot \|\mathbf{u}_D(k) - y\|_2 + \mathcal{O}_{\mathbb{P}}\left(\frac{1}{n^2}\right). \tag{70}$$

**_Step 3. Upper-bound of $\|\mathbf{T}_i\|_2$ on $i = 1, 2, 3, 4$._**

First, we work on getting the upper-bound on $\lambda_{\max}\left(\mathbf{H}_D(k)\right)$. By the Gershgorin's circle theorem [Varga, 2004], we know the maximum eigenvalue of symmetric positive semi-definite matrix is upper-bounded by the maximum absolute column sum of the matrix. Using this fact, we can

bound the $\lambda_{\max}\big(\mathbf{H}_D(k)\big)$ as :

$$
\begin{aligned}
\lambda_{\max}\big(\mathbf{H}_D(k)\big) &\leq \max_{i=1,\ldots,n}\sum_{j=1}^{n}|\mathbf{H}_{D,i,j}(k)| \\
&\leq \max_{i=1,\ldots,n}\sum_{j=1}^{n}\left|\frac{1}{m}\sum_{\ell=1}^{L}\left\langle\nabla_{\mathbf{W}_\ell}\big[f_{\mathbf{W}_D^{(k)}}(\mathbf{x}_i)\big],\nabla_{\mathbf{W}_\ell}\big[f_{\mathbf{W}_D^{(k)}}(\mathbf{x}_j)\big]\right\rangle_{\mathrm{Tr}}\right| \\
&\leq \max_{i=1,\ldots,n}\sum_{j=1}^{n}\frac{1}{m}\sum_{\ell=1}^{L}\underbrace{\left\|\nabla_{\mathbf{W}_\ell}\big[f_{\mathbf{W}_D^{(k)}}(\mathbf{x}_i)\big]\right\|_F}_{\leq\mathcal{O}(\sqrt{m\omega})}\underbrace{\left\|\nabla_{\mathbf{W}_\ell}\big[f_{\mathbf{W}_D^{(k)}}(\mathbf{x}_j)\big]\right\|_F}_{\leq\mathcal{O}(\sqrt{m\omega})} \\
&\leq \mathcal{O}\big(nL\omega\big).
\end{aligned}
\tag{71}
$$

Recall the decomposition (62). Our goal is to obtain the upper-bound on $\mathbf{T}_i$ for $i = 1, 2, 3, 4$.

***Control on*** $\mathbf{T}_1$***.*** By using the inequality $2\eta_2\mu L(1-\eta_2\mu L)y^\top\big(y-\mathbf{u}_D(k)\big) \leq \eta_2\mu L\,\|y\|_2^2+\eta_2\mu L(1-\eta_2\mu L)^2\,\|y-\mathbf{u}_D(k)\|_2^2$, we have

$$
\begin{aligned}
\|y-(1-\eta_2\mu L)\mathbf{u}_D(k)\|_2^2 &= \big\|(1-\eta_2\mu L)\big(y-\mathbf{u}_D(k)\big)+\eta_2\mu Ly\big\|_2^2 \\
&= (1-\eta_2\mu L)^2\,\|y-\mathbf{u}_D(k)\|_2^2+\eta_2^2\mu^2 L^2\,\|y\|_2^2 \\
&\quad+2\eta_2\mu L(1-\eta_2\mu L)y^\top\big(y-\mathbf{u}_D(k)\big) \\
&\leq (\eta_2\mu L+\eta_2^2\mu^2 L^2)\,\|y\|_2^2+\big(1+\eta_2\mu L\big)\big(1-\eta_2\mu L\big)^2\,\|y-\mathbf{u}_D(k)\|_2^2.
\end{aligned}
\tag{72}
$$

***Control on*** $\mathbf{T}_2$***.*** Recall the equality (65). Then, through applications of the Young's inequality $\|a+b\|_2^2 \leq 2\,\|a\|_2^2+2\,\|b\|_2^2$ for $a, b \in \mathbb{R}^n$, we have

$$
\begin{aligned}
\|\mathbf{u}_D(k+1)-(1-\eta_2\mu L)\mathbf{u}_D(k)\|_2^2 &= \left\|-m\eta_1\cdot\mathbf{H}_D(k)\big(\mathbf{u}_D(k)-y\big)+\mathbf{I}_D^{(k)}\right\|_2^2 \\
&\leq 2m^2\eta_1^2\lambda_{\max}\big(\mathbf{H}_D(k)\big)^2\,\|y-\mathbf{u}_D(k)\|_2^2+2\left\|\mathbf{I}_D^{(k)}\right\|_2^2.
\end{aligned}
\tag{73}
$$

Similarly with $\mathbf{T}_1$ and $\mathbf{T}_2$, we can control $\mathbf{T}_3$ and $\mathbf{T}_4$ as follows:

***Control on*** $\mathbf{T}_3$***.*** Recall $\mathbf{H}_D(k)$ is a Gram matrix by definition. Then, by using the fact $\lambda_{\min}\big(\mathbf{H}_D(k)\big) \geq 0$ and Cauchy-Schwarz inequality, we have

$$
\begin{aligned}
2m\eta_1&\big(y-(1-\eta_2\mu L)\mathbf{u}_D(k)\big)^\top\mathbf{H}_D(k)\big(\mathbf{u}_D(k)-y\big) \\
&= -2m\eta_1(1-\eta_2\mu L)\big(y-\mathbf{u}_D(k)\big)^\top\mathbf{H}_D(k)\big(y-\mathbf{u}_D(k)\big)+\big(2m\eta_1\eta_2\mu L\big)\cdot y^\top\mathbf{H}_D(k)\big(\mathbf{u}_D(k)-y\big) \\
&\leq \big(2m\eta_1\eta_2\mu L\big)\cdot\lambda_{\max}\big(\mathbf{H}_D(k)\big)\,\|y-\mathbf{u}_D(k)\|_2^2+\big(2m\eta_1\eta_2\mu L\big)\cdot\big(\lambda_{\max}\big(\mathbf{H}_D(k)\big)\,\|y\|_2\,\|y-\mathbf{u}_D(k)\|_2\big) \\
&\quad- 2m\eta_1\lambda_{\min}\big(\mathbf{H}_D(k)\big)\,\|y-\mathbf{u}_D(k)\|_2^2 \\
&= \big(4m\eta_1\eta_2\mu L\big)\cdot\lambda_{\max}\big(\mathbf{H}_D(k)\big)\,\|y-\mathbf{u}_D(k)\|_2^2+\big(4m\eta_1\eta_2\mu L\big)\cdot\lambda_{\max}\big(\mathbf{H}_D(k)\big)\,\|y\|_2^2.
\end{aligned}
\tag{74}
$$

***Control on*** $\mathbf{T}_4$***.*** By a simple Cauchy-Schwarz and Young's inequality, we have

$$
\begin{aligned}
-2&\big(y-(1-\eta_2\mu L)\mathbf{u}_D(k)\big)^\top\mathbf{I}_D^{(k)} \\
&= -2(1-\eta_2\mu L)\big(y-\mathbf{u}_D(k)\big)^\top\mathbf{I}_D^{(k)}+2\eta_2\mu L\cdot y^\top\mathbf{I}_D(k) \\
&\leq 2\big(1-\eta_2\mu L\big)\,\|y-\mathbf{u}_D(k)\|_2\left\|\mathbf{I}_D^{(k)}\right\|_2+\eta_2\mu L\,\|y\|_2^2+\eta_2\mu L\left\|\mathbf{I}_D^{(k)}\right\|_2^2
\end{aligned}
\tag{75}
$$

***Step 4. Upper-bound of the decomposition on training error (62).***

Before getting the upper bound of the decomposition (62), we first work on obtaining the bound of (76). Set $\kappa = \mathcal{O}\left(\frac{1}{n^2}\right)$ and notice $\eta_2\mu L = \mathcal{O}\left(\frac{1}{n}\right)$ by (69), then we have

$$2\left\|\mathbf{I}_D^{(k)}\right\|_2^2 + 2\left(1 - \eta_2\mu L\right)\|y - \mathbf{u}_D(k)\|_2 \left\|\mathbf{I}_D^{(k)}\right\|_2 + \eta_2\mu L \left\|\mathbf{I}_D^{(k)}\right\|_2^2 \tag{76}$$

$$= \left(2 + \eta_2\mu L\right)\left\|\mathbf{I}_D^{(k)}\right\|_2^2 + 2\kappa\left(1 - \eta_2\mu L\right)\|y - \mathbf{u}_D(k)\|_2 \cdot \frac{1}{\kappa}\left\|\mathbf{I}_D^{(k)}\right\|_2$$

$$\leq \left(2 + \eta_2\mu L + \frac{1}{\kappa^2}\right)\left\|\mathbf{I}_D^{(k)}\right\|_2^2 + \kappa^2\left(1 - \eta_2\mu L\right)^2\|y - \mathbf{u}_D(k)\|_2^2$$

$$= \frac{1}{\kappa^2}\cdot\left\|\mathbf{I}_D^{(k)}\right\|_2^2 + \kappa^2\left(1 - \eta_2\mu L\right)^2\|y - \mathbf{u}_D(k)\|_2^2$$

$$\leq \left\{\frac{1}{\kappa^2}\cdot\mathcal{O}\left(L^{37/6}n^{-\frac{9d-8}{6d-3}}\frac{\log(m)}{m^{1/3}\delta^{2/3}}\right) + \kappa^2\left(1 - \eta_2\mu L\right)^2\right\}\cdot\|y - \mathbf{u}_D(k)\|_2^2 + \frac{1}{\kappa^2}\cdot\mathcal{O}_{\mathbb{P}}\left(\frac{1}{n^4}\right)$$

$$\leq \left(\eta_2\mu L\right)^4\left(1 - \eta_2\mu L\right)^2\cdot\|y - \mathbf{u}_D(k)\|_2^2 + \eta_2\mu L\cdot\|y\|_2^2, \tag{77}$$

where in the second inequality, the Eq. (70) is used with $(a+b)^2 \leq 2a^2 + 2b^2$ for $a, b \in \mathbb{R}$, and in the last inequality, we used $\|y\|_2^2 = \mathcal{O}(n)$ and the sufficiently large $m$ to control the order of the coefficient terms of $\|y - \mathbf{u}_D(k)\|_2^2$. Specifically, we choose $m \geq \Omega\left(L^{19}n^{20}\frac{\log^3(m)}{\delta^2}\right)$.

Now, by combining the inequalities (72), (73), (74), (75), (71) and (77), we obtain the upper-bound on the decomposition (62);

$$\|\mathbf{u}_D(k+1) - y\|_2^2$$

$$\leq \left(2\eta_2\mu L + \eta_2^2\mu^2 L^2 + 4m\eta_1\eta_2\mu L\cdot\lambda_{\max}\left(\mathbf{H}_D(k)\right)\right)\cdot\|y\|_2^2$$

$$+ \left(\left(1 + \eta_2\mu L\right)\left(1 - \eta_2\mu L\right)^2 + 2m^2\eta_1^2\lambda_{\max}\left(\mathbf{H}_D(k)\right)^2 + 4m\eta_1\eta_2\mu L\cdot\lambda_{\max}\left(\mathbf{H}_D(k)\right)\right)$$

$$\cdot\|y - \mathbf{u}_D(k)\|_2^2 + \left(2\left\|\mathbf{I}_D^{(k)}\right\|_2^2 + 2\left(1 - \eta_2\mu L\right)\|y - \mathbf{u}_D(k)\|_2\left\|\mathbf{I}_D^{(k)}\right\|_2 + \eta_2\mu L\left\|\mathbf{I}_D^{(k)}\right\|_2^2\right)$$

$$\leq \left\{3\eta_2\mu L + \eta_2^2\mu^2 L^2 + \mathcal{O}\left(\omega m n\eta_1\eta_2\mu L^2\right)\right\}\cdot\|y\|_2^2$$

$$+ \left\{\left(1 + \eta_2\mu L + \eta_2^4\mu^4 L^4\right)\left(1 - \eta_2\mu L\right)^2 + \mathcal{O}\left(\omega^2 m^2 n^2 \eta_1^2 L^2\right) + \mathcal{O}\left(\omega m n\eta_1\eta_2\mu L^2\right)\right\}$$

$$\cdot\|y - \mathbf{u}_D(k)\|_2^2$$

$$:= \mathcal{A}\cdot\|y\|_2^2 + (1 - \mathcal{B})\cdot\|y - \mathbf{u}_D(k)\|_2^2. \tag{78}$$

With the order choices of $\mu$, $\eta_1$ and $\eta_2$ as in (69), it is easy to see the leading terms of both $\mathcal{A}$ and $\mathcal{B}$ are same as $\eta_2\mu L = o(\frac{1}{n})$. Then, by recursively applying the inequality (78), we can get the upper-bound on the training error.

$$\|y - \mathbf{u}_D(k+1)\|_2^2 \leq \mathcal{A}\cdot\|y\|_2^2 + (1 - \mathcal{B})\cdot\|y - \mathbf{u}_D(k)\|_2^2$$

$$\leq \mathcal{A}\|y\|_2^2\cdot\left(\sum_{j=0}^{k}(1 - \mathcal{B})^j\right) + (1 - \mathcal{B})^{k+1}\cdot\|y - \mathbf{u}_D(0)\|_2^2$$

$$\leq \frac{\mathcal{A}}{\mathcal{B}}\cdot\|y\|_2^2 + (1 - \mathcal{B})^{k+1}\cdot\|y - \mathbf{u}_D(0)\|_2^2$$

$$\leq \mathcal{O}(n) + (1 - \eta_2\mu L)^{k+1}\cdot\|y - \mathbf{u}_D(0)\|_2^2. \tag{79}$$

In the last inequality, we used $\frac{\mathcal{A}}{\mathcal{B}} = o(1)$, $\mathcal{B} \geq \eta_2\mu L$ and $\|y\|_2^2 = \mathcal{O}(n)$.

**_Step 5. The order of the radius of perturbation region._** It remains us to prove the radius of perturbation region $\tau$ has the order $\mathcal{O}_{\mathbb{P}}\left(\frac{L\sqrt{\omega}}{\sqrt{m}}n^{\frac{d}{2d-1}}\right)$. First, recall that the $\ell_2$-regularized GD update rule is as:

$$\mathbf{W}_{D,\ell}^{(k)} = \left(1 - \eta_2\mu\right)\mathbf{W}_{D,\ell}^{(k-1)} - \eta_1 \nabla_{\mathbf{w}_\ell}\left[\mathcal{L}_{\mathbf{S}}\left(\mathbf{W}_D^{(k-1)}\right)\right] + \eta_2\mu\mathbf{W}_{D,\ell}^{(0)}, \quad \forall 1 \le \ell \le L \quad \text{and} \quad \forall k \ge 1. \tag{80}$$

Similarly with the proof in the Theorem 3.5, we employ the induction process for the proof. The induction hypothesis is

$$\left\|\mathbf{W}_{D,\ell}^{(s)} - \mathbf{W}_{D,\ell}^{(0)}\right\|_2 \le \mathcal{O}\left(\frac{\eta_1 n\sqrt{m\omega}}{\sqrt{\delta}\eta_2\mu}\right), \qquad \forall s \in [k+1]. \tag{81}$$

It is easy to see it holds for $s = 0$, and suppose it holds for $s = 0, 1, \ldots, k$, we consider $k+1$. Using the update rule (80), we have

$$\begin{aligned}
\left\|\mathbf{W}_{D,\ell}^{(k+1)} - \mathbf{W}_{D,\ell}^{(k)}\right\|_2 &\le \eta_2\mu\left\|\mathbf{W}_{D,\ell}^{(k)} - \mathbf{W}_{D,\ell}^{(0)}\right\|_2 + \eta_1\left\|\nabla_{\mathbf{w}_\ell}\left[\mathcal{L}_{\mathbf{S}}\left(\mathbf{W}_D^{(k)}\right)\right]\right\|_2 \\
&= \eta_2\mu\left\|\mathbf{W}_{D,\ell}^{(k)} - \mathbf{W}_{D,\ell}^{(0)}\right\|_2 + \eta_1\left\|\sum_{i=1}^n\left(y_i - \mathbf{u}_{D,i}(k)\right)\nabla_{\mathbf{w}_\ell}\left[f_{\mathbf{W}_D(k)}(\mathbf{x}_i)\right]\right\|_2 \\
&\le \mathcal{O}\left(\frac{\eta_1 n\sqrt{m\omega}}{\sqrt{\delta}}\right) + \mathcal{O}\left(\eta_1\sqrt{nm\omega}\right) \cdot \|y - \mathbf{u}_D(k)\|_2 \\
&\le \mathcal{O}\left(\frac{\eta_1 n\sqrt{m\omega}}{\sqrt{\delta}}\right) + \mathcal{O}\left(\eta_1\sqrt{nm\omega}\right) \cdot \left\{\mathcal{O}(\sqrt{n}) + (1 - \eta_2\mu L)^{\frac{k}{2}}\mathcal{O}\left(\sqrt{\frac{n}{\delta}}\right)\right\} \\
&\le \mathcal{O}\left(\frac{\eta_1 n\sqrt{m\omega}}{\sqrt{\delta}\eta_2\mu}\right).
\end{aligned}$$

In the first inequality, we use the induction hypothesis for $s = k$, and Lemma 4.4. In the second inequality, since the induction hypothesis holds for $s = 0, 1, \ldots, k$, we employ $\|y - \mathbf{u}_D(k)\|_2 \le \mathcal{O}(\sqrt{n}) + (1 - \eta_2\mu L)^{\frac{k}{2}}\|y - \mathbf{u}_D(0)\|_2$ with the Lemma 4.9. In the last inequality, we use $\eta_2\mu < 1$. By triangle inequality, the induction holds for $s = k + 1$. Plugging the proper choices of $\eta_1, \eta_2$ and $\mu$ as suggested in (69) to $\mathcal{O}\left(\frac{\eta_1 n\sqrt{m\omega}}{\sqrt{\delta}\eta_2\mu}\right)$ yields $\|\mathbf{W}_{D,\ell}^{(k)} - \mathbf{W}_{D,\ell}^{(0)}\|_2 \le \mathcal{O}_{\mathbb{P}}\left(\frac{L\sqrt{\omega}}{\sqrt{m}}n^{\frac{d}{2d-1}}\right)$.

## H PROOF OF THEOREM 3.10-KERNEL RIDGE REGRESSOR APPROXIMATION

We present a following proof sketch on the approximation of regularized DNN estimator to kernel ridge regressor.

1. The key idea for proving the second result in Theorem 3.8 is to write the distance between $\mathbf{u}_{i,D}(k)$ (where $D$ is to denote the prediction is obtained from regularized GD rule) and kernel regressor $\mathbf{B} := \mathbf{H}_L^\infty\left(C\mu \cdot \mathcal{I} + \mathbf{H}_L^\infty\right)^{-1}\mathbf{y}$ in terms of NTK matrix $\mathbf{H}_L^\infty$, which is as follows:

$$\mathbf{u}_D(k) - \mathbf{B} = \left(\left(1 - \eta_2\mu L\right) \cdot \mathcal{I} - m\eta_1\mathbf{H}_L^\infty\right)^k\left(\mathbf{u}_D(0) - \mathbf{B}\right) + \mathbf{e}_D(k).$$

Above equality describes how the regularized estimator evolves to fit the kernel regressor as iteration of algorithm goes by.

2. We can bound the $\ell_2$-norm of residual term $\mathbf{e}_D(k)$ as $\mathcal{O}(1/n)$, and show that the $\ell_2$ norm of the first term on the RHS of equation (4.3) decays at the rate $\mathcal{O}\left(\sqrt{n}\left(1 - \eta_2\mu L\right)^k\right)$. Here the $\sqrt{n}$ comes from the bound $\|\mathbf{B}\|_2 \le \mathcal{O}(\sqrt{n})$, since we know $\|\mathbf{u}(0)\|_2$ has $\mathcal{O}(\sqrt{n\omega})$ with small $\omega \le 1$. This yields the claim.

Recall the equality (65). Then, we have

$$
\begin{aligned}
\mathbf{u}_D(k+1) &- (1 - \eta_2\mu L)\mathbf{u}_D(k) \\
&= -m\eta_1 \cdot \mathbf{H}_D(k)\big(\mathbf{u}_D(k) - \mathbf{y}\big) + \mathbf{I}_{2,D}^{(k)} + \mathbf{I}_{3,D}^{(k)} + \mathbf{I}_{5,D}^{(k)} \\
&= -m\eta_1 \cdot \mathbf{H}_L^{\infty}\big(\mathbf{u}_D(k) - \mathbf{y}\big) \\
&\quad - m\eta_1 \cdot \big(\mathbf{H}_D(k) - \mathbf{H}_L^{\infty}\big)\big(\mathbf{u}_D(k) - \mathbf{y}\big) + \mathbf{I}_{2,D}^{(k)} + \mathbf{I}_{3,D}^{(k)} + \mathbf{I}_{5,D}^{(k)} \\
&= -m\eta_1 \cdot \mathbf{H}_L^{\infty}\big(\mathbf{u}_D(k) - \mathbf{y}\big) + \xi_D(k).
\end{aligned}
\tag{82}
$$

With $\tau = \mathcal{O}\Big(\frac{L\sqrt{\omega}}{\sqrt{m}\delta}n^{\frac{d}{2d-1}}\Big)$, similarly with Lemma 4.10 and a direct employment of the result from Lemma 4.11, we can control the distance from $\mathbf{H}_D(k)$ to $\mathbf{H}_L^{\infty}$ under operator norm as follows:

$$
\begin{aligned}
\|\mathbf{H}_D(k) - \mathbf{H}_L^{\infty}\|_2 &\leq \|\mathbf{H}_D(k) - \mathbf{H}(0)\|_2 + \|\mathbf{H}(0) - \mathbf{H}_L^{\infty}\|_2 \\
&\leq \mathcal{O}\left(\omega^{7/6}L^{10/3}n^{\frac{7d-3}{6d-3}}\sqrt[6]{\frac{\log^3(m)}{m\delta^2}}\right) + \mathcal{O}\left(\omega L^{5/2}n\sqrt[4]{\frac{\log(nL/\delta)}{m}}\right) \\
&\leq \mathcal{O}\left(L^{19/12}n^{-\frac{21d-8}{12d-6}}\sqrt[6]{\frac{\log^3(m)}{m\delta^2}}\right) + \mathcal{O}\left(Ln^{-\frac{18d-6}{12d-6}}\sqrt[4]{\frac{\log(nL/\delta)}{m}}\right) \\
&\leq \mathcal{O}\left(L^{19/12}n^{-\frac{21d-8}{12d-6}}\sqrt[6]{\frac{\log^3(m)}{m\delta^2}}\right),
\end{aligned}
\tag{83}
$$

where in the third inequality, $\omega = \mathcal{O}\big(\frac{1}{L^{3/2}}n^{-\frac{5d-2}{2d-1}}\big)$ is plugged-in. The last inequality holds with $d \geq 2$ with large enough $n$ and the condition on width $m \geq \Omega\big(L^{19}n^{20}\frac{\log^3(m)}{\delta^2}\big)$. Then, the $\ell_2$ norm of $\xi_D(k)$ can be bounded as:

$$
\begin{aligned}
\|\xi_D(k)\|_2 &\leq m\eta_1 \cdot \|\mathbf{H}_L^{\infty} - \mathbf{H}_D(k)\|_2 \|\mathbf{u}_D(k) - y\|_2 + \left\|\mathbf{I}_D^{(k)}\right\|_2 \\
&\leq \mathcal{O}\left(L^{19/12}n^{-\frac{12d-5}{6d-3}}\frac{\sqrt{\log(m)}}{m^{1/6}\delta^{1/3}}\right) \cdot \underbrace{\|\mathbf{u}_D(k) - y\|_2}_{\leq \mathcal{O}(\sqrt{n/\delta})} + \mathcal{O}_{\mathbb{P}}\left(\frac{1}{n^2}\right) \\
&\leq \mathcal{O}\left(L^{19/12}n^{-\frac{18d-7}{12d-6}}\frac{\sqrt{\log(m)}}{m^{1/6}\delta^{5/6}}\right) + \mathcal{O}_{\mathbb{P}}\left(\frac{1}{n^2}\right) = \mathcal{O}_{\mathbb{P}}\left(\frac{1}{n^2}\right),
\end{aligned}
\tag{84}
$$

where in the second inequality, we used (83) with $\eta_1 = \mathcal{O}\Big(\frac{1}{m}n^{-\frac{3d-2}{2d-1}}\Big)$ to control the first term and employed Eq. (70) to control the second term. In the last equality, we used $m \geq \Omega\big(L^{19}n^{20}\frac{\log^3(m)}{\delta^2}\big)$. Now, by setting $\mathbf{B} := \Big(\frac{\eta_2\mu L}{\eta_1 m}\mathcal{I} + \mathbf{H}_L^{\infty}\Big)^{-1}\mathbf{H}_L^{\infty}\mathbf{y}$, we can easily convert the equality (82) as follows: for $k \geq 1$,

$$
\mathbf{u}_D(k) - \mathbf{B} = \Big((1 - \eta_2\mu L) \cdot \mathcal{I} - m\eta_1\mathbf{H}_L^{\infty}\Big)\big(\mathbf{u}_D(k-1) - \mathbf{B}\big) + \xi_D(k-1).
\tag{85}
$$

The recursive applications of the equality (85) yields

$$
\begin{aligned}
\mathbf{u}_D(k) - \mathbf{B} &= \Big((1 - \eta_2\mu L) \cdot \mathcal{I} - m\eta_1\mathbf{H}_L^{\infty}\Big)^k \big(\mathbf{u}_D(0) - \mathbf{B}\big) \\
&\quad + \sum_{j=0}^{k}\Big((1 - \eta_2\mu L) \cdot \mathcal{I} - m\eta_1\mathbf{H}_L^{\infty}\Big)^j \xi_D(k-j-1) \\
&= \Big((1 - \eta_2\mu L) \cdot \mathcal{I} - m\eta_1\mathbf{H}_L^{\infty}\Big)^k \big(\mathbf{u}_D(0) - \mathbf{B}\big) + \mathbf{e}_D(k).
\end{aligned}
\tag{86}
$$

Now, we bound the $\ell_2$ norm of $\mathbf{e}_D(k)$ in (86):

$$
\begin{aligned}
\|\mathbf{e}_D(k)\|_2 &= \left\| \sum_{j=0}^{k} \left( \left(1 - \eta_2 \mu L\right) \cdot \mathcal{I} - m\eta_1 \mathbf{H}_L^\infty \right)^j \xi_D(k - j - 1) \right\|_2 \\
&\leq \sum_{j=0}^{k} \left\| \left(1 - \eta_2 \mu L\right) \cdot \mathcal{I} - m\eta_1 \mathbf{H}_L^\infty \right\|_2^j \|\xi_D(k - j - 1)\|_2 \\
&\leq \sum_{j=0}^{k} \left(1 - \eta_2 \mu L\right)^j \|\xi_D(k - j - 1)\|_2 = \mathcal{O}\left(\frac{1}{n}\right),
\end{aligned} \tag{87}
$$

in the last inequality, we used $\eta_2 \mu L = \mathcal{O}\left(\frac{1}{n}\right)$ and Eq. (84). Now, we control the $\ell_2$-norm of the first term in (86) as:

$$
\begin{aligned}
\left\| \left( \left(1 - \eta_2 \mu L\right) \cdot \mathcal{I} - m\eta_1 \mathbf{H}_L^\infty \right)^k \left(\mathbf{u}_D(0) - \mathbf{B}\right) \right\|_2 &\leq \left(1 - \eta_2 \mu L\right)^k \|\mathbf{u}_D(0) - \mathbf{B}\|_2 \\
&\leq \mathcal{O}\left( \sqrt{n}\left(1 - \eta_2 \mu L\right)^k \right),
\end{aligned} \tag{88}
$$

where in the second inequality, we used $\|\mathbf{u}_D(0)\|_2 \leq \mathcal{O}(\sqrt{n\omega}/\delta)$ and the fact that

$$
\|\mathbf{B}\|_2 \leq \left\| \left( \frac{\eta_2 \mu L}{\eta_1 m} \mathcal{I} + \mathbf{H}_L^\infty \right)^{-1} \mathbf{H}_L^\infty \right\|_2 \cdot \|\mathbf{y}\|_2 \leq \mathcal{O}(\sqrt{n}).
$$

By combining (87) and (88) and using a fact $(1 - \eta_2 \mu L)^k \leq \exp(-\eta_2 \mu L k)$, we conclude that after $k \geq \Omega\left((\eta_2 \mu L)^{-1} \log(n^{3/2})\right)$, the error $\|\mathbf{u}_D(k) - \mathbf{B}\|_2$ decays at the rate $\mathcal{O}\left(\frac{1}{n}\right)$.

## I  PROOF OF THEOREM 3.11

We begin the proof by decomposing the error $\widehat{f}_{\mathbf{W}_D^{(k)}}(x) - f^*(x)$ for any fixed $x \in \mathbf{Unif}(\mathcal{S}^{d-1})$ into two terms as follows:

$$
\widehat{f}_{\mathbf{W}_D^{(k)}}(x) - f^*(x) = \underbrace{\left(\widehat{f}_{\mathbf{W}_D^{(k)}}(x) - g_\mu^*(x)\right)}_{\Delta_{D,1}} + \underbrace{\left(g_\mu^*(x) - f^*(x)\right)}_{\Delta_{D,2}}. \tag{89}
$$

Here, we devise a solution of kernel ridge regression $g_\mu^*(x)$ in the decomposition (89):

$$
g_\mu^*(x) := \mathbf{Ker}(x, \mathbf{X})\left(C\mu \cdot \mathcal{I} + \mathbf{H}_L^\infty\right)^{-1}\mathbf{y},
$$

for some constant $C > 0$. Specifically, in the proof to follow, we choose $\eta_1$ and $\eta_2$ such that $C = \frac{\eta_2 L}{\eta_1 m}$ for the theoretical convenience. Our goal is to show that all the terms $\|\Delta_{D,1}\|_2^2$, and $\|\Delta_{D,2}\|_2^2$ have the order either equal to or smaller than $\mathcal{O}\left(n^{-\frac{d}{2d-1}}\right)$ with the proper choices on $m$, $\mu$, $\eta_1$ and $\eta_2$. Since the high-level proof idea is similar with that of Theorem 3.8, we omit the step-by-step proof sketch of Theorem 3.11. The most notable difference between the proof strategies of the two theorems is that the regularized DNN approximate the kernel ridge regressor of noisy data, whereas in Theorem 3.8, unregularized DNN approximate the interpolant based on noiseless data.

***Step 1. Control on*** $\Delta_{D,2}$***.***    First, note that there is a recent finding that the reproducing kernel Hilbert spaces induced from NTKs with any number of layers (i.e., $L \geq 1$) have the same set of functions, if kernels are defined on $\mathcal{S}^{d-1}$. See Chen & Xu [2020]. Along with this result, under the choice of model parameters as suggested in (69), we can apply exactly the same proof used in Theorem.3.2 in Hu et al. [2021] for proving a following :

$$
\|\Delta_{D,2}\|_2^2 := \left\| g_\mu^* - f^* \right\|_2^2 = \mathcal{O}_{\mathbb{P}}\left( n^{-\frac{d}{2d-1}} \right), \qquad \|g_\mu^*\|_{\mathcal{H}}^2 = \mathcal{O}_{\mathbb{P}}(1). \tag{90}
$$

**_Step 2. Control on_** $\Delta_{D,1}$. For $n$ data points $(\mathbf{x}_1, \ldots, \mathbf{x}_n)$ and for the $k^{\text{th}}$ updated parameter $\mathbf{W}_D^{(k)}$, denote:

$$\nabla_{\mathbf{W}_\ell}\big[f_{\mathbf{W}_D^{(k)}}(\mathbf{X})\big] = \left[\mathbf{vec}\Big(\nabla_{\mathbf{W}_\ell}\big[f_{\mathbf{W}_D^{(k)}}(\mathbf{x}_1)\big]\Big), \cdots, \mathbf{vec}\Big(\nabla_{\mathbf{W}_\ell}\big[f_{\mathbf{W}_D^{(k)}}(\mathbf{x}_n)\big]\Big)\right].$$

Note that when $\ell = 1$, $\nabla_{\mathbf{W}_\ell}\big[f_{\mathbf{W}_D^{(k)}}(\mathbf{X})\big] \in \mathbb{R}^{md \times n}$ and when $\ell = 2, \ldots, L$, $\nabla_{\mathbf{W}_\ell}\big[f_{\mathbf{W}_D^{(k)}}(\mathbf{X})\big] \in \mathbb{R}^{m^2 \times n}$.

With this notation, we can write the vectorized version of the update rule (80) as:

$$\mathbf{vec}\big(\mathbf{W}_{D,\ell}^{(k)}\big) = \mathbf{vec}\big(\mathbf{W}_{D,\ell}^{(0)}\big) - \eta_1 \sum_{j=0}^{k-1} (1 - \eta_2\mu)^j \nabla_{\mathbf{W}_\ell}\big[f_{\mathbf{W}_D(k-j-1)}(\mathbf{X})\big]\bigg(\mathbf{u}_D(k-j-1) - y\bigg),$$

$\forall 1 \le \ell \le L$ and $\forall k \ge 1$. Using the equality, we can get the decomposition :

$$\mathbf{vec}\big(\mathbf{W}_{D,\ell}^{(k)}\big) = \underbrace{\mathbf{vec}\big(\mathbf{W}_{D,\ell}^{(0)}\big)}_{:=\mathbf{E}_1} \underbrace{-\eta_1 \nabla_{\mathbf{W}_\ell}\big[f_{\mathbf{W}_D(0)}(\mathbf{X})\big] \sum_{j=0}^{k-1}(1-\eta_2\mu)^j\bigg(\mathbf{u}_D(k-j-1)-y\bigg)}_{:=\mathbf{E}_2}$$
$$\underbrace{-\eta_1 \sum_{j=0}^{k-1}(1-\eta_2\mu)^j\bigg[\nabla_{\mathbf{W}_\ell}\big[f_{\mathbf{W}_D(k-j-1)}(\mathbf{X})\big] - \nabla_{\mathbf{W}_\ell}\big[f_{\mathbf{W}_D(0)}(\mathbf{X})\big]\bigg]\bigg(\mathbf{u}_D(k-j-1)-y\bigg)}_{:=\mathbf{E}_3}.$$
$$\tag{91}$$

Let $z_{D,k}(x) := \mathbf{vec}\big(\nabla_{\mathbf{W}_\ell}\big[f_{\mathbf{W}_D^{(k)}}(x)\big]\big)$, and note that $f_{\mathbf{W}_D^{(k)}}(x) = \langle z_{D,k}(x), \mathbf{vec}\big(\mathbf{W}_{D,\ell}^{(k)}\big)\rangle$. Then, by the definition of $\Delta_{D,1}$ and the decomposition (91), we have

$$\Delta_{D,1} = \frac{1}{L}\sum_{\ell=1}^{L}\langle z_{D,k}(x), \mathbf{E}_1 + \mathbf{E}_2 + \mathbf{E}_3\rangle - \mathbf{Ker}(x, \mathbf{X})\bigg(\frac{\eta_2\mu L}{\eta_1 m}\mathcal{I} + \mathbf{H}_L^\infty\bigg)^{-1}\mathbf{y}$$

$$= \frac{1}{L}\sum_{\ell=1}^{L}\langle z_{D,k}(x), \mathbf{E}_1\rangle + \frac{1}{L}\sum_{\ell=1}^{L}\langle z_{D,k}(x), \mathbf{E}_3\rangle$$

$$+ \underbrace{\frac{1}{L}\sum_{\ell=1}^{L}\langle z_{D,k}(x), \mathbf{E}_2\rangle - \mathbf{Ker}(x, \mathbf{X})\bigg(\frac{\eta_2\mu L}{\eta_1 m}\mathcal{I} + \mathbf{H}_L^\infty\bigg)^{-1}\mathbf{y}}_{:=\mathcal{C}} \tag{92}$$

First, we focus on controlling the $\ell_2$ bound on the first two terms in (92). Observe that the first term can be bounded as:

$$\left|\frac{1}{L}\sum_{\ell=1}^{L}\langle z_{D,k}(x), \mathbf{E}_1\rangle\right|^2 \le \frac{1}{L}\sum_{\ell=1}^{L}|\langle z_{D,k}(x), \mathbf{E}_1\rangle|^2. \tag{93}$$

Recall that $\|z_{D,k}(x)\|_2 \le \mathcal{O}\big(\sqrt{m\omega}\big)$ by Lemma 4.4. Then, the random variable $z_{D,k}(x)^\top \mathbf{vec}\big(\mathbf{W}_{D,\ell}^{(0)}\big) \mid z_{D,k}(x)$ is simply a $\mathcal{N}\big(0, \mathcal{O}(\omega)\big)$ for $1 \le \ell \le L$. A straightforward application of Chernoff bound for normal random variable and taking union bound over the layer $1 \le \ell \le L$ yield that: with probability at least $1 - \delta$,

$$\frac{1}{L}\sum_{\ell=1}^{L}\left|z_{D,k}(x)^\top \mathbf{vec}\big(\mathbf{W}_{D,\ell}^{(0)}\big)\right|^2 \le \mathcal{O}\bigg(\omega\log\bigg(\frac{L}{\delta}\bigg)\bigg). \tag{94}$$

The $\ell_2$ norm of the second term in (92) can be similarly bounded as (93) in addition with the Cauchy-Schwarz inequality:

$$\left|\frac{1}{L}\sum_{\ell=1}^{L}\langle z_{D,k}(x), \mathbf{E}_3\rangle\right|^2 \le \frac{1}{L}\sum_{\ell=1}^{L}|\langle z_{D,k}(x), \mathbf{E}_3\rangle|^2 \le \frac{1}{L}\sum_{\ell=1}^{L}\|z_{D,k}(x)\|_2^2 \|\mathbf{E}_3\|_2^2. \tag{95}$$

The $\|\mathbf{E}_3\|_2$ is bounded as :

$$
\|\mathbf{E}_3\|_2 = \left\| \eta_1 \sum_{j=0}^{k-1} \left(1 - \eta_2\mu\right)^j \left[ \nabla_{\mathbf{w}_\ell}\left[f_{\mathbf{W}_D(k-j-1)}(\mathbf{X})\right] - \nabla_{\mathbf{w}_\ell}\left[f_{\mathbf{W}_D(0)}(\mathbf{X})\right] \right] \left(\mathbf{u}_D(k-j-1) - y\right) \right\|_2
$$

$$
\leq \eta_1 \sum_{j=0}^{k-1} \left(1 - \eta_2\mu\right)^j \cdot \left\| \nabla_{\mathbf{w}_\ell}\left[f_{\mathbf{W}_D(k-j-1)}(\mathbf{X})\right] - \nabla_{\mathbf{w}_\ell}\left[f_{\mathbf{W}_D(0)}(\mathbf{X})\right] \right\|_2 \|\mathbf{u}_D(k-j-1) - y\|_2
$$

$$
\leq \eta_1 \sum_{j=0}^{k-1} \left(1 - \eta_2\mu\right)^j \cdot \left\| \nabla_{\mathbf{w}_\ell}\left[f_{\mathbf{W}_D(k-j-1)}(\mathbf{X})\right] - \nabla_{\mathbf{w}_\ell}\left[f_{\mathbf{W}_D(0)}(\mathbf{X})\right] \right\|_F \|\mathbf{u}_D(k-j-1) - y\|_2
$$

$$
= \eta_1 \sum_{j=0}^{k-1} \left(1 - \eta_2\mu\right)^j \cdot \sqrt{\sum_{i=1}^{n} \left\| \nabla_{\mathbf{w}_\ell}\left[f_{\mathbf{W}_D(k-j-1)}(\mathbf{x}_i)\right] - \nabla_{\mathbf{w}_\ell}\left[f_{\mathbf{W}_D(0)}(\mathbf{x}_i)\right] \right\|_F^2} \|\mathbf{u}_D(k-j-1) - y\|_2
$$

$$
\leq \eta_1 \sum_{j=0}^{k-1} \left(1 - \eta_2\mu\right)^j \cdot \sqrt{2\sum_{i=1}^{n} \left\| \nabla_{\mathbf{w}_\ell}\left[f_{\mathbf{W}_D(k-j-1)}(\mathbf{x}_i)\right] - \nabla_{\mathbf{w}_\ell}\left[f_{\mathbf{W}_D(0)}(\mathbf{x}_i)\right] \right\|_2^2} \|\mathbf{u}_D(k-j-1) - y\|_2
$$

$$
\leq \frac{\eta_1}{\eta_2\mu} \cdot \mathcal{O}\left( \tau^{1/3} L^2 \sqrt{\omega m n \log(m)} \right) \cdot \mathcal{O}\left(\sqrt{n}\right) \leq \mathcal{O}\left( \frac{L^{10/3} \omega^{1/6}}{m^{2/3}\delta^{1/3}} n^{\frac{4d}{6d-3}} \sqrt{\log(m)} \right). \tag{96}
$$

In the first, second and third inequalities, we used a simple fact that for the matrix $A \in \mathbb{R}^{d_1 \times d_2}$ with rank $r$, then $\|A\|_2 \leq \|A\|_F \leq \sqrt{r}\|A\|_2$. Recall that the rank of the matrix $\nabla_{\mathbf{w}_\ell}\left[f_{\mathbf{W}_D(k-j-1)}(x)\right] - \nabla_{\mathbf{w}_\ell}\left[f_{\mathbf{W}_D(0)}(x)\right]$ is at most 2. In the second to the last inequality, we use the result of Lemma 4.6 and the $\|\mathbf{u}_D(i) - \mathbf{y}\|_2 \leq \mathcal{O}(\sqrt{n})$ for any $i \geq 1$. In the last inequality, we plug the correct orders as set in (69) to $\tau$, $\eta_1$, $\eta_2$ and $\mu$. Back to the inequality (95), using the $\|z_{D,k}(x)\|_2 \leq \mathcal{O}(\sqrt{m\omega})$ and (96), we can get

$$
\frac{1}{L} \sum_{\ell=1}^{L} \|z_{D,k}(x)\|_2^2 \|\mathbf{E}_3\|_2^2 \leq \mathcal{O}_{\mathbb{P}}\left( \frac{L^{20/3}\omega^{4/3}}{m^{1/3}} n^{\frac{8d}{6d-3}} \log(m) \right). \tag{97}
$$

Before controlling the $\ell_2$ norm of $\mathcal{C}$ in (92), recall that we set $\mathbf{B} := \left( \frac{\eta_2\mu L}{\eta_1 m}\mathcal{I} + \mathbf{H}_L^\infty \right)^{-1} \mathbf{H}_L^\infty \mathbf{y}$ and the dynamics of $\mathbf{u}_D(k) - \mathbf{B}$ can be expressed in terms of $\mathbf{H}_L^\infty$ as follows: For any $k \geq 1$,

$$
\mathbf{u}_D(k) - \mathbf{B} = \left( \left(1 - \eta_2\mu L\right) \cdot \mathcal{I} - m\eta_1 \mathbf{H}_L^\infty \right)^k \left(\mathbf{u}_D(0) - \mathbf{B}\right) + \mathbf{e}_D(k), \tag{98}
$$

with $\|\mathbf{e}_D(k)\|_2 \leq \mathcal{O}\left(\frac{1}{n}\right)$. Using (98), we can further decompose the term $\mathbf{E}_2$ in (91) as:

$$
\mathbf{E}_2 := -\eta_1 \nabla_{\mathbf{w}_\ell}\left[f_{\mathbf{W}_D(0)}(\mathbf{X})\right] \sum_{j=0}^{k-1} \left(1 - \eta_2\mu\right)^j \left(\mathbf{u}_D(k-j-1) - y\right)
$$

$$
= \eta_1 \nabla_{\mathbf{w}_\ell}\left[f_{\mathbf{W}_D(0)}(\mathbf{X})\right] \sum_{j=0}^{k-1} \left(1 - \eta_2\mu\right)^j \left( \left(1 - \eta_2\mu L\right) \cdot \mathcal{I} - m\eta_1 \mathbf{H}_L^\infty \right)^{k-j-1} \mathbf{B}
$$

$$
- \eta_1 \nabla_{\mathbf{w}_\ell}\left[f_{\mathbf{W}_D(0)}(\mathbf{X})\right] \sum_{j=0}^{k-1} \left(1 - \eta_2\mu\right)^j \left( \left(1 - \eta_2\mu L\right) \cdot \mathcal{I} - m\eta_1 \mathbf{H}_L^\infty \right)^{k-j-1} \mathbf{u}_D(0)
$$

$$
- \eta_1 \nabla_{\mathbf{w}_\ell}\left[f_{\mathbf{W}_D(0)}(\mathbf{X})\right] \sum_{j=0}^{k-1} \left(1 - \eta_2\mu\right)^j \mathbf{e}_D(k-j-1)
$$

$$
- \eta_1 \nabla_{\mathbf{w}_\ell}\left[f_{\mathbf{W}_D(0)}(\mathbf{X})\right] \sum_{j=0}^{k-1} \left(1 - \eta_2\mu\right)^j \left(\mathbf{B} - y\right)
$$

$$
= \mathbf{E}_{2,1} + \mathbf{E}_{2,2} + \mathbf{E}_{2,3} + \mathbf{E}_{2,4}. \tag{99}
$$

Then, we can re-write the error term $\mathcal{C}$ in (92) as:

$$\mathcal{C} = \frac{1}{L}\sum_{\ell=1}^{L}\langle z_{D,k}(x), \mathbf{E}_{2,1}\rangle + \frac{1}{L}\sum_{\ell=1}^{L}\langle z_{D,k}(x), \mathbf{E}_{2,2}\rangle + \frac{1}{L}\sum_{\ell=1}^{L}\langle z_{D,k}(x), \mathbf{E}_{2,3}\rangle$$
$$+ \underbrace{\left\{\frac{1}{L}\sum_{\ell=1}^{L}\langle z_{D,k}(x), \mathbf{E}_{2,4}\rangle - \mathbf{Ker}(x,\mathbf{X})\left(\frac{\eta_2\mu L}{\eta_1 m}\mathcal{I} + \mathbf{H}_L^\infty\right)^{-1}\mathbf{y}\right\}}_{:=\mathcal{D}}. \qquad (100)$$

Our goal is to control the $\ell_2$ norm of each summand in the equality (100). For the first three terms in (100), a simple Cauchy-Schwarz inequality can be applied: for $i = 1, 2, 3$:

$$\left|\frac{1}{L}\sum_{\ell=1}^{L}\langle z_{D,k}(x), \mathbf{E}_{2,i}\rangle\right|^2 \le \frac{1}{L}\sum_{\ell=1}^{L}|\langle z_{D,k}(x), \mathbf{E}_{2,i}\rangle|^2 \le \frac{1}{L}\sum_{\ell=1}^{L}\|z_{D,k}(x)\|_2^2 \cdot \|\mathbf{E}_{2,i}\|_2^2.$$

We work on obtaining the bound of $\sum_{\ell=1}^{L}\|\mathbf{E}_{2,1}\|_2^2$. Let $\mathcal{T}_k$ be defined as

$$\mathcal{T}_k := \sum_{j=0}^{k-1}(1 - \eta_2\mu)^j\left((1 - \eta_2\mu L)\cdot\mathcal{I} - m\eta_1\mathbf{H}_L^\infty\right)^{k-j-1}.$$

Then, we have

$$\sum_{\ell=1}^{L}\|\mathbf{E}_{2,1}\|_2^2 = \eta_1^2\sum_{\ell=1}^{L}\left(\mathbf{B}^\top\mathcal{T}_k^\top\nabla_{\mathbf{W}_\ell}[f_{\mathbf{W}_D(0)}(\mathbf{X})]^\top\nabla_{\mathbf{W}_\ell}[f_{\mathbf{W}_D(0)}(\mathbf{X})]\mathcal{T}_k\mathbf{B}\right)$$
$$= m\eta_1^2\mathbf{B}^\top\mathcal{T}_k^\top\mathbf{H}(0)\mathcal{T}_k\mathbf{B}$$
$$= m\eta_1^2\mathbf{B}^\top\mathcal{T}_k^\top\left(\mathbf{H}(0) - \mathbf{H}_L^\infty\right)\mathcal{T}_k\mathbf{B} + m\eta_1^2\mathbf{B}^\top\mathcal{T}_k^\top\mathbf{H}_L^\infty\mathcal{T}_k\mathbf{B}$$
$$\le m\eta_1^2\|\mathbf{H}(0) - \mathbf{H}_L^\infty\|_2\cdot\mathbf{B}^\top\mathcal{T}_k^2\mathbf{B} + m\eta_1^2\mathbf{B}^\top\mathcal{T}_k^\top\mathbf{H}_L^\infty\mathcal{T}_k\mathbf{B}. \qquad (101)$$

To obtain the upper-bound on (101), we need to control the terms $\mathcal{T}_k^\top\mathbf{H}_L^\infty\mathcal{T}_k$ and $\mathbf{B}^\top\mathcal{T}_k^2\mathbf{B}$. Let us denote $\mathbf{H}_L^\infty = \sum_{i=1}^{n}\lambda_i v_i v_i^\top$ be the eigen-decomposition of $\mathbf{H}_L^\infty$. Using $1 - \eta_2\mu L \le 1 - \eta_2\mu$, note that

$$\mathcal{T}_k = \sum_{j=0}^{k-1}(1 - \eta_2\mu)^j(1 - \eta_2\mu L)^{k-j-1}\left(\mathcal{I} - \frac{m\eta_1}{1 - \eta_2\mu L}\mathbf{H}_L^\infty\right)^{k-j-1}$$
$$\preceq (1 - \eta_2\mu)^{k-1}\sum_{i=0}^{k-1}\left(\mathcal{I} - \frac{m\eta_1}{1 - \eta_2\mu}\mathbf{H}_L^\infty\right)^i$$
$$= (1 - \eta_2\mu)^{k-1}\sum_{j=0}^{n}\left(\frac{1 - \left(1 - \frac{m\eta_1}{1-\eta_2\mu}\lambda_j\right)^k}{\frac{m\eta_1}{1-\eta_2\mu}\lambda_j}\right)v_j v_j^\top \preceq \frac{(1 - \eta_2\mu)^k}{m\eta_1\lambda_\infty}\cdot\mathcal{I}. \qquad (102)$$

A similar logic can be applied to bound $\mathcal{T}_k^\top\mathbf{H}_L^\infty\mathcal{T}_k$:

$$\mathcal{T}_k^\top\mathbf{H}_L^\infty\mathcal{T}_k \preceq (1 - \eta_2\mu)^{k-1}\sum_{j=0}^{n}\left(\frac{1 - \left(1 - \frac{m\eta_1}{1-\eta_2\mu}\lambda_j\right)^k}{\frac{m\eta_1}{1-\eta_2\mu}\lambda_j}\right)^2\lambda_j v_j v_j^\top$$
$$\preceq \frac{(1 - \eta_2\mu)^{2k}}{m^2\eta_1^2}\cdot\left(\mathbf{H}_L^\infty\right)^{-1}. \qquad (103)$$

Recall the definition of the notation $\mathbf{B} := \mathbf{H}_L^\infty \left( \frac{\eta_2 \mu L}{\eta_1 m} \mathcal{I} + \mathbf{H}_L^\infty \right)^{-1} \mathbf{y}$. Then, we can bound the term $\mathbf{B}^\top \mathcal{T}_k^\top \mathbf{H}_L^\infty \mathcal{T}_k \mathbf{B}$:

$$
\begin{aligned}
\mathbf{B}^\top \mathcal{T}_k^\top \mathbf{H}_L^\infty \mathcal{T}_k \mathbf{B} &\leq \frac{\left(1 - \eta_2 \mu\right)^{2k}}{m^2 \eta_1^2} \cdot \mathbf{B}^\top \left(\mathbf{H}_L^\infty\right)^{-1} \mathbf{B} \\
&= \frac{\left(1 - \eta_2 \mu\right)^{2k}}{m^2 \eta_1^2} \cdot \mathbf{y}^\top \left( \frac{\eta_2 \mu L}{\eta_1 m} \mathcal{I} + \mathbf{H}_L^\infty \right)^{-1} \mathbf{H}_L^\infty \left( \frac{\eta_2 \mu L}{\eta_1 m} \mathcal{I} + \mathbf{H}_L^\infty \right)^{-1} \mathbf{y} \\
&= \mathcal{O}\left( \frac{\left(1 - \eta_2 \mu\right)^{2k}}{m^2 \eta_1^2} \right),
\end{aligned}
\tag{104}
$$

where in the last equality, we used $\left\| g_\mu^* \right\|_{\mathcal{H}}^2 = \mathcal{O}_{\mathbb{P}}(1)$ in (90). Now we turn our attention to bound the term $\mathbf{B}^\top \mathcal{T}_k^2 \mathbf{B}$,

$$
\begin{aligned}
\mathbf{B}^\top \mathcal{T}_k^2 \mathbf{B} &\leq \frac{\left(1 - \eta_2 \mu\right)^{2k}}{m^2 \eta_1^2 \lambda_\infty^2} \mathbf{y}^\top \left( \frac{\eta_2 \mu L}{\eta_1 m} \mathcal{I} + \mathbf{H}_L^\infty \right)^{-1} \left(\mathbf{H}_L^\infty\right)^2 \left( \frac{\eta_2 \mu L}{\eta_1 m} \mathcal{I} + \mathbf{H}_L^\infty \right)^{-1} \mathbf{y} \\
&= \mathcal{O}\left( \frac{\left(1 - \eta_2 \mu\right)^{2k} n}{m^2 \eta_1^2 \lambda_\infty^2} \right),
\end{aligned}
\tag{105}
$$

where we used $\|y\|_2^2 = \mathcal{O}(n)$ in the last inequality. Combining the bounds (104), (105) and the result from Lemma 4.11, we can further bound (101) and have:

$$
\sum_{\ell=1}^{L} \left\| \mathbf{E}_{2,1} \right\|_2^2 \leq \mathcal{O}\left( \omega \frac{\left(1 - \eta_2 \mu\right)^{2k}}{m \lambda_\infty^2} n^2 L^{5/2} \sqrt[4]{\frac{\log(nL/\delta)}{m}} + \frac{\left(1 - \eta_2 \mu\right)^{2k}}{m} \right) \leq \mathcal{O}\left( \frac{\left(1 - \eta_2 \mu\right)^{2k}}{m} \right),
\tag{106}
$$

where in the second inequality, we used $m \geq \Omega\left( L^{19} n^{20} \frac{\log^3(m)}{\delta^2} \right)$. Similarly, we can bound $\sum_{\ell=1}^{L} \left\| \mathbf{E}_{2,2} \right\|_2^2$:

$$
\begin{aligned}
\sum_{\ell=1}^{L} \left\| \mathbf{E}_{2,2} \right\|_2^2 &= \eta_1^2 \sum_{\ell=1}^{L} \left( \mathbf{u}_D(0)^\top \mathcal{T}_k^\top \nabla_{\mathbf{w}_\ell} \left[ f_{\mathbf{W}_D(0)}(\mathbf{X}) \right]^\top \nabla_{\mathbf{w}_\ell} \left[ f_{\mathbf{W}_D(0)}(\mathbf{X}) \right] \mathcal{T}_k \mathbf{u}_D(0) \right) \\
&= m \eta_1^2 \mathbf{u}_D(0)^\top \mathcal{T}_k^\top \mathbf{H}(0) \mathcal{T}_k \mathbf{u}_D(0) \\
&= m \eta_1^2 \mathbf{u}_D(0)^\top \mathcal{T}_k^\top \left( \mathbf{H}(0) - \mathbf{H}_L^\infty \right) \mathcal{T}_k \mathbf{u}_D(0) + m \eta_1^2 \mathbf{u}_D(0)^\top \mathcal{T}_k^\top \mathbf{H}_L^\infty \mathcal{T}_k \mathbf{u}_D(0) \\
&\leq m \eta_1^2 \left\| \mathbf{H}(0) - \mathbf{H}_L^\infty \right\|_2 \cdot \mathbf{u}_D(0)^\top \mathcal{T}_k^2 \mathbf{u}_D(0) + m \eta_1^2 \mathbf{u}_D(0)^\top \mathcal{T}_k^\top \mathbf{H}_L^\infty \mathcal{T}_k \mathbf{u}_D(0) \\
&\leq m \eta_1^2 \frac{\left(1 - \eta_2 \mu\right)^{2k}}{m^2 \eta_1^2 \lambda_\infty^2} \mathcal{O}\left( \omega n L^{5/2} \sqrt[4]{\frac{\log(nL/\delta)}{m}} \right) \left\| \mathbf{u}_D(0) \right\|_2^2 \\
&\quad + m \eta_1^2 \frac{\left(1 - \eta_2 \mu\right)^{2k}}{m^2 \eta_1^2} \mathbf{u}_D(0)^\top \left( \mathbf{H}_L^\infty \right)^{-1} \mathbf{u}_D(0) \\
&\leq \mathcal{O}\left( \frac{\left(1 - \eta_2 \mu\right)^{2k} n^2 \omega^2 L^{5/2}}{m \lambda_\infty^2 \delta^2} \sqrt[4]{\frac{\log(nL/\delta)}{m}} + \frac{n \omega \left(1 - \eta_2 \mu\right)^{2k}}{m \lambda_\infty \delta^2} \right) \\
&= \mathcal{O}_{\mathbb{P}}\left( \frac{n \omega \left(1 - \eta_2 \mu\right)^{2k}}{m \lambda_\infty} \right).
\end{aligned}
\tag{107}
$$

Here, in the second inequality, we used the inequalities (102) and (103) and Lemma 4.11. In the third inequality, we used the Lemma 4.8, $\|\mathbf{u}(0)\|_2 = \mathcal{O}\left(\frac{\sqrt{n\omega}}{\delta}\right)$ with probability at least $1 - \delta$. In the last equality, we used $m \geq \Omega\left( L^{19} n^{20} \frac{\log^3(m)}{\delta^2} \right)$.

Next, we bound $\sum_{\ell=1}^{L} \|\mathbf{E}_{2,3}\|_2^2$ as:

$$
\sum_{\ell=1}^{L} \|\mathbf{E}_{2,3}\|_2^2 = m\eta_1^2 \cdot \left( \sum_{j=0}^{k-1} \left(1-\eta_2\mu\right)^j \left(\mathbf{e}_{k-j-1}\right) \right)^{\top} \mathbf{H}_D(0) \left( \sum_{j=0}^{k-1} \left(1-\eta_2\mu\right)^j \left(\mathbf{e}_{k-j-1}\right) \right)
$$

$$
\leq \frac{m\eta_1^2}{\eta_2^2\mu^2} \cdot \lambda_{\max}\left(\mathbf{H}_D(k)\right) \cdot \|\mathbf{e}_{k-j-1}\|_2^2 \leq \frac{m\eta_1^2}{\eta_2^2\mu^2} \cdot \mathcal{O}(\omega n L) \cdot \mathcal{O}\left(\frac{1}{n^2}\right) = \mathcal{O}\left(\frac{L^3}{m}\omega \cdot n^{-\frac{4d-3}{2d-1}}\right).
$$

$$(108)$$

Now, we focus on obtaining the $\ell_2$ norm bound on $\mathcal{D}$ in (100). Recall the definition of the notation $\mathbf{B} := \mathbf{H}_L^{\infty}\left(\frac{\eta_2\mu L}{\eta_1 m}\mathcal{I} + \mathbf{H}_L^{\infty}\right)^{-1}\mathbf{y}$. A simple calculation yields that

$$
\mathbf{B} - \mathbf{y} = \mathbf{H}_L^{\infty}\left(\frac{\eta_2\mu L}{\eta_1 m}\mathcal{I} + \mathbf{H}_L^{\infty}\right)^{-1}\mathbf{y} - \mathbf{y} = -\frac{\eta_2\mu L}{m\eta_1}\left(\frac{\eta_2\mu L}{\eta_1 m}\mathcal{I} + \mathbf{H}_L^{\infty}\right)^{-1}\mathbf{y}.
$$

Then, we can re-write the expression of the $\mathcal{D}$ as :

$$
\mathcal{D} := \left(\frac{\eta_2\mu L}{m\eta_1}\right) \cdot \eta_1 \frac{1}{L} \sum_{\ell=1}^{L} \left\langle z_{D,k}(x), \nabla_{\mathbf{W}_\ell}\left[f_{\mathbf{W}_D(0)}(\mathbf{X})\right]\right\rangle \sum_{j=0}^{k-1}\left(1-\eta_2\mu\right)^j\left(\frac{\eta_2\mu L}{\eta_1 m}\mathcal{I} + \mathbf{H}_L^{\infty}\right)^{-1}\mathbf{y}
$$

$$
- \mathbf{Ker}(x,\mathbf{X})\left(\frac{\eta_2\mu L}{\eta_1 m}\mathcal{I} + \mathbf{H}_L^{\infty}\right)^{-1}\mathbf{y}
$$

$$
= \left(\frac{1}{m}\sum_{\ell=1}^{L}\left\langle z_{D,k}(x), \nabla_{\mathbf{W}_\ell}\left[f_{\mathbf{W}_D(0)}(\mathbf{X})\right]\right\rangle - \mathbf{Ker}(x,\mathbf{X})\right)\left(\frac{\eta_2\mu L}{\eta_1 m}\mathcal{I} + \mathbf{H}_L^{\infty}\right)^{-1}\mathbf{y}
$$

$$
- \left(1-\eta_2\mu\right)^k\frac{1}{m}\sum_{\ell=1}^{L}\left\langle z_{D,k}(x), \nabla_{\mathbf{W}_\ell}\left[f_{\mathbf{W}_D(0)}(\mathbf{X})\right]\right\rangle\left(\frac{\eta_2\mu L}{\eta_1 m}\mathcal{I} + \mathbf{H}_L^{\infty}\right)^{-1}\mathbf{y}
$$

$$
= \left(\frac{1}{m}\sum_{\ell=1}^{L}\left\langle z_{0,k}(x), \nabla_{\mathbf{W}_\ell}\left[f_{\mathbf{W}_D(0)}(\mathbf{X})\right]\right\rangle - \mathbf{Ker}(x,\mathbf{X})\right)\left(\frac{\eta_2\mu L}{\eta_1 m}\mathcal{I} + \mathbf{H}_L^{\infty}\right)^{-1}\mathbf{y}
$$

$$
- \left(1-\eta_2\mu\right)^k\frac{1}{m}\sum_{\ell=1}^{L}\left\langle z_{0,k}(x), \nabla_{\mathbf{W}_\ell}\left[f_{\mathbf{W}_D(0)}(\mathbf{X})\right]\right\rangle\left(\frac{\eta_2\mu L}{\eta_1 m}\mathcal{I} + \mathbf{H}_L^{\infty}\right)^{-1}\mathbf{y}
$$

$$
+ \left(1-\left(1-\eta_2\mu\right)^k\right)\left(\frac{1}{m}\sum_{\ell=1}^{L}\left\langle z_{D,k}(x)-z_{D,0}(x), \nabla_{\mathbf{W}_\ell}\left[f_{\mathbf{W}_D(0)}(\mathbf{X})\right]\right\rangle\right)\left(\frac{\eta_2\mu L}{\eta_1 m}\mathcal{I} + \mathbf{H}_L^{\infty}\right)^{-1}\mathbf{y},
$$

$$(109)$$

where in the second equality, $\sum_{j=0}^{k-1}(1-\eta_2\mu)^j = \frac{1-(1-\eta_2\mu)^k}{\eta_2\mu}$ is used. The $\ell_2$ norm of first term in the (109) can be bounded as:

$$
\left\|\left(\frac{1}{m}\sum_{\ell=1}^{L}\left\langle z_{D,k}(x), \nabla_{\mathbf{W}_\ell}\left[f_{\mathbf{W}_D(0)}(\mathbf{X})\right]\right\rangle - \mathbf{Ker}(x,\mathbf{X})\right)\left(\frac{\eta_2\mu L}{\eta_1 m}\mathcal{I} + \mathbf{H}_L^{\infty}\right)^{-1}\mathbf{y}\right\|_2
$$

$$
\leq \left\|\left(\frac{1}{m}\sum_{\ell=1}^{L}\left\langle z_{D,k}(x), \nabla_{\mathbf{W}_\ell}\left[f_{\mathbf{W}_D(0)}(\mathbf{X})\right]\right\rangle - \mathbf{Ker}(x,\mathbf{X})\right)\right\|_2 \cdot \left\|\left(\frac{\eta_2\mu L}{\eta_1 m}\mathcal{I} + \mathbf{H}_L^{\infty}\right)^{-1}\mathbf{y}\right\|_2
$$

$$
= \sqrt{\sum_{i=1}^{n}\left(\frac{1}{m}\sum_{\ell=1}^{L}\left\langle z_0(x), \nabla_{\mathbf{W}_\ell}\left[f_{\mathbf{W}_D(0)}(\mathbf{x}_i)\right]\right\rangle - \mathbf{Ker}(x,\mathbf{x}_i)\right)^2} \cdot \left\|\left(\frac{\eta_2\mu L}{\eta_1 m}\mathcal{I} + \mathbf{H}_L^{\infty}\right)^{-1}\mathbf{y}\right\|_2
$$

$$
\leq \mathcal{O}\left(\omega\sqrt{n}L^{5/2}\sqrt[4]{\frac{\log(nL/\delta)}{m}}\right) \cdot \mathcal{O}\left(\frac{\eta_1 m}{\eta_2\mu L}\sqrt{n}\right)
$$

$$
= \mathcal{O}\left(\frac{\omega\eta_1 mn}{\eta_2\mu L}L^{5/2}\sqrt[4]{\frac{\log(nL/\delta)}{m}}\right) = \mathcal{O}\left(\omega L^{5/2}n^{\frac{d}{2d-1}}\sqrt[4]{\frac{\log(nL/\delta)}{m}}\right),
$$

$$(110)$$

where, in the second inequality, we used Lemma 4.11, and also we used

$$\left\|\left(\frac{\eta_2\mu L}{\eta_1 m}\mathcal{I}+\mathbf{H}_L^\infty\right)^{-1}\mathbf{y}\right\|_2 \le \sqrt{\mathbf{y}^\top\left(\frac{\eta_2\mu L}{\eta_1 m}\mathcal{I}+\mathbf{H}_L^\infty\right)^{-2}\mathbf{y}} \le \sqrt{\frac{\eta_1^2 m^2}{\eta_2^2\mu^2 L^2}\cdot\|\mathbf{y}\|_2^2} = \mathcal{O}\left(\frac{\eta_1 m}{\eta_2\mu L}\sqrt{n}\right).$$

(111)

The $\ell_2$ norm of the second term in (109) can be easily bounded as:

$$\left\|(1-\eta_2\mu)^k\frac{1}{m}\sum_{\ell=1}^{L}\big\langle z_0(x),\nabla_{\mathbf{W}_\ell}[f_{\mathbf{W}_D(0)}(\mathbf{X})]\big\rangle\left(\frac{\eta_2\mu L}{\eta_1 m}\mathcal{I}+\mathbf{H}_L^\infty\right)^{-1}\mathbf{y}\right\|_2$$

$$\le \left\|(1-\eta_2\mu)^k\left(\frac{1}{m}\sum_{\ell=1}^{L}\big\langle z_0(x),\nabla_{\mathbf{W}_\ell}[f_{\mathbf{W}_D(0)}(\mathbf{X})]\big\rangle-\mathbf{Ker}(x,\mathbf{X})\right)\left(\frac{\eta_2\mu L}{\eta_1 m}\mathcal{I}+\mathbf{H}_L^\infty\right)^{-1}\mathbf{y}\right\|_2$$

$$+ \left\|(1-\eta_2\mu)^k\mathbf{Ker}(x,\mathbf{X})\left(\frac{\eta_2\mu L}{\eta_1 m}\mathcal{I}+\mathbf{H}_L^\infty\right)^{-1}\mathbf{y}\right\|_2$$

$$\le (1-\eta_2\mu)^k\left\|\frac{1}{m}\sum_{\ell=1}^{L}\big\langle z_0(x),\nabla_{\mathbf{W}_\ell}[f_{\mathbf{W}_D(0)}(\mathbf{X})]\big\rangle-\mathbf{Ker}(x,\mathbf{X})\right\|_2\cdot\left\|\left(\frac{\eta_2\mu L}{\eta_1 m}\mathcal{I}+\mathbf{H}_L^\infty\right)^{-1}\mathbf{y}\right\|_2$$

$$+ (1-\eta_2\mu)^k\left\|\mathbf{Ker}(x,\mathbf{X})\left(\frac{\eta_2\mu L}{\eta_1 m}\mathcal{I}+\mathbf{H}_L^\infty\right)^{-1}\mathbf{y}\right\|_2$$

$$\le (1-\eta_2\mu)^k\cdot\mathcal{O}\left(\omega\sqrt{n}L^{3/2}\sqrt[4]{\frac{\log(nL/\delta)}{m}}\right)\cdot\mathcal{O}\left(\frac{\eta_1 m}{\eta_2\mu L}\sqrt{n}\right)+\mathcal{O}\left((1-\eta_2\mu)^k\right)$$

$$\le (1-\eta_2\mu)^k\cdot\mathcal{O}\left(\omega L^{3/2}n^{\frac{d}{2d-1}}\sqrt[4]{\frac{\log(nL/\delta)}{m}}\right)+\mathcal{O}\left((1-\eta_2\mu)^k\right).$$

(112)

Lastly, the $\ell_2$ norm of the third term in (109) is bounded as:

$$\left\|(1-(1-\eta_2\mu)^k)\left(\frac{1}{m}\sum_{\ell=1}^{L}\big\langle z_{D,k}(x)-z_{D,0}(x),\nabla_{\mathbf{W}_\ell}[f_{\mathbf{W}_D(0)}(\mathbf{X})]\big\rangle\right)\left(\frac{\eta_2\mu L}{\eta_1 m}\mathcal{I}+\mathbf{H}_L^\infty\right)^{-1}\mathbf{y}\right\|_2$$

$$\le (1-(1-\eta_2\mu)^k)\cdot\left\|\frac{1}{m}\sum_{\ell=1}^{L}\big\langle z_{D,k}(x)-z_{D,0}(x),\nabla_{\mathbf{W}_\ell}[f_{\mathbf{W}_D(0)}(\mathbf{X})]\big\rangle\right\|_2\cdot\left\|\left(\frac{\eta_2\mu L}{\eta_1 m}\mathcal{I}+\mathbf{H}_L^\infty\right)^{-1}\mathbf{y}\right\|_2$$

$$\le (1-(1-\eta_2\mu)^k)\cdot\left(\frac{1}{m}\sum_{\ell=1}^{L}\|z_{D,k}(x)-z_{D,0}(x)\|_F\|\nabla_{\mathbf{W}_\ell}[f_{\mathbf{W}_D(0)}(\mathbf{X})]\|_F\right)\cdot\left\|\left(\frac{\eta_2\mu L}{\eta_1 m}\mathcal{I}+\mathbf{H}_L^\infty\right)^{-1}\mathbf{y}\right\|_2$$

$$\le (1-(1-\eta_2\mu)^k)\cdot\left(\frac{L}{m}\mathcal{O}(\tau^{1/3}L^2\sqrt{\omega m\log(m)})\cdot\mathcal{O}(\sqrt{\omega mn})\right)\cdot\mathcal{O}\left(\frac{\eta_1 m}{\eta_2\mu L}\sqrt{n}\right)$$

$$\le (1-(1-\eta_2\mu)^k)\cdot\mathcal{O}\left(\omega^{7/6}L^{10/3}n^{\frac{4d}{6d-3}}\frac{\sqrt{\log(m)}}{m^{1/6}\delta^{1/3}}\right)\le\mathcal{O}\left(\omega^{7/6}L^{10/3}n^{\frac{4d}{6d-3}}\frac{\sqrt{\log(m)}}{m^{1/6}\delta^{1/3}}\right),$$

(113)

where in the fourth inequality, $\tau = \mathcal{O}_\mathbb{P}\left(\frac{L\sqrt{\omega}}{\sqrt{m}}n^{\frac{d}{2d-1}}\right)$ is plugged in. Combining the inequalities (110), (112) and (113), we get the bound on $\|\mathcal{D}\|_2$ in (109):

$$\|\mathcal{D}\|_2^2 \le \mathcal{O}\left(\omega^2 L^5 n^{\frac{2d}{2d-1}}\sqrt{\frac{\log(nL/\delta)}{m}}\right)+(1-\eta_2\mu)^{2k}\mathcal{O}\left(\omega^2 L^3 n^{\frac{2d}{2d-1}}\sqrt{\frac{\log(nL/\delta)}{m}}\right)+\mathcal{O}\left((1-\eta_2\mu)^{2k}\right)$$

$$+ \mathcal{O}\left(\omega^{7/3}L^{20/3}n^{\frac{8d}{6d-3}}\frac{\log(m)}{m^{1/3}\delta^{2/3}}\right)$$

$$\le \mathcal{O}\left(\omega^{7/3}L^{20/3}n^{\frac{8d}{6d-3}}\frac{\log(m)}{m^{1/3}\delta^{2/3}}\right)+\mathcal{O}\left((1-\eta_2\mu)^{2k}\right).$$

(114)

**Step 3. Combining all pieces.** Recall $\|z_{D,k}(x)\|_2 \leq \mathcal{O}(\sqrt{m\omega})$. With this fact, combining the bounds (94), (97), (106), (107), (108) and (114), we can bound the $\|\Delta_{D,1}\|_2^2$ via the decomposition (92) as follows:

$$
\begin{aligned}
\|\Delta_{D,1}\|_2^2 &\leq \frac{1}{L} \sum_{\ell=1}^{L} \left| z_{D,k}(x)^\top \mathbf{vec}\big(\mathbf{W}_{D,\ell}^{(0)}\big) \right|^2 + \frac{1}{L} \sum_{\ell=1}^{L} \|z_{D,k}(x)\|_2^2 \|\mathbf{E}_3\|_2^2 \\
&\quad + \frac{1}{L} \sum_{\ell=1}^{L} \|z_{D,k}(x)\|_2^2 \|\mathbf{E}_{2,1}\|_2^2 + \frac{1}{L} \sum_{\ell=1}^{L} \|z_{D,k}(x)\|_2^2 \|\mathbf{E}_{2,2}\|_2^2 + \frac{1}{L} \sum_{\ell=1}^{L} \|z_{D,k}(x)\|_2^2 \|\mathbf{E}_{2.3}\|_2^2 \\
&\quad + \left\| \frac{1}{L} \sum_{\ell=1}^{L} \langle z_{D,k}(x), \mathbf{E}_{2,4} \rangle - \mathbf{Ker}(x, \mathbf{X}) \left( \frac{\eta_2 \mu L}{\eta_1 m} \mathcal{I} + \mathbf{H}_L^\infty \right)^{-1} \mathbf{y} \right\|_2^2 \\
&\leq \mathcal{O}\left( \omega \log\left( \frac{L}{\delta} \right) \right) + \mathcal{O}_{\mathbb{P}}\left( \frac{L^{20/3} \omega^{4/3}}{m^{1/3}} n^{\frac{8d}{6d-3}} \log(m) \right) \\
&\quad + \mathcal{O}_{\mathbb{P}}\left( \frac{\omega(1 - \eta_2 \mu)^{2k}}{L} \right) + \mathcal{O}_{\mathbb{P}}\left( \frac{n\omega^2 (1 - \eta_2 \mu)^{2k}}{L\lambda_\infty} \right) + \mathcal{O}_{\mathbb{P}}\left( \frac{L^2}{m} \omega \cdot n^{-\frac{4d-3}{2d-1}} \right) \\
&\quad + \mathcal{O}\left( \omega^{7/3} L^{20/3} n^{\frac{8d}{6d-3}} \frac{\log(m)}{m^{1/3} \delta^{2/3}} \right) + \mathcal{O}_{\mathbb{P}}\left( (1 - \eta_2 \mu)^{2k} \right) \\
&\leq \mathcal{O}_{\mathbb{P}}\left( n^{-\frac{d}{2d-1}} \right).
\end{aligned}
$$

