# OpenReview forum: "A NON-PARAMETRIC REGRESSION VIEWPOINT : GENERALIZATION OF OVERPARAMETRIZED DEEP RELU NETWORK UNDER NOISY OBSERVATIONS"
_ICLR.cc/2022/Conference — ICLR 2022 Poster_

### Official Review · Reviewer_XydK · 2021-10-30

**Correctness:** 3
**Technical Novelty And Significance:** 2
**Empirical Novelty And Significance:** 2
**Recommendation:** 6
**Confidence:** 3

**Main Review:**

Strengths:
- Theoretical understanding of neural networks, and in particular, the understanding of multi-layer NN, is an important direction of research for advancing deep learning; so while I'm not following the latest literature, I believe this paper has merit along this ongoing line of research; specifically, the line of research leveraging the NTK (Neural Tangent Kernel, i.e., linearized) representation of a multi-layer NN
- The paper has mainly made technically incremental improvements along existing works, but they have made clear what is their unique contribution and has made good effort in citing relevant work and making comparisons.

Weaknesses:
- While I would like to trust the technical quality of the current submission, I feel the authors can spend more time on this:
   - perhaps the authors can summarize the intuition behind each theorem/proof;
   - perhaps the authors should re-check correctness of their statements/results: In all of Theorem 3.4, Theorem 3.6, Theorem 3.8, and Theorem 3.9, the network width m has a dependence on itself (log m); is this a typo?
- The practical interpretability of their result is limited: The bound on the network width seems unrealistically high (e.g., n^24, L^20); how does it compare to existing work?
- Experiments are not very convincing: in the middle plot of Figure 1, it seems that the best un-regularized generalization prediction risk is lower than (or at least on par with) the best regularized one, and that the best result of un-regularized generalization error is achieved with very few epochs. How does this experiment support the result of Theorem 3.6?

**Summary Of The Paper:**

The paper made technical contribution along the recent line of theoretical work on DNN: It proves that when data is generated from a nonparametric regression model with i.i.d. Gaussian noise, training (over-parametrized, i.e., wide) ReLU DNN with a regularization term with GD approximately converges, and furthermore that the generalization error decays in order faster than O(1/n^{1/2}), where n is sample size.

**Summary Of The Review:**

While the contribution of the paper seems a bit incremental and I am not a hundred percent confident about the technical quality of the paper, I think the paper works on an important topic and has merit to be accepted.

---

> ### Author Response · Authors · 2021-11-14
> **Responses to Reviewer 4**
>
> ## **Q1**
> perhaps the authors can summarize the intuition behind each theorem/proof;
> perhaps the authors should re-check correctness of their statements/results: In all of Theorem 3.4, Theorem 3.6, Theorem 3.8, and Theorem 3.9, the network width m has a dependence on itself $\log(m)$; is this a typo?
>
> ## **A1**
> As for the answer to the first comment, we will add proof sketch of each theorem in our revised manuscript.
> Due to the page limit, we will add each of these to the beginning of the proofs in the Appendix!
>
> As for the answer to the second comment of Reviewer $4$, this is not a typo!
> One can understand this condition as $\frac{m}{\log^{3}(m)}\geq\Omega\big( \frac{\omega^{7} n^{8} L^{18} }{\lambda_{0}^{8}\delta^{2}} \big)$ in Theorem $3.4$ and $3.6$,
> and $\frac{m}{\log^{3}(m)}\geq\Omega\big(\frac{L^{20}n^{24}}{\delta^{2}}\big)$ in Theorem $3.8$ and $3.9$.
> This condition on $m$ is quite similar with that in [1].
> See page $33$ of [1].
> But this is a reasonable concern, so we will add a footnote or additional explanation on this condition!
> Thank you for pointing out!
>
> [1] Z. Allen-Zhu, Y. Li, Z. Song. A Convergence Theory for Deep Learning via Over-Parameterization, arXiv, 2019.
>
> ---
> ## **Q2**
> The practical interpretability of their result is limited: The bound on the network width seems unrealistically high (e.g., $n^{24}$, $L^{20}$); how does it compare to existing work?
>
> ## **A2**
> Reducing the order of network width is definitely another line of interesting research direction.
> We are aware of some works in literature, but we chose not to adopt the techniques since this can make the analysis overly complicated.
> To the best of our knowledge, the paper that most neatly summarizes this line of literature is [2].
> See the table in page $3$ in their paper.
> The order of width they obtained is $\Omega\big(\frac{n^{8}L^{12}}{\phi^{8}}\big)$, where they impose $\phi$-separateness assumption.
>
> [2] Zou, D., Gu, Q. (2019). An improved analysis of training over-parameterized deep neural networks. arXiv preprint arXiv:1906.04688.
>
> ---
> ## **Q3**
> Experiments are not very convincing: in the middle plot of Figure 1, it seems that the best un-regularized generalization prediction risk is lower than (or at least on par with) the best regularized one, and that the best result of un-regularized generalization error is achieved with very few epochs. How does this experiment support the result of Theorem $3.6$?
>
> ## **A3**
> In Theorem $3.6$, our theorem tells us that for the iteration less than the order $\mathcal{O}\big(\frac{1}{\eta m \omega L}\big)$, the prediction error is bounded away from $0$.
> In the experiment for unregularized case, we set $\eta=0.01$, $m=2000$, $L=8$, and $\omega=1$.
> Plugging in these parameters in the bound says that the minimum can be achieved within a very few iterations.
>
> However, we have to acknowledge that there is a discrepancy between our experiment setting and theory.
> Specifically, due to the limited computing power, we could not run the experiment under the regime of width $\frac{m}{\log^{3}(m)}\geq\Omega\big( \frac{\omega^{7} n^{8} L^{18} }{\lambda_{0}^{8}\delta^{2}} \big)$.
> But the prediction risk behaves similarly as expected by our theorem, which can be a partial evidence that the statement in Theorem $3.6$ still holds in the narrower width of the network.
> Most likely, we can implement this by employing the idea presented in the paper [2].
> We feel that it would be great if we can add these comments in the revised manuscript!
> Thank you!
> ---

---

> > ### Comment · Reviewer_XydK · 2021-11-24
> > **Response to authors' feedback**
> >
> > We thank the authors for your feedback. Admittedly, we didn't read the paper very carefully/thoroughly during the review and realized that we made some typos in the first round of review comments. After reading the authors' response, our review score remains the same.

---

### Official Review · Reviewer_ytWV · 2021-11-01

**Correctness:** 3
**Technical Novelty And Significance:** 2
**Empirical Novelty And Significance:** Not applicable
**Recommendation:** 6
**Confidence:** 4

**Main Review:**

The paper addresses interesting and relevant topic. Namely, it is one of few papers which thinks of a neural network learning as a nonparametric problem. This is justifiable as learning is always overparameterized. The question is whether such learning procedure can recover complex regression functions in a presence of label noise, as typically considered in the nonparametric literature. To this end, the paper embraces an NTK point of view and considers regression functions which belong to a corresponding Hilbert space. The paper gives a positive answer to this question and moreover shows that this can be done with an optimal rate. The proof idea is also straightforward: It is well-known that with a right tuning, ridge regression enjoys an optimal rate and so all that is left to do is to establish that deep ReLU networks trained by GD (with regularization) "track" predictions of the ridge regressor: This is done in Theorem 3.8.


The paper can be considered as an extension of Hu et al. 2021, who considered a very similar setting albeit with a shallow network.

Technical details:
* It is known that interpolants are not consistent in general, e.g. see [1]. It is not clear what theorem 3.6 adds to that.
* The paper highlights that one of the main technical novelties is a control of the distance parameters travel from initialization, which is $\\|W^{(k)} - W^{(0)}\\| \\leq \tilde{O}_P(1/\\sqrt{m})$ vs $\\|W^{(k)} - W^{(0)}\\| \\leq \\tilde{O}_P(1)$ (where $m$ is the width).
I guess this should be $\\|W^{(k)} - W^{(0)}\\| \\leq \tilde{O}_P(n/(\\sqrt{m} \\lambda_0))$ vs $\\|W^{(k)} - W^{(0)}\\| \\leq \\tilde{O}_P(n/ \\lambda_0)$ (sample size and the eigenvalue are relevant here)? It seems that $1/\\sqrt{m}$ comes from the factor $\\sqrt{m}$ which multiplies the neural network, this makes its way to Part 5 of the proof of Theorem 3.8 (explanation on page 3 seems to be a bit unclear about that).
One detail which is unclear, and would be nice if authors could clarify, how is the factor $\\sqrt{m}$ "balanced-out" in the part of the proof where you need to track the ridge regressor (second result of Theorem 3.8): It seems to be important because both predictors have to be of the same scale. Is it done by having an extremely small step size of order $1/m$? In most NTK-style proofs this factor is not present.
* Remark 3.5. $\lambda_0 = \Omega(\phi n^{-2})$ is either extremely loose, or there's a mismatch in normalization. Lower bounds on $\lambda_0 = \Omega(d)$ are known under certain assumptions on the input distribution. See, e.g. [4].
* It is not very surprising that networks can indeed learn regression functions in RKHS of NTK, because such regression functions are essentially by design what networks learn in an extremely overparameterized regime. However, a somewhat more interesting detail is what kind of functions are representable there? In other words, is this RKHS small?
There are some works discussing this and it would be interesting if the paper could put them in perspective.


Some recent relevant references about nonparametric regression with GD+neural networks not present in the work: [1,2,3,4].









[1] M. Kohler and A. Krzyzak. Over-parametrized deep neural networks do not generalize well.arXivpreprint arXiv:1912.03925, 2019.

[2] Z. Ji, J. D. Li, and M. Telgarsky. Early-stopped neural networks are consistent, NeurIPS 2021.

[3] I. Kuzborskij and C. Szepesvari. Nonparametric Regression with Shallow Overparameterized Neural Networks Trained by GD with Early Stopping, COLT 2021.

[4] P. L. Bartlett, A. Montanari, and A. Rakhlin. Deep learning: a statistical viewpoint. Acta Numerica, 2021.

**Summary Of The Paper:**

The paper proves a bound on the excess risk in a nonparametric regression setting, where the estimator is a deep ReLU network trained by GD with L2 regularization, and the regression function belongs to the RKHS induced by the Neural Tangent Kernel (NTK) related to a deep network.
In particular, this work considers presence of label noise and argues that without an explicit regularization such learning procedure is not consistent in general as the network reaches interpolating regime. The paper proposes to use a relative L2 regularization (relative to the initialization) and shows that with an appropriate tuning of parameters the procedure is consistent and yields an optimal rate for regression on RKHS. The proof is heavily based on the NTK machinery, and in particular on the "coupling" between predictions of the network and a ridge regressor given the NTK kernel matrix.

**Summary Of The Review:**

The paper proves a bound on the excess risk in a nonparametric regression setting, where the estimator is a deep ReLU network trained by GD with L2 regularization, and the regression function belongs to the RKHS induced by the Neural Tangent Kernel (NTK) related to a deep network. This is an incremental contribution w.r.t. Hu et al. 2021. Some technical parts necessary for the proof come from cited works. In this respect this work is also quite incremental. However, I believe that contribution points in a relevant direction of understanding neural nets from nonparametric point of view. I'd be happy to raise the score if authors clarify some details.

---

> ### Author Response · Authors · 2021-11-14
> **Responses to reviewer 3 (part 1)**
>
> # **Q1**
> It is known that interpolants are not consistent in general, e.g. see [1]. It is not clear what theorem 3.6 adds to that.
>
> # **A1**
> Thank you for pointing out this work!
> This work is relevant with our paper in terms of topic.
> Specifically, it shows the interpolants obtained through deep neural network by GD is inconsistent.
> Based on our reading, in terms of architectural structure of the network, our paper is more general in a sense that
> [1] considers the overparametrized DNN that is a linear combination of $\Omega(n^{10d^{2}})$ smaller neural network,
> and the activation function they consider is sigmoid function, which is smooth and differentiable.
> Lastly the function class they consider is H\"older class, which is also different from ours.\
> \
> However, it is worth pointing out that this paper shows that when GD achieves $o(1/n)$ empirical risk for any $n$ large enough, then
> there exists a sufficiently regular (smooth) joint probability measure $\rho$, even when the marginal distribution of feature data $\rho_{\mathbf{x}}$ is an atomic distribution.
> See Corollary $1$ of this paper.
> On the other hand, our paper restricts to the case when $\mathbf{x}\in\textbf{Unif}\big(\mathcal{S}^{d-1}\big)$.
> ---
>
> # **Q3**
> Remark 3.5. $\lambda_{0}=\Omega\big(\phi n^{-2}\big)$ is either extremely loose, or there's a mismatch in normalization. Lower bounds on $\lambda_{0}=\Omega(d)$ are known under certain assumptions on the input distribution. See, e.g. [4].
>
> # **A3**
> We apologize for the confusion, but in Remark $3.5.$, the main point of the conjecture that we are trying to make is $\phi$-separateness assumption and positivity assumption of NTK can be exchangeable in some sense.
> We want to note that we are not claiming that $\lambda_{0}=\Omega\big(\phi n^{-2}\big)$ is the tightest bound we can get.
> From our readings on Lemma $5.3$ of the paper [4], the lower bound for $\lambda_{0} = \Omega(d/n)$, and $\lambda_{0}=\Omega\big(\phi n^{-2}\big)$ is still loose bound.
> But the result from paper [4] is from the shallow-neural network setting, and to the best of our knowledge, there is neither rigorous proof on the lower-bound of $\lambda_{0}$ under DNN setting nor formula for describing the relations between $\phi$-separateness assumption and postivity assumption of NTK.
> We think this type of explanation is lacking in Remark 3.5, and will add more explanations in this regards.
> ---
>
> # **Q4**
> It is not very surprising that networks can indeed learn regression functions in RKHS of NTK, because such regression functions are essentially by design what networks learn in an extremely overparameterized regime. However, a somewhat more interesting detail is what kind of functions are representable there? In other words, is this RKHS small? There are some works discussing this and it would be interesting if the paper could put them in perspective.
>
> # **A4**
> Yes, it is definitely an interesting and important question on investigating the size of NTK-induced RKHS, since the minimax rate we can get is directly related with complexity of function class of interest.
> As one of the reviewer $3$'s suggested references, the paper [3] mentioned that RKHS-induced by shallow ReLU neural net can represent all even functions on the $\mathcal{S}^{d-1}$ when at least its first $\ceil{\frac{d}{2}}$ derivatives are bounded.
> In addition to this, it has recently been shown by several researchers that the set of functions in RKHSs induced by shallow ReLU neural net and deep ReLU Neural net are equivalent, meaning that the two function classes are same.
> See the papers [1], [2], [3].
>
> [1] Lin Chen and Sheng Xu. Deep neural tangent kernel and Laplace kernel have the same RKHS. In International Conference on Learning Representations, 2020.
>
> [2] Amnon Geifman, Abhay Yadav, Yoni Kasten, Meirav Galun, David Jacobs, and Ronen Basri.  Onthe similarity between the Laplace and neural tangent kernels. arXiv preprint arXiv: 2007.01580,2020.
>
> [3] Alberto Bietti and Francis Bach. Deep equals shallow for ReLU networks in kernel regimes. In ICLR 2021-International Conference on Learning Representations.
>
> Specifically, above papers showed theoretically that for large enough harmonic function frequency $k$, the decay rate of the eigenvalues $\mu_{k}$s of the PSD function $\mathbf{Ker}(\mathbf{x},\mathbf{x'})$ (equation $(8)$ in our paper) is in the order of $\Theta\big( k^{-d} \big)$, which turns out to be same decay rate with RKHS induced from Laplace Kernel.
> We had mentioned about this comment in the Appendix $A.1.$ with more detailed explanations in our original submission, but we decide to move this section to the main manuscript.
> ---
>
> # **Q5**
> Some recent relevant references about nonparametric regression with GD+neural networks not present in the work: [1,2,3,4].
> (Due to the space limit, we didn't rewrite the suggested papers!)
>
> # **A5**
> Thank you for the recommendations!
> We read these literature and include them in our literature review section!

---

> > ### Author Response · Authors · 2021-11-14
> > **Responses to reviewer 3 (part 2)**
> >
> > We divide the answers to the 2nd question into two parts!
> >
> > # **Q2 (1)**
> > The paper highlights that one of the main technical novelties is a control of the distance parameters travel from initialization, which is
> > $\|\| W_{D,\ell}^{(k)}-W_{D,\ell}^{(0)} \|\|\leq\mathcal{O}(\frac{1}{\sqrt{m}})$ vs $\|\| W_{D,\ell}^{(k)}-W_{D,\ell}^{(0)} \|\|\leq\mathcal{O}(1)$
> > (where $m$ is the width). I guess this should be
> > $\|\| W_{D,\ell}^{(k)}-W_{D,\ell}^{(0)} \|\|\leq\mathcal{O}(\frac{n}{\sqrt{m}\lambda_0})$ vs $\|\| W_{D,\ell}^{(k)}-W_{D,\ell}^{(0)} \|\|\leq\mathcal{O}(\frac{n}{\lambda_0})$
> > (sample size and the eigenvalue are relevant here)? It seems that  comes from the factor  which multiplies the neural network, this makes its way to Part 5 of the proof of Theorem 3.8 (explanation on page 3 seems to be a bit unclear about that).
> >
> > # **A2 (1)**
> > We acknowledge that this type of confusion can come from our writings on page $3$.
> > For the regularized case relative to the initialization, we did prove the result $\|\|W_{D,\ell}^{(k)}-W_{D,\ell}^{(0)}\|\|_{2} \leq \mathcal{O}(\frac{L\sqrt{\omega}}{\sqrt{m}} n^{\frac{d}{2d-1}})$
> >
> > through induction with proper choices of tuning parameters.
> > The proof is presented in page $33$ of our manuscript.
> > Although the $\sqrt{m}$ factor might have an effect on the bound, but in fact the reasoning is slightly more complicated than that.\
> > \
> > This can be more clear if we heuristically prove the case when $\|\| W_{D,\ell}^{(k)}-W_{D,\ell}^{(0)} \|\|_{2}\leq\mathcal{O}(1)$,
> >
> > where $W_{D,\ell}^{(k)}$ is the model parameter of $\ell$th layer in $k$th iteration of algorithm.
> > Here, we regularize solely on the model parameter, instead on the relative to the initialization.
> > In this case, we can write the update rule as follows :
> >
> > $$
> > W_{D,\ell}^{(k)} = (1-\eta_{2}\mu)W_{D,\ell}^{(k-1)}  - \eta_{1}\nabla_{W_{\ell}}[ L_{S}(W_{D}^{(k-1)} ],
> > \qquad 1\leq \ell \leq L , \qquad k \geq 1.
> > $$
> >
> > By recursively applying above equation $(4.3)$, we can write $W_{D,\ell}^{(k)}$ with respect to $W_{D,\ell}^{(0)}$ as follows:
> >
> > $$
> > W_{D,\ell}^{(k)} = (1-\eta_{2}\mu)^{k} W_{D,\ell}^{(0)} -\eta_{1} \sum_{\ell=0}^{k-1}(1-\eta_{2}\mu)^{\ell} \nabla_{W_{\ell}}[ L_{S}(W_{D}^{(k-\ell-1)} )].
> > $$
> >
> > Then, we can control the bound as follows:
> >
> > $$
> > \|\| W_{D,\ell}^{(k)} - W_{D,\ell}^{(0)}\|\|_{2} \leq (1-(1- \eta_2 \mu)^{k} ) \|\| W^{(0)}_\ell \|\|_2 + \frac{\eta_1}{\eta_2 \mu} max \|\| \nabla_W [ L_S (W^{(k-\ell-1)}_D ) ] \|\|_2 .
> > $$
> >
> > We know under the initialization setting in our paper, $\|\| W_{D,\ell}^{(0)}\|\|_{2}\leq\mathcal{O}(1)$
> >
> > with high-probability, and
> > as long as we can prove the $\ell_{2}$-norm of gradient is bounded, then
> > we can conclude
> > $\|\|W_{D,\ell}^{(k)}- W_{D,\ell}^{(0)} \|\|_{2} \leq \mathcal{O}\big(1\big)$.
> >
> > However, we are not aware of works in which they control the size of $\|\| \nabla_{W_{\ell}}[ L_{S}(W_{D}^{(k-\ell-1)} )] \|\|_{2}$
> >
> > when the non-convex interactions between model parameters across the hidden layers are allowed.
> > To the best of our knowledge, we know the work [1] deals with the three layer case under this setting.
> > But we are not sure if the techniques they used in this paper can be generalized to arbitrary $L$-hidden layer setting.
> > However, there is an empirical evidence that indeed
> > $\|\|W_{D,\ell}^{(k)}- W_{D,\ell}^{(0)} \|\|_{2} \leq \mathcal{O}\big(1\big)$,
> > when we regularize solely on the model parameter.
> > Reviewers can refer the Figure $3$ of paper [2].\
> > \
> > We admit that this type of explanation is lacking in page $3$, we will incorporate this type of explanations in our revised manuscript.
> >
> > [1] Allen-Zhu, Zeyuan, Yuanzhi Li, and Yingyu Liang. "Learning and generalization in overparameterized neural networks, going beyond two layers." Advances in neural information processing systems (2019).
> >
> > [2] Hu, Wei, Zhiyuan Li, and Dingli Yu. "Simple and effective regularization methods for training on noisily labeled data with generalization guarantee." arXiv preprint arXiv:1905.11368 (2019).

---

> > > ### Author Response · Authors · 2021-11-15
> > > **Responses to reviewer 3 (part 3)**
> > >
> > > # **Q2 (2)**
> > > One detail which is unclear, and would be nice if authors could clarify, how is the factor  "balanced-out" in the part of the proof where you need to track the ridge regressor (second result of Theorem 3.8): It seems to be important because both predictors have to be of the same scale. Is it done by having an extremely small step size of order ? In most NTK-style proofs this factor is not present.
> > >
> > > # **A2 (2)**
> > > At initialization, the factor $\sqrt{m}$ in Equation $(2)$ in the paper can be incorporated into output layer.
> > > Recall that the entry of output layer is initialized by $v_{j}\sim \mathcal{N}(0,\frac{\omega}{m})$
> > > for $j\in\\{1,\dots,m\\}$.
> > > If we let $v_{j}^{\prime}=\sqrt{m}v_{j}$, then, it is easy to see $v_{j}^{\prime}\sim\mathcal{N}(0,\omega)$.
> > > By the important Lemma $I.1.$ in the appendix, we know $\\| x_{\ell,i} \\|_{2}=\mathcal{O}(1)$,
> > >
> > > for any $i\in\\{1,2,\dots,n\\}$ and $\ell\in\\{1,\dots,L\\}$.
> > > So for each data-set index $i$, we know $u_{i}(0)=\big(v^{\prime}\big)^{\top}x_{L,i}\sim\mathcal{N}(0,\mathcal{O}(\omega))$,
> > > where $u_{i}(0)$ is the output of neural network with $i^{\text{th}}$ sample at the initialization.\
> > > \
> > > And we can easily prove the $\\ell_{2}$ norm $\|\|u(0) \|\|$ has roughly the order $\mathcal{O}(\sqrt{n\omega})$.
> > > The key idea for proving the second result in Theorem $3.8$ is to write the distance between $u_{i,D}(k)$ (where $D$ is to denote the prediction is obtained from regularized GD rule) and kernel regressor $B:=H_L^{\infty} ( C \mu \cdot \mathcal{I} + H_L^{\infty} )^{-1}y$
> > > in terms of NTK matrix ${H}_{L}^{\infty}$, which is as follows:
> > >
> > > $$
> > > {u}_{D}(k)-{B} = ( (1-\eta_2 \mu L)\cdot \mathcal{I} - m\eta_1 H^{\infty}_L )^{k} (u_D(0) - B ) + e_D(k)
> > > $$
> > >
> > > See the detailed proof for the derivation of this equality in the Appendix $E$.\
> > > \
> > > The above equation describes how the regularized estimator evolves to fit the kernel regressor as iteration of algorithm goes by. We can bound the $\ell_{2}$-norm of residual term $e_{D}(k)$ as $\mathcal{O}(1/n)$, and show that the $\ell_{2}$ norm of the first term on the RHS of the equation decays at the rate $\mathcal{O}\big( \sqrt{n}\big( 1 - \eta_{2} \mu L \big)^{k}\big)$.
> > > Here the $\sqrt{n}$ comes from the bound $\|\|B\|\|\leq\mathcal{O}(\sqrt{n})$, and since we know $\|\|u(0)\|\|$ has $\mathcal{O}(\sqrt{n\omega})$ has with $\omega\leq1$. This yields the claim.
> > > A detailed investigation of the proof unveils that we do not use the specific choice of order of $\mu_{1}$, which is involved with the factor $\frac{1}{m}$ for deriving the result.

---

> > > > ### Comment · Reviewer_ytWV · 2021-11-22
> > > > **Re: Responses to reviewer 3 (part 3)**
> > > >
> > > > Thanks for explanation, now it's somewhat clearer. It seems that tuning step sizes and $k$ allows for a correct tracking.

---

> > > ### Comment · Reviewer_ytWV · 2021-11-22
> > > **Re: Responses to reviewer 3 (part 2)**
> > >
> > > Thank you for your explanation. Now it's clearer that L2 regularization helps to control $||W_{D,\ell}^{(k)} - W_{D,\ell}^{(0)}||_2$. I guess the norm $|| \nabla_W [ L_S (W^{(k-\ell-1)}_D ) ] ||_2$ can be bounded simply when it is controlled by the loss (as would be in case of smooth activations) and descent lemma.

---

### Official Review · Reviewer_eixk · 2021-11-02

**Correctness:** 4
**Technical Novelty And Significance:** 3
**Empirical Novelty And Significance:** 3
**Recommendation:** 8
**Confidence:** 3

**Main Review:**

This paper is well written and shows some novel results. Although I didn't go over all the proof, those theoretical seems correct to me. I also appreciate that the authors provide a detailed comparison with [Hu et. al, 2021] in order to highlight the contribution of this paper. The following are some detailed comments.

1. The condition about $\mu$ in Eq.~(12) seems a bit strange to me. It basically means that $\mu$ should increase with $n$ at a certain speed, which is kind of a strong restriction on the regularized factor. It would be nice to see some explanation on why such requirement is needed and how the performance changes when such requirement is not satisfied.

2. I notice that the noise considered in this paper follows i.i.d. Gaussian. I wonder whether the stated theoretical results (e.g., generalization error of unregularized case is bounded away from zero by a constant) are likely to hold for other types of noise.

3. The presentation of Fig. 1(c) is not very clear. Fig. 1(c) is used to illustrate that the prediction risk goes to zero as the number of training samples $n$ increases. However, it is not clear with the x-axis being "Epochs". I suggest plotting a figure with the x-axis being the number of training samples $n$.

**Summary Of The Paper:**

This paper considers fully connected DNNs with finite equal width $m$ in each layer. Gradient descent is used to minimize the unregularized or L2-norm regularized training error. For unregularized situations, this paper proves that the training error decreases to zero at a linear rate and the generalization error is bounded away from zero by some constant factor if the training process is too long or too short when there exists noise. For the L2-norm regularized situation, this paper provides an upper bound on the training error that consists of two terms, one converges linearly to 0, the other is a constant. Additionally, this paper also proves that the training dynamics of the regularized neural network can approximate the corresponding kernel ridge regression. For the generalization performance of the regularized situation, this paper proves that with enough large width, the prediction risk decreases with the number of training samples $n$ at least at the speed of $n^{-d/(2d-1)}$, which is a little bit faster than the existing results $n^{-1/2}$. Numerical results are provided to support those theoretical results.

**Summary Of The Review:**

This paper is a good paper with sufficient contribution. Some minor places could be stated more clearly.

---

> ### Author Response · Authors · 2021-11-14
> **Responses to Reviewer 2**
>
> # **Q1**
> The condition about  in Eq.~(12) seems a bit strange to me.
> It basically means that $\mu$ should increase with $n$ at a certain speed, which is kind of a strong restriction on the regularized factor.
> It would be nice to see some explanation on why such requirement is needed and how the performance changes when such requirement is not satisfied.
>
> # **A1**
> It is a reasonable concern.
> We find out part of this question can be answered through the answer to the second question of reviewer $1$.
> We are guessing your confusion probably comes from the un-normalized optimization problem we are solving in page $8$ of our manuscript.
> If we normalize the optimization problem, we can get the optimal choice of tuning parameter $\mu$ which goes to $0$ as $n$ increases.
> And as answered to the second question from Reviewer $1$, this choice of tuning parameter affects the quality of convergence rate of prediction error
> we can get!
> Hope this will be the answer to your concern, and we will revise our manuscript for more clear explanation on this part.
>
> ---
>
> # **Q2**
> I notice that the noise considered in this paper follows i.i.d. Gaussian.
> I wonder whether the stated theoretical results (e.g., generalization error of unregularized case is bounded away from zero by a constant) are likely to hold for other types of noise.
>
> # **A2**
> Yes, the statement in Theorem $3.6$ will not change as long as the random variable's second moment is bounded by some positive constant.
> A close investigation of our proof on Theorem $3.6$ unveils that the part that is involved random noise is when proving Lemma $1.9$ and controlling $\|\|\Delta_{11}\|\|_{2}$ in page $27\sim 28$.
> And in these parts, we only use the bounded second moment of noise.\
> \
> But as for Theorem $3.9$, it remains an open question whether the statement still holds or not.
> The reason is as follows:
> the proof of the result that we present in the answer to the second question of Reviewer $1$, that is:
>
> $$
>     \|\| \widehat{f} - f^*  \|\| _{n}^{2}
>     = \mathcal{O}_\mathbb{P}\bigg( \big(\mu^{\prime}\big)^{2}\bigg),
> $$
> where $\widehat{f}$ is a kernel regressor, requires the sub-gaussian properties of random error.
>
> ---
> # **Q3**
> The presentation of Fig. 1(c) is not very clear. Fig. 1(c) is used to illustrate that the prediction risk goes to zero as the number of training samples  increases. However, it is not clear with the x-axis being "Epochs". I suggest plotting a figure with the x-axis being the number of training samples .
>
> # **A3**
> We agree to the reviewer's opinion, and will change the Fig. 1(c) accordingly.

---

> > ### Comment · Reviewer_eixk · 2021-11-24
> > **Re: Responses to Reviewer 2**
> >
> > Thanks for your response. I'll keep my score.

---

### Official Review · Reviewer_FGhp · 2021-11-03

**Correctness:** 4
**Technical Novelty And Significance:** 3
**Empirical Novelty And Significance:** Not applicable
**Recommendation:** 8
**Confidence:** 3

**Main Review:**

Overall I think this paper is well organized and well written. The proof of this paper seems solid. However, I still have the following concerns:

1.  Theorem 3.6 suggests that GD fails if the DNN is trained for either too short or too long. Could you add a comment on whether early stopping is hard to implement in this scenario?

2. Does the regularization sensitive to the hyperparameter $\mu$? In section 3.3., the authors directly choose $mu = \Theta(n^{\frac{d-1}{2d-1}})$. What if we choose $\mu$ to be larger or smaller than a $log(n)$ factor?

3. Typo: Theorem 3.6 this should be $O((\frac{\lambda_{0}\delta}{n})^{2/3})$

**Summary Of The Paper:**

In this paper, the authors analyze the convergence rate of both the unregularized and the L2 regularized gradient descent for a regression problem. Under a positivity assumption of NTK,  this paper shows that without early stopping, the vanilla GD may fail. This can be solved by using L2 regularization and could achieve a better convergence rate.

**Summary Of The Review:**

Overall I think this paper is well written, and the proof seems solid. Given Hu et al. [2021], the results of this paper are not that surprising. But the proof technique is different. I still have some concerns. So currently, I would like to suggest a rejection, but I am open to discussion and willing to change my score.

---

> ### Author Response · Authors · 2021-11-14
> **Responses to reviewer 1**
>
> # **Q1**
> Theorem $3.6$ suggests that GD fails if the DNN is trained for either too short or too long. Could you add a comment on whether early stopping is hard to implement in this scenario?
>
> # **A1**
> Analytically deriving a data-dependent stopping time in our scenario requires further studies, since we need a sharp characterization of eigen-distribution of NTK of ReLU DNN, denoted as $\mathbf{H}^{\infty}_{L}$ in our paper.
> It would be greatly appreciated if you can refer the comments below Theorem $4.2.$ of paper $[1]$ in shallow-neural network and Equation number $(6)$ in $[2]$ in kernel regression context.
> To the best of our knowledge, this characterization of eigen-distribution has not been studied in the literature yet and we think this is beyond the scope of the topic of our paper.\
> \
> From the practitioner's point of view, we can simply stop the GD before it hits the nearly-zero empirical risk, but monitor the performance on a held-out validation sample, and stop training early when a minimum on the validation sample has been reached.
>
> [1] Hu, Tianyang, Wenjia Wang, Cong Lin, and Guang Cheng. "Regularization Matters: A Nonparametric Perspective on Overparametrized Neural Network." In International Conference on Artificial Intelligence and Statistics, pp. 829-837. PMLR, 2021
>
> [2] Raskutti, Garvesh, Martin J. Wainwright, and Bin Yu. "Early stopping and non-parametric regression: an optimal data-dependent stopping rule." The Journal of Machine Learning Research 15.1 (2014): 335-366.
>
> ---
> # **Q2**
> Does the regularization sensitive to the hyperparameter $\mu$?
> In section $3.3.$, the authors directly choose $\mu = \Theta\big( n^{\frac{d-1}{2d-1}} \big)$.
> What if we choose  to be larger or smaller than a $\log(n)$ factor?
>
> # **A2**
> A particular choice of $\mu = \Theta\big( n^{\frac{d-1}{2d-1}} \big)$ in Section $3.3$ is for obtaining an optimal minimax rate for the prediction error.
> It is known in non-parametric regression literature that as long as the regression function $f^{\star}$ is bounded in RKHS norm: that is,
> $\|\| f^{\star} \|\|_{\mathcal{H}}^{2}<C$ for some constant $C>0$, under sub-gaussian error assumption, we have
>
> $$
>  \|\| \widehat{f} - f^* \|\|_{n}^{2}
> = \mathcal{O}_\mathbb{P}\bigg( \big(\mu^{\prime}\big)^{2}\bigg),
> $$
>
> where $\|\| \cdot \|\|_{n}$ is an empirical norm, and $\widehat{f}$ is a minimizer of a following optimization problem,
>
> $$
> \frac{1}{n} \sum_{i=1}^{n}  ( y_{i} - f(x_{i}) )^{2} + (\mu^{\prime})^{2} \cdot \|\| f \|\|_{\mathcal{H}}^{2}
> $$
>
> Here, $f$ belongs to RKHS induced from NTK function.
> Reviewers can refer Lemma 10.2 in [3] for this result.
> Then, we can control the distance between the squared empirical norm and squared $L_{2}$ norm uniformly over the function within a certain neighborhood of $f^{\star}$ in a $\mathcal{O}\big(\frac{1}{n}\big)$ accuracy with high-probability.
> Furthermore, we can prove $\widehat{f}$ belongs to the certain neighborhood of $f^{\star}$ with positive radius.\
> \
> For this part, reviewers can refer the Theorem $2.2$ in the book [3].
> It is worth noting that the normalization scheme in the above optimization problem is different from the optimization problem we solve in page $8$ in our paper.
> In our paper, we simply follow the setting of [1], but we can simply set $\big(\mu'\big)^{2}=\frac{C\mu}{n}$, where $\mu$ is the notation we use in our paper.\
> \
> As reviewer $2$ had pointed out, if we choose $\mu = \Theta\big(\log(n)\big)$, following the argument above, we can easily see the optimal rate we can achieve is $\mathcal{O}\big( \sqrt{\frac{\log(n)}{n}}\big)$, which is slower than $\mathcal{O}\big( n^{-1/2} \big)$ or the minimax rate we achieve in our paper.
>
> [3] Sara van de Geer. Empirical Processes in M-estimation. Vol. 6. Cambridge university press, 2000.
>
> ---
> # **Q3**
> Typo: Theorem 3.6 this should be $\mathcal{O}{\color{red}\big(}\big(\frac{\lambda_{0}\delta}{n}\big)^{2/3}{\color{red}\big)}$.
>
> # **A3**
> Thanks for pointing out the typo!
> We corrected it accordingly as suggested.

---

> > ### Comment · Reviewer_FGhp · 2021-11-23
> > **Re: Responses to reviewer 1**
> >
> > Thanks for the explanation. Since my concerns have been addressed, I would like to raise my score to 8.

---

### Author Response · Authors · 2021-11-20
**Summary of Revision**

Dear Reviewers and AC

First of all, we greatly appreciate for the insightful comments from reviewers on our work.
We re-upload the revised version of our manuscript, and all the revised parts are colored in red.
Followings are the main summary of the revised manuscript.

1. The title of paper is changed :
From :  Generalization of overparametrized deep neural network under noisy observation.
To : A non-parametric regression viewpoint : Generalization of overparametrized deep ReLU network under noisy observation.

2. Following Reviewer 1, we add the additional comments on early stopping in unregularized case in the Remark  3.9 and 3.10.

3. Following Reviewer 4's suggestion, we provide the proof idea at the beginning of proofs of each theorems in the Appendix.

4. We provide the summary of the references suggested by Reviewer 3 to the "Additional related works".

5. Add additional explanations on the complexity of RKHS in Subsection 3.2 by following Reviewer 3.

6. Due to the space limit, we moved the numerical experiment part to the Appendix, and note the readers about this in remark 3.4.
Following Reviewer 2's suggestion, we changed the plot of (c).

7. Following Reviewer 1 and 2's comments, we add the additional remarks on the choice of $\mu$ in remark 3.13.

8. We add the additional remarks on the $\|\|W_{D,\ell}^{(k)} - W_{D,\ell}^{(0)}\|\|_{2}\leq\mathcal{O}(1)$ when we solely regularize the model parameter, inspired by Reviewer 3 in the Appendix C.

9. Since we admit that the condition on $m$ in each Theorems can cause confusions to the readers, we change the condition to the ratio of $m$ and its logarithm accordingly.

10. We take care of all the minor comments from Reviewer 1, Reviewer 2, Reviewer 3, and Reviewer 4 through footnote or directly incorporate the answers to the comments into the paragraphs in the revised paper.

Thank you all for the valuable suggestions, again!  Please let us know if you have additional questions.
Thank you,

Authors

---

### Decision · Program_Chairs · 2022-01-20

**Decision:**

Accept (Poster)

**Comment:**

The paper contributes a theoretical understanding of training over-parametrized deep neural networks with rectified linear unit (ReLU) activations using gradient descent with respect to square loss in the neural tangent kernel (NTK) regime. Authors consider a non-parametric regression framework wherein the labels are generated using a ground truth function, which is assumed to be in the RKHS associated with the NTK, perturbed with noise. Authors show that gradient descent based training without early stopping fails whereas \ell-2 regularized gradient descent achieved minimax optimal convergence rate. The paper is clearly written and the results are solid. Overall, a good paper.